# Learning accurate path integration in ring attractor models of the head direction system

**Pantelis Vafidis[1,2,3]\*, David Owald[4,5,6], Tiziano D'Albis[2,3†], Richard Kempter[2,3,6]\*†**

[1]Computation and Neural Systems, California Institute of Technology, Pasadena, United States; [2]Bernstein Center for Computational Neuroscience, Berlin, Germany; [3]Institute for Theoretical Biology, Department of Biology, Humboldt-Universität zu Berlin, Berlin, Germany; [4]Institute of Neurophysiology, Charité – Universitätsmedizin Berlin, corporate member of Freie Universität Berlin and Humboldt-Universität zu Berlin, and Berlin Institute of Health, Berlin, Germany; [5]NeuroCure, Charité - Universitätsmedizin Berlin, Berlin, Germany; [6]Einstein Center for Neurosciences, Berlin, Germany

**Abstract** Ring attractor models for angular path integration have received strong experimental support. To function as integrators, head direction circuits require precisely tuned connectivity, but it is currently unknown how such tuning could be achieved. Here, we propose a network model in which a local, biologically plausible learning rule adjusts synaptic efficacies during development, guided by supervisory allothetic cues. Applied to the *Drosophila* head direction system, the model learns to path-integrate accurately and develops a connectivity strikingly similar to the one reported in experiments. The mature network is a quasi-continuous attractor and reproduces key experiments in which optogenetic stimulation controls the internal representation of heading in flies, and where the network remaps to integrate with different gains in rodents. Our model predicts that path integration requires self-supervised learning during a developmental phase, and proposes a general framework to learn to path-integrate with gain-1 even in architectures that lack the physical topography of a ring.

**\*For correspondence:**
pvafeidi@caltech.edu (PV);
r.kempter@biologie.hu-berlin.
de (RK)

†These authors contributed
equally to this work

**Competing interest:** The authors
declare that no competing
interests exist.

**Reviewing Editor:** Srdjan
Ostojic, Ecole Normale
Superieure Paris, France

## Editor's evaluation

This paper will be of interest to neuroscientists studying the navigation system, and in particular, those who study the ability of animals to path integrate. This study proposes an elegant synaptic plasticity rule that maintains the connectivity required for path integration by integrating visual and self-motion input arriving at different dendritic locations in a neuron. This idea is applied to the central complex of *Drosophila*, a well-characterized experimental system.

## Introduction

Spatial navigation is crucial for the survival of animals in the wild and has been studied in many model organisms (*Tolman, 1948*; *O'Keefe and Nadel, 1978*; *Gallistel, 1993*; *Eichenbaum, 2017*). To orient themselves in an environment, animals rely on external sensory cues (e.g. visual, tactile, or auditory), but such allothetic cues are often ambiguous or absent. In these cases, animals have been found to update internal representations of their current location based on idiothetic cues, a process that is termed path integration (PI, *Darwin, 1873*; *Mittelstaedt and Mittelstaedt, 1980*; *McNaughton et al., 1996*; *Etienne et al., 1996*; *Neuser et al., 2008*; *Burak and Fiete, 2009*). The head direction

(HD) system partakes in PI by performing one of the computations required: estimating the current HD by integrating angular velocities; namely angular integration. Furthermore, head direction cells in rodents and flies provide an internal representation of orientation that can persist in darkness (*Ranck, 1984*; *Mizumori and Williams, 1993*; *Seelig and Jayaraman, 2015*).

In rodents, the internal representation of heading takes the form of a localized "bump" of activity in the high-dimensional neural manifold of HD cells (*Chaudhuri et al., 2019*). It has been proposed that such a localized activity bump could be sustained by a ring attractor network with local excitatory connections (*Skaggs et al., 1995*; *Redish et al., 1996*; *Hahnloser, 2003*; *Samsonovich and McNaughton, 1997*; *Song and Wang, 2005*; *Stringer et al., 2002*; *Xie et al., 2002*), resembling reverberation mechanisms proposed for working memory (*Wang, 2001*). Ring attractor networks used to model HD cells fall in the theoretical framework of continuous attractor networks (*Amari, 1977*; *Ben-Yishai et al., 1995*; *Seung, 1996*). In this setting, HD cells can update the heading representation in darkness by smoothly moving the bump around the ring obeying idiothetic angular-velocity cues.

Interestingly, a physical ring-like attractor network of HD cells was observed in the *Drosophila* central complex (CX, *Seelig and Jayaraman, 2015*; *Green et al., 2017*; *Green et al., 2019*; *Franconville et al., 2018*; *Kim et al., 2019*; *Fisher et al., 2019*; *Turner-Evans et al., 2020*). Notably, in *Drosophila* (from here on simply referred to as 'fly'), HD cells (named E-PG neurons, also referred to as 'compass' neurons) are physically arranged in a ring, and an activity bump is readily observable from a small number of cells (*Seelig and Jayaraman, 2015*). Moreover, as predicted by some computational models (*Skaggs et al., 1995*; *Samsonovich and McNaughton, 1997*; *Stringer et al., 2002*; *Song and Wang, 2005*), the fly HD system also includes cells (named P-EN1 neurons) that are conjunctively tuned to head direction and head angular velocity. We refer to these neurons as head rotation (HR) cells because of their putative role in shifting the HD bump across the network according to the head's angular velocity (*Turner-Evans et al., 2017*; *Turner-Evans et al., 2020*).

A model for PI needs to both sustain a bump of activity and move it with the right speed and direction around the ring. The latter presents a great challenge, since the bump has to be 'pushed' for the right amount starting from any location and for all angular velocities. Therefore, ring attractor models that act as path integrators require that synaptic connections are precisely tuned (*Hahnloser, 2003*). If the circuit was completely hardwired, the amount of information that an organism would need to genetically encode connection strenghts would be exceedingly high. Additionally, it would be unclear how these networks could cope with variable sensory experiences. In fact, remarkable experimental studies in rodents have shown that when animals are placed in an augmented reality environment where visual and self-motion information can be manipulated independently, PI capabilities adapt accordingly (*Jayakumar et al., 2019*). These findings suggest that PI networks are able to self-organize and to constantly recalibrate. Notably, in mature flies there is no evidence for such plasticity (*Seelig and Jayaraman, 2015*) — however, the presence of plasticity has not been tested in young animals.

Here, we propose that a simple local learning rule could support the emergence of a PI circuit during development and its re-calibration once the circuit has formed. Specifically, we suggest that accurate PI is achieved by associating allothetic and idiothetic inputs at the cellular level. When available, the allothetic sensory input (here chosen to be visual) acts as a 'teacher' to guide learning. The learning rule is an example of self-supervised multimodal learning, where one sense acts as a teaching signal for the other and the need for an external teacher is obviated. It exploits the relation between the allothetic heading of the animal (given by the visual input) and the idiothetic self-motion cues (which are always available), to learn how to integrate the latter.

The learning rule is inspired by previous experimental and computational work on mammalian cortical pyramidal neurons, which are believed to associate inputs to different compartments through an in-built cellular mechanism (*Larkum, 2013*; *Urbanczik and Senn, 2014*; *Brea et al., 2016*). In fact, it was shown that in layer 5 pyramidal cells internal and external information about the world arrive at distinct anatomical locations, and active dendritic gating controls learning between the two (*Doron et al., 2020*). In a similar fashion, we propose that learning PI in the HD system occurs by associating inputs at opposite poles of compartmentalized HD neurons, which we call 'associative neurons' (*Urbanczik and Senn, 2014*; *Brea et al., 2016*). Therefore, to accomplish PI the learning rule relies on structural inductive biases in terms of the morphology and arborization of HD cells.

In summary, here we show for the first time how a biologically plausible synaptic plasticity rule enables to learn and maintain the complex circuitry required for PI. We apply our framework to the fly HD system because it is well characterized; yet our model setting is general and can be used to learn PI in other animal models once more details about the HD circuit there are known (*Abbott et al., 2020*). We find that the learned network is a ring attractor with a connectivity that is strikingly similar to the one found in the fly CX (*Turner-Evans et al., 2020*) and that it can accurately path-integrate in darkness for the entire range of angular velocities that the fly displays. Crucially, the learned network accounts for several key findings in the experimental literature, and it generates predictions, including the presence of plasticity in young animals, that could be tested experimentally.

## Results

To illustrate basic principles of how PI could be achieved, we study a computational model of the HD system and show that synaptic plasticity could shape its circuitry through visual experience. In particular, we simulate the development of a network that, after learning, provides a stable internal representation of head direction and uses only angular-velocity inputs to update the representation in darkness. The internal representation of heading (after learning) takes the form of a localized bump of activity in the ring of HD cells. All neurons in our model are rate-based, i.e., spiking activity is not modeled explicitly.

### Model setup

The gross model architecture closely resembles the one found in the fly CX (*Figure 1A*). It comprises HD cells organized in a ring, and HR cells organized in two wings. One wing is responsible for leftward and the other for rightward movement of the internal heading representation. HD cells receive visual input from the so-called 'ring' neurons; this input takes the form of a disinhibitory bump centered at the current HD (*Figure 1B*, *Omoto et al., 2017*; *Fisher et al., 2019*). The location of this visual bump in the network is controlled by the current head direction. We simulate head movements by sampling head-turning velocities from an Ornstein-Uhlenbeck process (Materials and methods), and we provide the corresponding velocity input to the HR cells (*Figure 1C*). HR cells provide direct input to HD cells, and HR cells also receive input from HD cells (*Figure 1A*). Both HR and HD cells receive global inhibition, which is in line with a putative 'local' model of HD network organization (*Kim et al., 2017*). The connections from HR to HD cells ($W^{HR}$) and the recurrent connections among HD cells ($W^{rec}$) are assumed to be plastic. The goal of learning is to tune these plastic connections so that the network can achieve PI in the absence of visual input.

The unit that controls plasticity in our network is an 'associative neuron'. It is inspired by pyramidal neurons of the mammalian cortex whose dendrites act, via backpropagating action potentials, as coincidence detectors for signals arriving from different layers of the cortex and targeting different compartments of the neuron (*Larkum et al., 1999*). Paired with synaptic plasticity, coincidence detection can lead to long-lasting associations between these signals (*Larkum, 2013*). To map the morphology of a cortical pyramidal cell to the one of a HD cell in the fly, we first point out that all relevant inputs arrive at the dendrites of HD cells within the ellipsoid body (EB) of the fly (*Xu, 2020*) moreover, the soma itself is externalized in the fly brain, and it is unlikely to contribute considerably to computations (*Gouwens and Wilson, 2009*; *Tuthill, 2009*). We thus link the dendrites of the pyramidal associative neuron to the axon-distal dendritic compartment of the associative HD neuron in the fly, and we link the soma of the pyramidal associative neuron to the axon-proximal dendritic compartment of the associative HD neuron in the fly. Furthermore, we assume that the axon-proximal compartment is electrotonically closer to the axon initial segment, and therefore, similarly to the somatic compartment in pyramidal neurons, inputs there can more readily initiate action potentials. Note that our model does not require *active* backpropagation of action potentials — *passive* spread of voltage to the axon-distal compartment would be sufficient (for details, see Materials and methods and Discussion). We also assume that associative HD cells receive visual input ($I^{vis}$) in the axon-proximal compartment, and both recurrent input ($W^{rec}$) and HR input ($W^{HR}$) in the axon-distal compartment; accordingly, we model HD neurons as two-compartment units (*Figure 1D*). The associative neuron can learn the synaptic weights of the incoming connections in the axon-distal compartment, therefore, as mentioned, we let $W^{rec}$ and $W^{HR}$ be plastic.

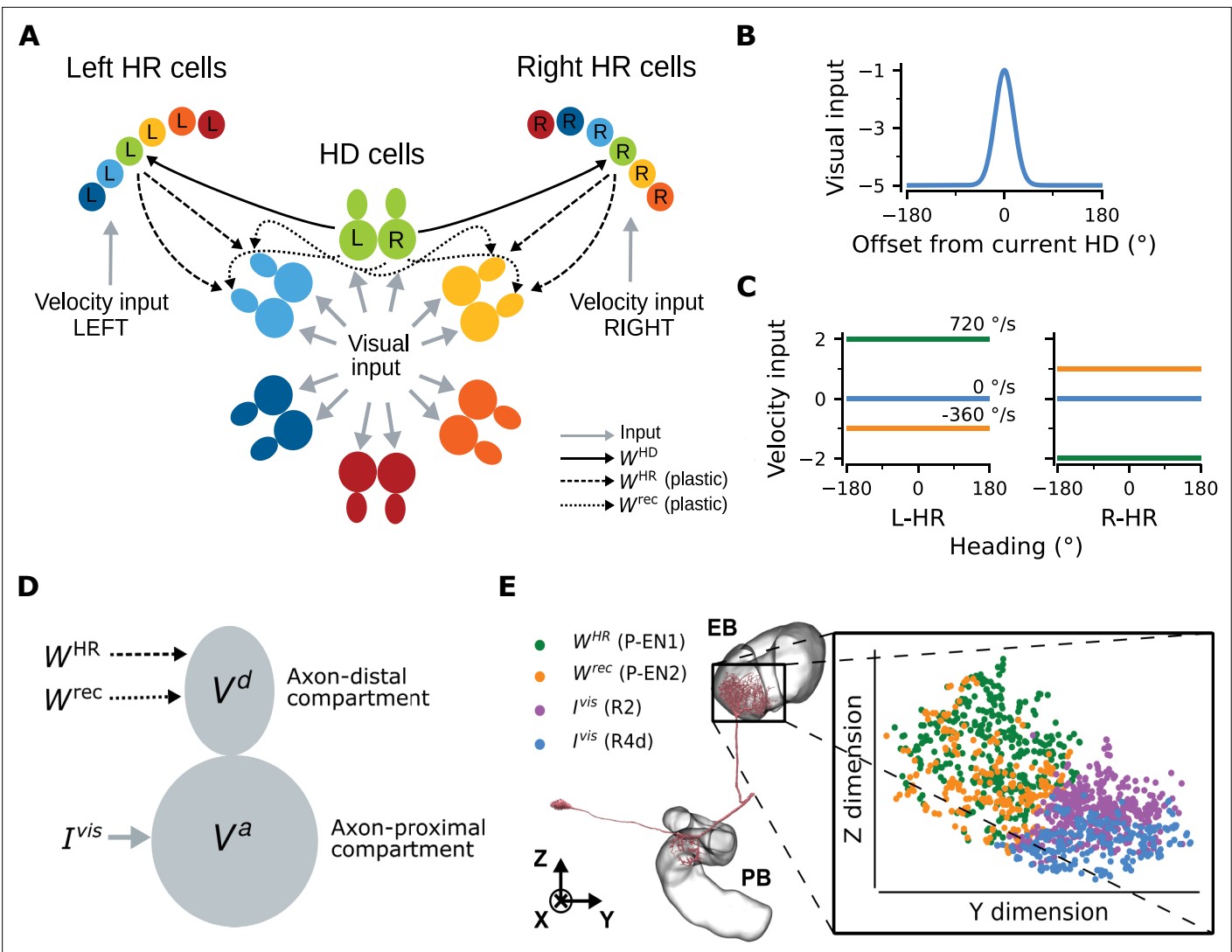

**Figure 1.** Network architecture. (**A**) The ring of HD cells projects to two wings of HR cells, a leftward (Left HR cells, abbreviated as L-HR) and a rightward (Right HR cells, or R-HR), so that each wing receives selective connections only from a specific HD cell (L: left, R: right) for every head direction. For illustration purposes, the network is scaled-down by a factor of 5 compared to the cell numbers $N^{HR} = N^{HD} = 60$ in the model. The schema shows the outgoing connections ($W^{HD}$ and $W^{rec}$) only from the green HD neurons and the incoming connections ($W^{HR}$ and $W^{rec}$) only to the light blue and yellow HD neurons. Furthermore, the visual input to HD cells and the velocity inputs to HR cells are indicated. (**B**) Visual input to the ring of HD cells as a function of radial distance from the current head direction (see *Equation 5*). (**C**) Angular-velocity input to the wings of HR cells for three angular velocities: 720 (green), 0 (blue), and -360 (orange) deg/s (see *Equation 10*). (**D**) The associative neuron: $V^a$ and $V^d$ denote the voltage in the axon-proximal (i.e. closer to the axon initial segment) and axon-distal (i.e. further away from the axon initial segment) compartment, respectively. Arrows indicate the inputs to the compartments, as in (**A**), and $I^{vis}$ is the visual input current. (**E**) Left: skeleton plot of an example HD (E-PG) neuron (Neuron ID =416642425) created using neuPrint (*Clements et al., 2020*) the ellipsoid body (EB) and protocerebral bridge (PB) are overlaid. Right: zoomed in area in the EB indicated by the box, showing postsynaptic locations in the EB for this E-PG neuron; for details, see Methods. The neuron receives recurrent and HR input (green and orange dots, corresponding to inputs from P-EN1 and P-EN2 cells, respectively) and visual input (purple and blue dots, corresponding to inputs from visually responsive R2 and R4d cells, respectively) in distinct spatial locations.

The online version of this article includes the following video and figure supplement(s) for figure 1:

**Figure supplement 1.** Separation of axon-proximal and axon-distal inputs to HD (E-PG) neurons in the *Drosophila* EB.

**Figure 1—video 1.** A three-dimensional rotating video of the synapse locations in Figure 1E.

https://elifesciences.org/articles/69841/figures#fig1video1

We find that the assumption of spatial segregation of postsynapses of HD cells is consistent with our analysis of EM data from the fly (*Xu, 2020*). For an example HD (E-PG) neuron, *Figure 1E* depicts that head rotation and recurrent inputs (mediated by P-EN1 and P-EN2 cells, respectively [*Turner-Evans et al., 2020*]) contact the E-PG cell in locations within the EB that are distinct compared to those of visually responsive neurons R2 and R4d (*Omoto et al., 2017*; *Fisher et al., 2019*), as hypothesized. The same pattern was observed for a total of 16 E-PG neurons (one for each 'wedge' of the EB) that we analyzed (*Figure 1—figure supplement 1A*). To further support the assumption that visual inputs are separated from recurrent and HR-to-HD inputs, we perform binary classification between the two classes, using SVMs (for details, see Materials and methods). *Figure 1—figure supplement 1B* shows that predicting class identity from spatial location alone in held-out test data is excellent (test accuracy >0.95 across neurons and model runs).

The connections from HD to HR cells ($W^{HD}$) are assumed to be fixed, and HR cells are modeled as single-compartment units. Projections are organized such that each wing neuron receives input from only one specific HD neuron for every HD (*Figure 1A*). This simple initial wiring makes HR cells conjunctively tuned to HR and HD, and we assume that it has already been formed, for example, during circuit assembly. We note that the conditions for 1-to-1 wiring and constant amplitude of the HD-to-HR connections can be relaxed, because the learning rule can balance asymmetries in the initial architecture (see Appendix 3). In addition, the connections carrying the visual and angular velocity inputs are also assumed to be fixed. Although plasticity in the visual inputs has been shown to exist (*Fisher et al., 2019*; *Kim et al., 2019*), here we focus on how the path-integrating circuit itself originally self-organizes. Therefore, to simplify the setting and without loss of generality, we assume a fixed anchoring to environmental cues as the animal moves in the same environment (for details, see Discussion).

In our model, the visual input acts as a supervisory signal during learning (as in *D'Albis and Kempter, 2020*), which is used to change weights of synapses onto the axon-distal compartment of HD cells. We utilize the learning rule proposed by *Urbanczik and Senn, 2014* (for details, see Materials and methods), which tunes the incoming synaptic connections in the axon-distal compartment in order to minimize the discrepancy between the firing rate of the neuron $f(V^a)$ (where $V^a$ is the axon-proximal voltage, primarily controlled by the visual input) and the prediction of the firing rate by the axon-distal compartment from axon-distal inputs alone, $f(pV^d)$ (where $p$ is a constant and $V^d$ is the axon-distal voltage, which depends on head rotation velocity). From now on, we refer to this discrepancy as 'learning error', or simply 'error' (*Equation 18*; in units of firing rate). The synaptic weight change $\Delta W_{pre,post}$ from a presynaptic (HD or HR) neuron to a postsynaptic HD neuron is then given by:

$$\Delta W_{pre,post} = \eta \left[ f(V^a_{post}) - f(pV^d_{post}) \right] P_{pre} \tag{1}$$

where $\eta$ is the constant learning rate and $P_{pre}$ is the postsynaptic potential from the presynaptic neuron. When implementing this learning rule, we low-pass filter the prospective weight change $\Delta W_{pre,post}$ to ensure smoothness of learning.

Importantly, this learning rule is biologically plausible because the firing rate of an associative neuron $f(V^a)$ is locally available at every synapse in the axon-distal compartment due to the (passive or active) backpropagation of axonal activity to the axon-distal dendrites. The other two signals that enter the learning rule are the voltage of the axon-distal compartment $V^d$ and the postsynaptic potential $P$, which are also available locally at the synapse; for details, see Materials and methods. Furthermore, recent behavioral experiments show that conditioning in *Drosophila* (*Zhao et al., 2021*) is not well explained by classical correlation-based plasticity, but it can be well accounted for by predictive synaptic plasticity. The latter is in line with the learning rule utilized here.

## Mature network can path-integrate in darkness

*Figure 2A* shows an example of the performance of a trained network, for the light condition (i.e. when visual input is available; yellow overbars) and for PI in darkness (purple overbars); the performance is quantified by the PI error (in units of degrees) over time. PI error refers to the accumulated difference between the internal representation of heading and the true heading, and it is different from the learning error introduced previously.

A unique bump of activity is clearly present at all times in the HD network (*Figure 2A*, top), in both light and darkness conditions, and this bump moves smoothly across the network for a variable

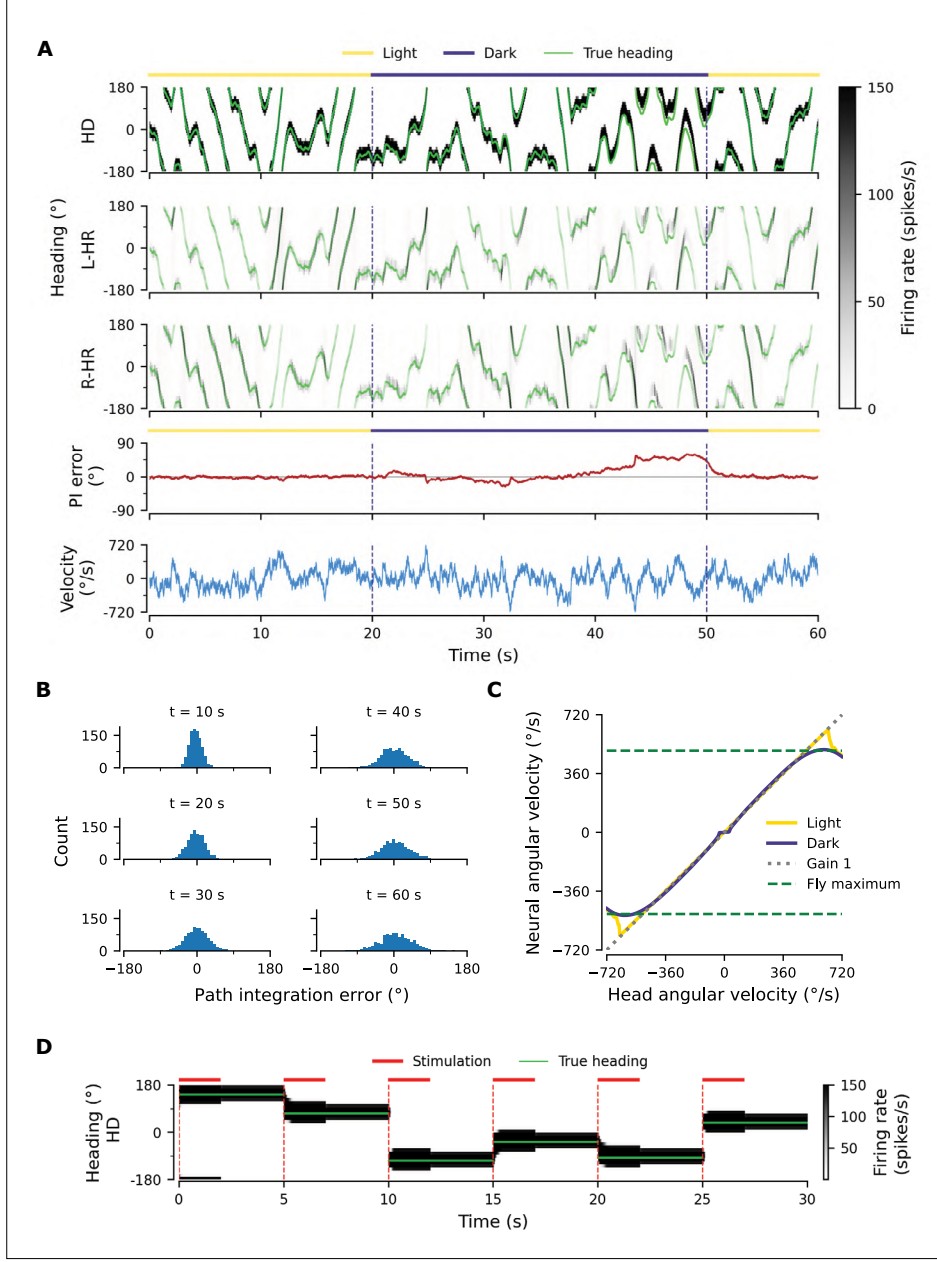

**Figure 2.** Path integration (PI) performance of the network. (**A**) Example activity profiles of HD, L-HR, and R-HR neurons (firing rates gray-scale coded). Activities are visually guided (yellow overbars) or are the result of PI in the absence of visual input (purple overbar). The ability of the circuit to follow the true heading is slightly degraded during PI in darkness. The PI error, that is, the difference between the PVA and the true heading of the animal as well as the instantaneous head angular velocity are plotted separately. (**B**) Temporal evolution of the distribution of PI errors in darkness, for 1000 simulations. The distribution gets wider with time, akin to a diffusion process. We estimate the diffusion coefficient to be $D = 24.5 \, \mathrm{deg}^2/\mathrm{s}$ (see 'Diffusion Coefficient' in Materials and methods). Note that, unless otherwise stated, for this type of plot we limit the range of angular velocities to those normally exhibited by the fly, i.e. |v| < 500 deg/s. (**C**) Relation between head angular velocity and neural angular velocity, i.e., the speed with which the bump moves in the network. There is almost perfect (gain 1) PI in darkness for head angular velocities within the range of maximum angular velocities that are displayed by the fly (dashed green horizontal lines; see Methods). (**D**) Example of consecutive stimulations in randomly permeated HD locations, simulating optogenetic stimulation experiments in *Kim et al., 2017*. Red overbars indicate when the network is stimulated with stronger than normal visual-like input, at the location indicated by the animal's true heading (light green line), while red dashed vertical lines indicate the onset of the stimulation. The network is then left in the dark. Our simulations show that the bump remains at the stimulated positions.

angular velocity (*Figure 2A*, bottom). The position of the bump is defined as the population vector average (PVA) of the neural activity in the HD network. The HD bump also leads to the emergence of bumps in the HR network, separately for L-HR and R-HR cells (*Figure 2A*, second and third panel from top). In light conditions (0–20 s in *Figure 2A*), the PVA closely tracks the head direction of the animal in HD, L-HR, and R-HR cells alike, which is expected because the visual input guides the network activity. Importantly, however, in darkness (20–50 s in *Figure 2A*), the self-motion input alone is enough to track the animal's heading, leading to a small PI error between the internal representation of heading and the ground truth. This error is corrected after the visual input reappears (at 50 s in *Figure 2A*). Such PI errors in darkness are qualitatively consistent with data reported in the experimental literature (*Seelig and Jayaraman, 2015*). The correction of the PI error also reproduces in silico the experimental finding that the visual input (whenever available) exerts stronger control on the bump location than the self-motion input (*Seelig and Jayaraman, 2015*), which suggests that even the mature network does not rely on PI when visual cues are available.

To quantify the accuracy of PI in our model, we draw 1,000 trials, each 60 s long, for constant synaptic weights and in the absence of visual input. We also limit the angular velocities in these trials to retain only velocities that flies realistically display (see dashed green lines in *Figure 2C* and Methods). We then plot the distribution of PI errors every 10 s (*Figure 2B*). We find that average absolute PI errors (widths of distributions) increase with time in darkness, but most of the PI errors at 60 s are within 60 deg of the true heading. This vastly exceeds the PI performance of flies (*Seelig and Jayaraman, 2015*). In flies, the correlation between the PVA estimate and the true heading in darkness varied widely across animals in the range [0.3, 0.95] (*Seelig and Jayaraman, 2015*), whereas for the model it is close to 1. However, it should be noted that the model here corresponds to an ideal scenario that serves as a proof of principle. We will later incorporate irregularities owing to biological factors (asymmetry in the weights, biological noise) that bring the network's performance closer to the fly's behavior.

To further assess the network's ability to integrate different angular velocities, we simulate the system both with and without visual input in 5 s intervals during which the angular velocity is constant. We then compute the average movement velocity of the bump across the network, that is the neural velocity, and compare it to the real velocity provided as input. *Figure 2C* shows that the network achieves a PI gain (defined as the ratio between neural and real velocity) close to 1 both with and without supervisory visual input, meaning that the neural velocity matches very well the angular velocity of the animal, for all angular velocities that are observed in experiments ($|v| < 500$ deg/s for walking and flying) (*Geurten et al., 2014*; *Stowers et al., 2017*). Although expected in light conditions, the fact that gain 1 is achieved in darkness shows that the network predicts the missing visual input from the velocity input, that is, the network path integrates accurately. Note that PI is impaired in our model for very small angular velocities (*Figure 2C*, flat purple line for $|v| < 30$ deg/s), similarly to previous hand-tuned theoretical models (*Turner-Evans et al., 2017*). This is a direct consequence of the fact that maintaining a stable activity bump and moving it across the network at very small angular velocities are competing goals. Crucially, it has been reported that such an impairment of PI for small angular velocities exists in flies (*Seelig and Jayaraman, 2015*). Note that if we increase the number of HD neurons from 60 (~50 were reported in the fly by *Turner-Evans et al., 2020*; *Xu, 2020*) to 120 or 240, this flat region is no longer observed (data not shown).

## The network is a quasi-continuous attractor

A continuous attractor network (CAN) should be able to maintain a localised bump of activity in virtually a continuum of locations around the ring of HD cells. To prove that the learned network approximates this property, we seek to reproduce in silico experimental findings in *Kim et al., 2017*. There it was shown that local optogenetic stimulation of HD cells in the ring can cause the activity bump to jump to a new position and persist in that location — supported by internal dynamics alone.

To reproduce the experiments by *Kim et al., 2017*, we simulate optogenetic stimulation of HD cells in our network as visual input of increased strength and extent (for details, see Materials and methods). We find that the strength and extent of the stimulation needs to be increased relative to that of the visual input; only in this case, a bump at some other location in the network can be suppressed, and a new bump emerges at the stimulated location. The stimuli are assumed to appear instantaneously at random locations, but we restrict our set of stimulation locations to the discrete

angles represented by the finite number of HD neurons. Furthermore, the velocity input is set to zero for the entire simulation, signaling lack of head movement.

*Figure 2D* shows network activity in response to several stimuli, when the stimulation location changes abruptly every 5 s. During stimulation (2 s long, red overbars), the bump is larger than normal due to the use of a stronger than usual visual-like input to mimic optogenetic stimulation. The way in which the network responds to a stimulation depends on how far away from the 'current' location it is stimulated: for shorter distances, the bump activity shifts to the new location, as evidenced by the transient dynamics at the edges of the bump resembling a decay from an initial to a new location (see *Figure 2D* at {5,15,20} s). However, for larger phase shifts $\Delta\theta$ the bump first emerges in the new location and subsequently disappears at the initial location, a mechanism akin to a 'jump' (*Figure 2D*, all other transitions). Similar effects have been observed in the experimental literature (*Seelig and Jayaraman, 2015*; *Kim et al., 2017*). The way the network responds to stimulation indicates that it operates in a CAN manner, and not as a winner-takes-all network where changes in bump location would always be instantaneous (*Carpenter and Grossberg, 1987*; *Itti et al., 1998*; *Wang, 2002*). That is to say, the network operates as expected from a quasi-continuous attractor. Furthermore, we find that the transition strategy in our model changes from predominantly smooth transitions to jumps at $\Delta\theta \approx 90\,\deg$, which matches experiments well (*Kim et al., 2017*).

Following a 2 s stimulation, the network activity has converged to the new cued location. After the stimulation has been turned off, the bump remains at the new location (within the angular resolution $\Delta\phi$ of the network), supported by internal network dynamics alone (*Figure 2D*). We confirmed in additional simulations that the bump does not drift away from the stimulated location for extended periods of time (3 min duration tested, only 3 s shown), and for all discrete locations in the HD network (only six locations shown). Therefore, we conclude that the HD network is a quasi-continuous attractor that can reliably sustain a heading representation over time in all HD locations. Note that for the network size used ($N^{HD} = 60$) we still obtain discrete attractors with separated basins of attraction; however it is expected that with increasing $N^{HD}$ adjacent attractors will merge when the intrinsic noise overcomes the barrier separating them. Indeed, we find that for $N^{HD} = N^{HR} = 120$ it is easier to diffuse to adjacent attractors in the presence of synaptic input noise; for the impact of noise, see *Appendix 1—figure 1C*. In reality, the bump may drift away due to asymmetries in the connectivity of the biological circuit as well as intrinsic noise (*Burak and Fiete, 2012*) see also Appendix 1. In flies, for instance, the bump can stay put only for several seconds (*Kim et al., 2017*).

## Learning results in synaptic connectivity that matches the one in the fly

To gain more insight into how the network achieves PI and attains CAN properties, we show how the synaptic weights of the network are tuned during a developmental period (*Figure 3*). *Figure 3A and B* shows the learned recurrent synaptic weights among the HD cells, $W^{rec}$, and the learned synaptic weights from HR to HD cells, $W^{HR}$, respectively. Circular symmetry is apparent in both matrices, a crucial property for a symmetric ring attractor. Therefore, we also plot the profiles of the learned weights as a function of receptive field difference in *Figure 3C*. Note that pixelized appearance in these plots is due to the fact that two adjacent HD neurons are tuned for the same HD, and develop identical synaptic strengths.

First, we discuss the properties of the learned weights. Local excitatory connections have developed along the main diagonal of $W^{rec}$, similar to what is observed in the CX (*Turner-Evans et al., 2020*). This local excitation can be readily seen in the weight profile of $W^{rec}$ in *Figure 3C*, and it is the substrate that allows the network to support stable activity bumps in virtually any location. In addition, we observe inhibition surrounding the local excitatory profile in both directions. This inhibition emerges despite the fact that we provide global inhibition to all HD cells ($I_{inh}^{HD}$ parameter, Materials and methods), in line with suggestions from previous work (*Kim et al., 2017*). Surrounding inhibition was a feature we observed consistently in learned networks of different sizes and for different global inhibition levels. Finally, the angular offset of the two negative sidelobes in the connectivity depends on the size and shape of the entrained HD bump (for details, see Appendix 5).

Furthermore, we find a consistent pattern of both L-HR and R-HR populations to excite the direction for which they are selective (*Figure 3C*), which is also similar to what is observed in the CX (*Turner-Evans et al., 2020*). Excitation in one direction is accompanied by inhibition in the reverse direction in the learned network. As a result of the symmetry in our learning paradigm, the connectivity profiles

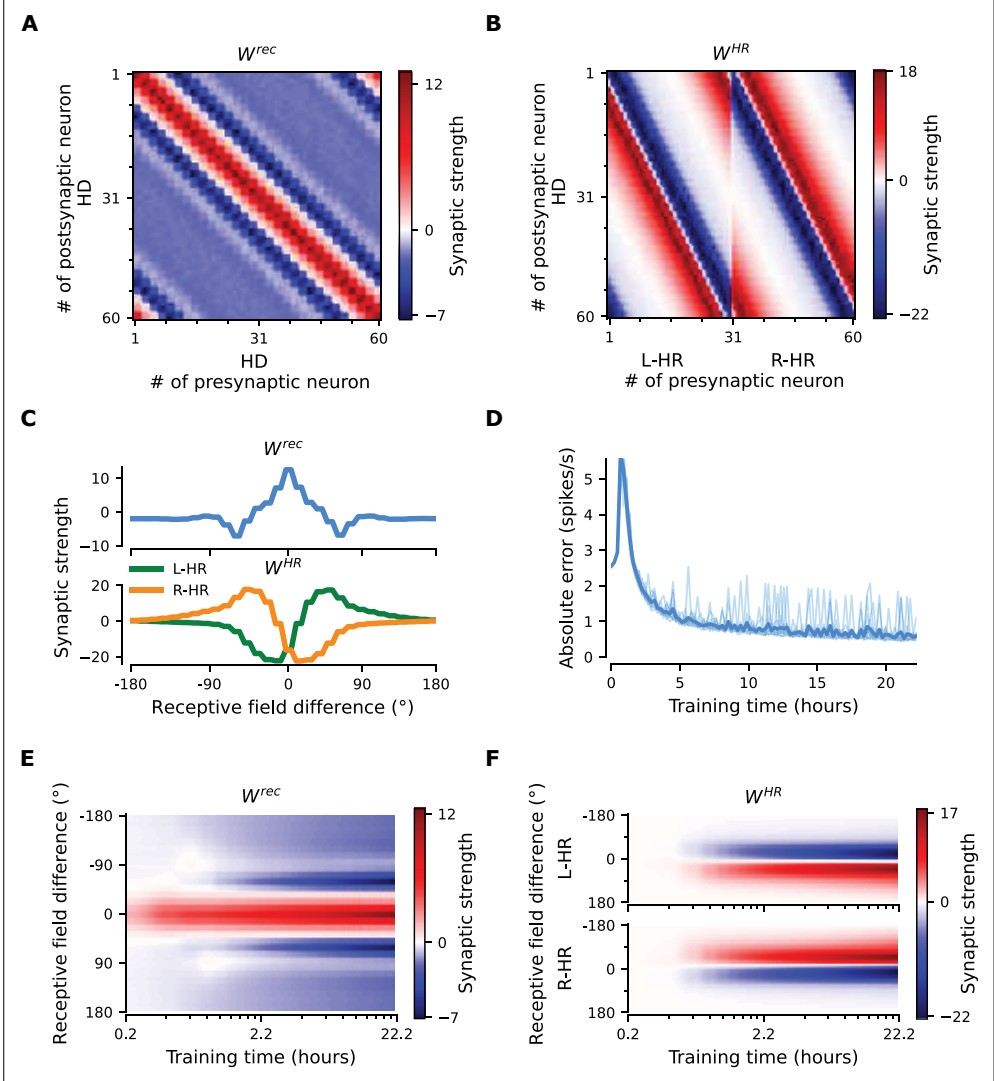

**Figure 3.** The network connectivity during and after learning. (**A**), (**B**) The learned weight matrices (color coded) of recurrent connections in the HD ring, $W^{rec}$, and of HR-to-HD connections, $W^{HR}$, respectively. Note the circular symmetry in both matrices. (**C**) Profiles of (**A**) and (**B**), averaged across presynaptic neurons. (**D**) Absolute learning error in the network (*Equation 19*) for 12 simulations (transparent lines) and average across simulations (opaque line). At time $t = 0$, we initialize all the plastic weights at random and train the network for $8 \times 10^4$ s (~22 hr). The mean learning error increases in the beginning while a bump in $W^{rec}$ is emerging, which is necessary to generate a pronounced bump in the network activity. For weak activity bumps, absolute errors are small because the overall network activity is low. After ~1 hr of training, the mean learning error decreases with increasing training time and converges to a small value. (**E**), (**F**) Time courses of development of the profiles of $W^{rec}$ and $W^{HR}$, respectively. Note the logarithmic time scale.

The online version of this article includes the following figure supplement(s) for figure 3:

**Figure supplement 1.** Removal of long-range excitatory projections impairs PI for high angular velocities.

**Figure supplement 2.** Details of learning.

**Figure supplement 3.** PI performance of a perturbed network.

---

of L-HR and R-HR cells are mirrored versions of each other, which is also clearly visible in *Figure 3C*. The inhibition of the reverse direction has a width comparable to the bump size and acts as a 'break' to prevent the bump from moving in this direction. The excitation in the selective direction, on the other hand, has a wider profile, which allows the network to path integrate for a wide range of angular velocities, that is for high angular velocities neurons further downstream can be 'primed' and activated

in rapid succession. Indeed, when we remove the wide projections from the excitatory connectivity, PI performance is impaired for the higher angular velocities exclusively (*Figure 3—figure supplement 1*). The even weight profile in $W^{rec}$ and the mirror symmetry for L-HR vs. R-HR profiles in $W^{HR}$, together with the circular symmetry of the weights throughout the ring, guarantee that there is no side bias (i.e. tendency of the bump to favor one direction of movement versus the other) during PI. Indeed, the PI error distribution in *Figure 2B* remains symmetric throughout the 60 s simulations.

Next, we focus our attention on the dynamics of learning. For training times larger than a few hours, the absolute learning error drops and settles to a low value, indicating that learning has converged after ~20 hr (or 4000 cycles, each cycle lasting $1/\eta$) of training time (*Figure 3D*). The non-zero value of the final error is only due to errors occurring at the edges of the bump (*Figure 3—figure supplement 2A*, top panel). An intuitive explanation of why these errors persist is that the velocity pathway is learning to predict the visual input; as a result, when the visual input is present, the velocity pathway creates errors that are consistent with PI velocity biases in darkness.

*Figure 3E and F* shows the weight development history for the entire simulation. The first structure that emerges during learning is the local excitatory recurrent connections in $W^{rec}$. For these early stages of learning, the initial connectivity is controlled by the autocorrelation of the visual input, which gets imprinted in the recurrent connections by means of Hebbian co-activation of adjacent HD neurons. As a result, the width of the local excitatory profile mirrors the width of the visual input. Once a clear bump is established in the HD ring, the HR connections are learned to support bump movement, and negative sidelobes in $W^{rec}$ emerge. To understand the shape of the learned connectivity profiles and the dynamics of their development, we study a reduced version of the full model, which follows learning in bump-centric coordinates (see Appendix 5). The reduced model produces a connectivity strikingly similar to the full model, and highlights the important role of non-linearities in the system.

So far, we have shown results in which our model far outperforms flies in terms of PI accuracy. To bridge this gap, we add noise to the weight connectivity in *Figure 3A and B* and obtain the connectivity matrices in *Figure 3—figure supplement 3A,B*, respectively. This perturbation of the weights could account for irregularities in the fly HD system owing to biological factors such as uneven synaptic densities. The resulting neural velocity gain curve in *Figure 3—figure supplement 3E* is impaired mainly for small angular velocities (*Figure 2C*). Interestingly, it now bears greater similarity to the one observed in flies, because the previously flat area for small angular velocities is wider (flat for $|v| < 60$ deg/s, cf. extended data fig. 7G,J in *Seelig and Jayaraman, 2015*). This happens because the noisy connectivity is less effective in initiating bump movement. Finally, the PI errors in the network with noisy connectivity grow much faster and display a strong side bias (*Figure 3—figure supplement 3D*, *Figure 2B*). The latter can be attributed to the fact that the noise in the connectivity generates local minima that are easier to transverse from one direction vs. the other. Side bias can also emerge if the learning rate $\eta$ in *Equation 16* is increased, effectively forcing learning to converge faster to a local minimum, which results in slight deviations from circularly symmetric connectivity (data not shown). It is therefore expected that different animals will display different degrees and directions of side bias during PI, owing either to fast learning or asymmetries in the underlying neurobiology. Since the exact behavior of the network with noise in the connectivity depends on the specific realization, we also generate multiple such networks and estimate the diffusion coefficient during path integration, which quantifies how fast the width of the PI error distribution in *Figure 3—figure supplement 3D* increases. We find the grand average to be $82.3 \pm 15.7$ $\text{deg}^2/\text{s}$, which is considerably larger (Student's t-test, 95% conf. intervals for a total of 12 networks) than the diffusion coefficient for networks without a perturbation in the weights ($24.5$ $\text{deg}^2/\text{s}$ in *Figure 2B*). Finally, in Appendix 1 we also incorporate random Gaussian noise to all inputs, which can account for noisy percepts or stochasticity of spiking, and show that learning is not disrupted even for high noise levels.

## Fast adaptation of neural velocity gain

Having shown how PI and CAN properties are learned in our model, we now turn our attention to the flexibility that our learning setup affords. Motivated by augmented-reality experiments in rodents where the relative gain of visual and self-motion inputs is manipulated (*Jayakumar et al., 2019*), we test whether our network can rewire to learn an arbitrary gain between the two. In other words, we attempt to learn an arbitrary gain $g$ between the idiothetic angular velocity $v$ sensed by the HR cells

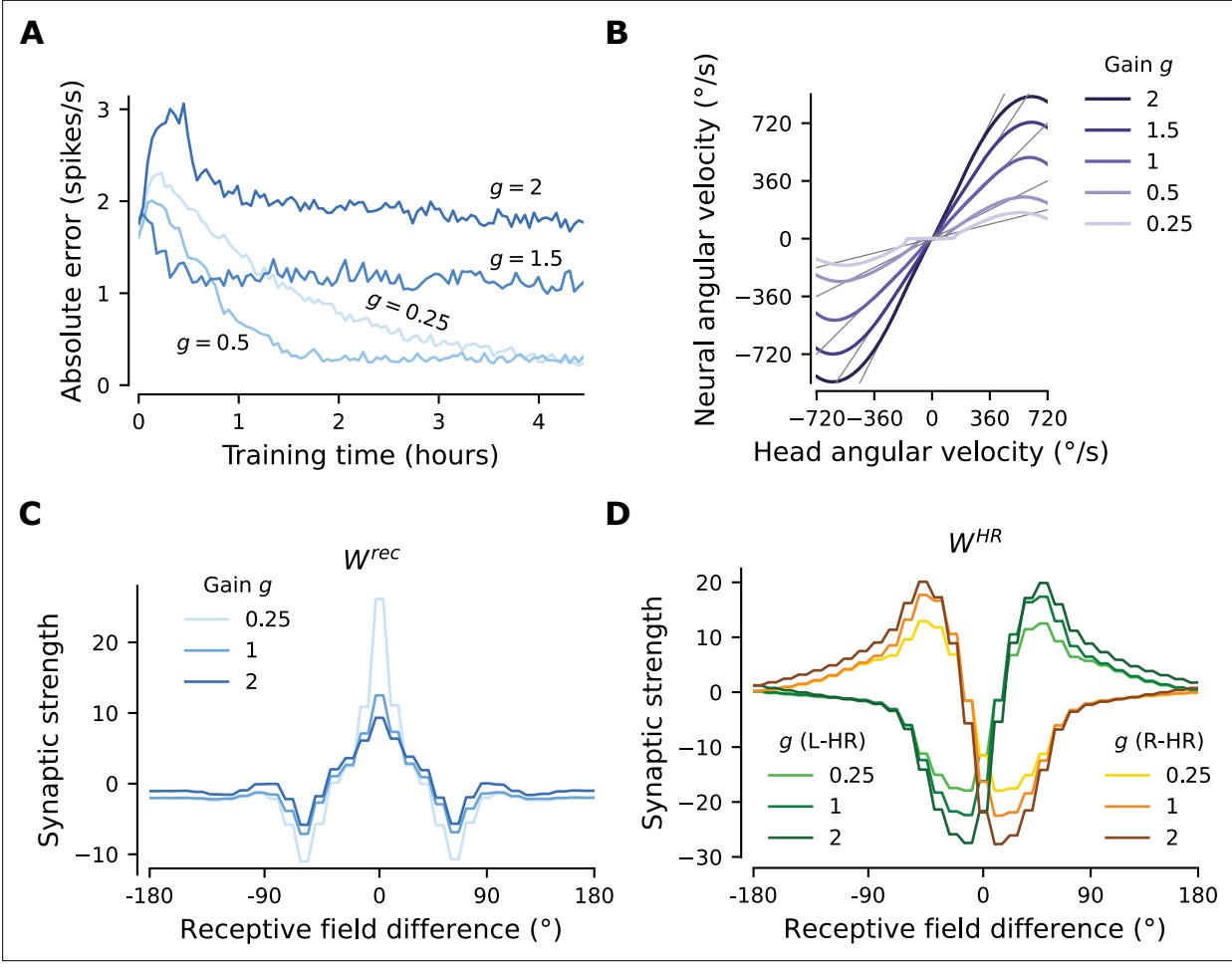

**Figure 4.** The network adapts rapidly to new gains. Starting from the converged network in *Figure 3*, we change the gain $g$ between visual and self-motion inputs, akin to experiments conducted in VR in flies and rodents (*Seelig and Jayaraman, 2015*; *Jayakumar et al., 2019*). (**A**) The mean learning error averaged across 12 simulations for each gain. After an initial increase due to the change of gain, the errors decrease rapidly and settle to a lower value. The steady-state values depend on the gain due to the by-design impairment of high angular velocities, which affects high gains preferentially. Crucially, adaptation to a new gain is much faster than learning the HD system from scratch (*Figure 3D*). (**B**) Velocity gain curves for different gains. The network has remapped to learn accurate PI with different gains for the entire dynamic range of head angular velocity inputs (approx. [-500, 500] deg/s). (**C**), (**D**) Final profiles of $W^{rec}$ and $W^{HR}$, respectively, for different gains.

The online version of this article includes the following figure supplement(s) for figure 4:

**Figure supplement 1.** Limits of PI gain adaptation.

and the neural velocity $g \cdot v$ dictated by the allothetic visual input. This simulates the conditions in an augmented reality environment, where the speed at which the world around the animal rotates is determined by the experimenter, but the proprioceptive sense of head angular velocity remains the same.

Starting with the learned network shown in *Figure 3*, which displayed gain $g = 1$, we suddenly switch to a different gain, that is we learn weights for $g \in \{0.25, 0.5, 1.5, 2\}$. In all cases, we observe that the network readily rewires to achieve the new gain. The mean learning error after the gain switch is initially high, but reaches a lower, constant level after at most 3 hr of training (*Figure 4A*). We note that convergence is much faster compared to the time it takes for the gain-1 network to emerge from scratch (compare to *Figure 3D*), especially for the smaller gain changes. Importantly, *Figure 4B* shows that PI performance in the resulting networks is excellent for the new gains, with some degradation only for very low and very high angular velocities. There are two reasons why high angular velocities are not learned that well: limited training of these velocities, and saturation of HR cell activity. Both reasons are by design and do not reflect a fundamental limit of the network.

In Appendix 2, we show that without the aforementioned limitations the network learns to path-integrate up to an angular velocity limit set by synaptic delays and that the bump width sets a trade-off between location and velocity-integration accuracy in the HD system.

*Figure 4C and D* compare the weight profiles of the circularly symmetric matrices $W^{rec}$ and $W^{HR}$ resulting from the initial gain $g = 1$, with the weight profiles resulting from adaptation to the most extreme gains shown in *Figure 4*, that is $g \in \{0.25, 2\}$. An increase in gain slightly suppresses the recurrent connections and slightly amplifies the HR-to-HD connections, while a decrease in gain substantially amplifies the recurrent connections and slightly suppresses the HR-to-HD connections. The latter explains why the flat region for small angular velocities in *Figure 4B* has been extended for $g \in \{0.25, 0.5\}$: it is now harder for small angular velocities to overcome the attractor formed by stronger recurrent weights and move the bump.

Finally, we address the limits of the ability of the network to rewire to new gains (*Figure 4—figure supplement 1*). We find that after rewiring the performance is excellent for gains between 0.25 and 4.5. The network can even reverse its gain to $g = -1$, that is, when allothetic and idiothetic inputs are signaling movement in opposite directions. However, for larger gain changes, learning takes longer.

## Discussion

The ability of animals to navigate in the absence of external cues is crucial for their survival. Head direction, place, and grid cells provide internal representations of space (*Ranck, 1984*; *Moser et al., 2008*) that can persist in darkness and possibly support path integration (PI) (*Mizumori and Williams, 1993*; *Quirk et al., 1990*; *Hafting et al., 2005*). Extensive theoretical work has focused on how the spatial navigation system might rely on continuous attractor networks (CANs) to maintain and update a neural representation of the animal's current location. Special attention was devoted to models representing orientation, with the ring attractor network being one of the most famous of these models (*Amari, 1977*; *Ben-Yishai et al., 1995*; *Skaggs et al., 1995*; *Seung, 1996*). So far, modelling of the HD system has been relying on hand-tuned synaptic connectivity (*Zhang, 1996*; *Xie et al., 2002*; *Turner-Evans et al., 2017*; *Page et al., 2019*) without reference to its origin; or has been relying on synaptic plasticity rules that either did not achieve gain-1 PI (*Stringer et al., 2002*) or were not biologically plausible (*Hahnloser, 2003*).

### Summary of findings

Inspired by the recent discovery of a ring attractor network for HD in *Drosophila* (*Seelig and Jayaraman, 2015*), we show how a biologically plausible learning rule leads to the emergence of a circuit that achieves gain-1 PI in darkness. The learned network features striking similarities in terms of connectivity to the one experimentally observed in the fly (*Turner-Evans et al., 2020*), and reproduces experiments on CAN dynamics (*Kim et al., 2017*) and gain changes between external and self-motion cues in rodents (*Jayakumar et al., 2019*). Furthermore, an impairment of PI for small angular velocities is observed in the mature network, which is a feature that has been reported in experiments (*Seelig and Jayaraman, 2015*). Finally, the proposed learning rule can serve to compensate deviations from circular symmetry in the synaptic weight profiles; such deviations are expected in biological systems and — if not compensated — could lead to large PI errors.

The mature circuit displays two properties characteristic of CANs: (1) it can support and actively maintain a local bump of activity at a virtual continuum of locations, and (2) it can move the bump across the network by integrating self-motion cues. Note that we did not explicitly train the network to achieve these CAN properties, but they rather emerged in a self-organized manner.

To achieve gain-1 PI performance, our network must attribute learning errors to the appropriate weights. The learning rule we adopt in *Equation 1* is a 'delta-like' rule, with a learning error that gates learning in the network, and a Hebbian component that comes in the form of the postsynaptic potential and assigns credit to synapses that are active when errors are large. The learning rule leads to the emergence of both symmetric local connectivity between HD cells (which is required for bump maintenance and stability), and asymmetric connectivity from HR to HD cells (which is required for bump movement in darkness). The first happens because adjacent neurons are co-active due to correlated visual input; the second because only one HR population is predominantly active during rotation: the population that corresponds to the current rotation direction. Crucial to the understanding of the

learning dynamics of the model was the development of a reduced model, which follows learning in bump-centric coordinates and is analytically tractable (see Appendix 5). The reduced model can be extended to higher dimensional manifolds (*Gardner et al., 2022*), and therefore it offers a general framework to study how activity-dependent synaptic plasticity shapes CANs.

## Relation to experimental literature

Our work comes at a time at which the fly HD system receives a lot of attention (*Seelig and Jayaraman, 2015*; *Turner-Evans et al., 2020*; *Kim et al., 2017*; *Kim et al., 2019*; *Fisher et al., 2019*), and suggests a mechanism of how this circuit could self-organize during development. Synaptic plasticity has been shown to be important in this circuit for anchoring the visual input to the HD neurons when the animal is exposed to a new environment (*Kim et al., 2019*; *Fisher et al., 2019*). This has also been demonstrated in models of the mammalian HD system (*Skaggs et al., 1995*; *Zhang, 1996*; *Song and Wang, 2005*). Here, we assume that an initial anchoring of the topographic visual input to the HD neurons with arbitrary offset with respect to external landmarks already exists prior to the development of the PI circuit; such an anchoring could even be prewired. In our model, it is sufficient that the visual input tuning is local and topographically arranged. Once the PI circuit has developed, visual connections could be anchored to different environments, as shown by *Kim et al., 2019* and *Fisher et al., 2019*. Alternatively, the HD system itself could come prewired with an initial gross connectivity, sufficient to anchor the visual input; in this case, our learning rule would enable fine tuning of this connectivity for gain-1 PI. In either case, for the sake of simplicity and without loss of generality, we study the development of the path-integrating circuit while the animal moves in the same environment, and keep the visual input tuning fixed. Therefore, the present work addresses the important question of how the PI circuit itself could be formed, and it is complementary to the problem of how allothetic inputs to the PI circuit are wired (*Fisher et al., 2019*; *Kim et al., 2019*). The interplay of the two forms of plasticity during development would be of particular future interest.

A requirement for the learning rule we use is that information about the firing rate of HD neurons is available at the axon-distal compartment. There is no evidence for active backpropagation of APs in E-PG neurons in the fly, but passive backpropagation would suffice in this setting. In fact, passive spread of activity has been shown to attenuate weakly in central fly neurons (*Gouwens and Wilson, 2009*). In HD neurons, the axon-proximal and axon-distal compartments belong to the same dendritic tuft (*Figure 1E*), and since we assume that the axon initial segment is close to the axon-proximal compartment, the generated AP would need to propagate only a short distance compared to the effective electrotonic length. This means that APs would not be attenuated much on their way from the axon initial segment to the axon-distal compartment, and thus would maintain some of their high-frequency component, which could be used at synapses to differentiate them from slower post-synaptic potentials.

In *Figure 4*, we show that our network can adapt to altered gains much faster than the time required to learn the network from scratch. Our simulations are akin to experiments where rodents are placed in a VR environment and the relative gain between visual and proprioceptive signals is altered by the experimenter (*Jayakumar et al., 2019*). In this scenario, *Jayakumar et al., 2019* found that the PI gain of place cells can be recalibrated rapidly. In contrast, *Seelig and Jayaraman, 2015* found that PI gain in darkness is not significantly affected when flies are exposed to different gains in light conditions. We note, however, that *Seelig and Jayaraman, 2015* tested mature animals (8–11 days old), whereas plasticity in the main HD network is presumably stronger in younger animals. Also note that the manipulation we use to address adaptation of PI to different gains differs from the one in *Kim et al., 2019* who used optogenetic stimulation of the HD network combined with rotation of the visual scene to trigger a remapping of the visual input to the HD cells in a Hebbian manner. The findings in *Jayakumar et al., 2019* can only be reconciled by plasticity in the PI circuit, and not in the sensory inputs to the circuit.

In order to address the core mechanisms that underlie the emergence of a path integrating network, we use a model that is a simplified version of the biological circuit. For example, we did not model inhibitory neurons explicitly and omitted some of the recurrent connectivity in the circuit, whose functional role is uncertain (*Turner-Evans et al., 2020*). We also choose to separate PI from other complex processes that occur in the CX (*Raccuglia et al., 2019*). Finally, we do not force the network to obey Dale's law and do not model spiking explicitly.

Nevertheless, after learning, we obtain a network connectivity that is strikingly similar to the one of the fly HD system. Indeed, the mature model exhibits local excitatory connectivity in the HD neurons (*Figure 3A and C*), which in the fly is mediated by the excitatory loop from E-PG to P-EG to P-EN2 and back to E-PG (*Turner-Evans et al., 2020*), a feature that hand-tuned models of the fly HD system did not include (*Turner-Evans et al., 2017*). Furthermore, the HR neurons have excitatory projections towards the directions they are selective for (*Figure 3B and C*), similar to P-EN1 neurons in the fly. Interestingly, these key features that we uncover from learning have been utilized in other hand-tuned models of the system (*Turner-Evans et al., 2017*; *Kim et al., 2017*; *Kim et al., 2019*). Future work could endeavor to come closer to the architecture of the fly HD system and benefit from the incorporation of more neuron types and the richness of recurrent connectivity that has been discovered in the fly (*Turner-Evans et al., 2020*).

Compared to the fly, our network achieved better PI performance. As a simple way to match the performances, we added noise to the learned connectivity in the model; however, this is not an explanation why the fly performs worse. Indeed, there could be multiple reasons why PI performance is worse in the biological circuit. For instance, a confounder that would affect performance but not necessarily learning could be the presence of inputs that are unrelated to path integration, for example, inputs related to circadian cycles and sleep (*Raccuglia et al., 2019*). In the presence of such confounders, a precise tuning of the weights might be crucial in order to reach the performance of the fly. In other words, only if the model outperforms the biological circuit in a simplified setting, it has a chance to perform as well in a realistic setting, with all the additional complexities the latter comes with.

## Relation to theoretical literature

A common problem with CANs is that they require fine tuning: even a slight deviation from the optimal synaptic weight tuning leads to catastrophic drifting (*Goldman et al., 2009*). A way around this problem is to sacrifice the continuity of the attractor states in favor of a discrete number of stable states that are much more robust to noise or weight perturbations (*Kilpatrick et al., 2013*). In our network, the small number of HD neurons enables a coarse-grained representation of heading; the network is a CAN only in a quasi-continuous manner, and the number of discrete attractors corresponds to the number of HD neurons. This makes it harder to transition to adjacent attractors, since a 'barrier' has to be overcome in the quasi-continuous case (*Kilpatrick et al., 2013*). The somewhat counter-intuitive conclusion follows that a CAN with more neurons and, as a result, finer angular resolution, will not be as potent in maintaining activity, and diffusion to nearby attractors will be easier since the barrier will be lower. Indeed, we found that doubling the number of neurons produces a CAN that is less robust to noise. Overall, the quasi-continuous and coarse nature of the attractor shields the internal representation of heading against the ever-present biological noise, which would otherwise lead to diffusion of the bump with time. The fact that the network can still path-integrate accurately with this coarse-grained representation of heading is remarkable.

Seminal theoretical work on ring attractors has proven that in order to achieve gain-1 PI, the asymmetric component of the network connectivity (corresponding here to $W^{HR}$) needs to be proportional to the derivative of the symmetric component (corresponding to $W^{rec}$) (*Zhang, 1996*). However, this result rests on the assumption that asymmetric and symmetric weight profiles are mediated by the same neuronal population, as in the double-ring architecture proposed by *Xie et al., 2002* and *Hahnloser, 2003*, but does not readily apply to the architecture of the fly HD system where HD and HR cells are separate. In our learned network, we find that the HR weight profile is not proportional to the derivative of the recurrent weight profile, therefore this requirement is not necessary for gain-1 PI in our setting. Note that our learning setup can also learn gain-1 PI for a double-ring architecture, which additionally obeys Dale's law (*Vafidis, 2019*). Finally, we emphasize that circular symmetry is not a necessary condition for a ring attractor (*Darshan and Rivkind, 2021*). Rather, symmetry in our model results from the symmetry in the architecture, the symmetrically prewired weights, and the symmetric stimulus space. If any of those were to be relaxed, the resulting network would not be circular symmetric; then, the reduced model analysis that we perform in Appendix 5 would also not be feasible, because local asymmetries in the setup would result in non-local deviations from circular symmetry of the learned weights, which was our main assumption there. Nevertheless, we

demonstrated that the full model can handle such asymmetries in the setup and learn accurate PI (see Appendix 3).

Our learning setup, inspired by *Urbanczik and Senn, 2014*, is similar to the one in *Guerguiev et al., 2017* in the sense that both involve compartmentalized neurons that receive 'target' signals in a distinct compartment. It differs, however, in the algorithm and learning rule used. *Guerguiev et al., 2017* use local gradient descent during a 'target' phase, which is separate from a forward propagation phase, akin to forward/backward propagation stages in conventional deep learning. In contrast, we use a modified Hebbian rule, and in our model 'forward' computation and learning happen at the same time; time multiplexing, whose origin in the brain is unclear, is not required. Our setting would be more akin to the one in *Guerguiev et al., 2017* if an episode of PI in darkness would be required before an episode of learning in light conditions, which does not seem in line with the way animals naturally learn.

Previous theoretical work showed that head direction cells, head rotation cells, and grid cells emerge in neural networks trained for PI (*Banino et al., 2018*; *Cueva and Wei, 2018*). These networks were trained with backpropagation, therefore achieving gain-1 PI was not their primary focus; rather, this work elegantly demonstrated that the aforementioned cell types are efficient representations for spatial navigation that could be learned from experience.

## Testable predictions

We devote this section to discussing predictions of our model, and we suggest future experiments in flies and, potentially, other animal models. An obvious prediction of our model is that synaptic plasticity is critical for the development of the PI network for heading, and the lack of a supervisory allothetic sensory input (e.g. visual) during development should disrupt the formation of the PI system. Previous experimental work showed that head direction cells in rat pups displayed mature properties already in their first exploration of the environment outside their nest (*Langston et al., 2010*), which may seem to contradict our assumption that the PI circuit wires during development; however, directional selectivity of HD cells in the absence of allothetic inputs and PI performance were not tested in this study. In addition, it has been shown that visually impaired flies were not able to learn to accurately estimate the size of their body. This type of learning also requires visual inputs and, upon consolidation, remains stable (*Krause et al., 2019*).

We also predict that HD neurons have a compartmental structure where idiothetic inputs are separated from allothetic sensory inputs, which initiate action potentials more readily due to being electrotonically closer to the axon initial segment. While we already demonstrate the separation of allothetic and idiothetic inputs to E-PG neurons in the fly EB (*Figure 1E*, *Figure 1—figure supplement 1*), our prediction can only be tested with electrophysiological experiments. Another model prediction that can be tested only with electrophysiology is that APs backpropagate from the axon-proximal compartment (at least passively but with little attenuation) to the axon-distal compartment. Then spikes could be separated from postsynaptic potentials locally at the synapse by cellular mechanisms sensitive to the spectral density of the voltage.

Finally, similarly to place cell studies in rodents (*Jayakumar et al., 2019*), we predict that during development the PI system can adapt to experimenter-defined gain manipulations, and that it can do so faster than the time required for the system to develop from scratch. Therefore, a suggestion from this study would be to repeat in young flies the adaptation experiments by *Seelig and Jayaraman, 2015*.

## Outlook

The present study adds to the growing literature of potential computational abilities of compartmentalized neurons (*Poirazi et al., 2003*; *Gidon et al., 2020*; *Payeur et al., 2021*). The associative HD neuron used in this study is a coincidence detector, which serves to associate external and internal inputs arriving at different compartments of the cell. Coupled with memory-specific gating of internally generated inputs, coincidence detection has been suggested to be the fundamental mechanism that allows the mammalian cortex to form and update internal knowledge about external contingencies (*Doron et al., 2020*; *Shin et al., 2021*). This structured form of learning does not require engineered 'hints' during training, and it might be the reason why neural circuits evolved to be so efficient at reasoning about the world, with the mammalian cortex being the pinnacle of this achievement.

Here, we demonstrate that learning at the cellular level can predict external inputs (visual information) by associating firing activity with internally generated signals (velocity inputs) during training. This effect is due to the anti-Hebbian component of the learning rule in *Equation 12*, where the product of postsynaptic axon-distal and presynaptic activity comes with a negative sign. Specifically, it has previously been demonstrated that anti-Hebbian synaptic plasticity can stabilize persistent activity (*Xie and Seung, 2000*) and perform predictive coding (*Bell et al., 1997*; *Hahnloser, 2003*). At the population level, this provides a powerful mechanism to internally produce activity patterns that are identical to the ones induced from an external stimulus. This mechanism can serve as a way to anticipate external events or, as in our case, as a way of 'filling in' missing information in the absence of external inputs.

Local, Hebb-like learning rules are considered a weak form of learning, due to their inability to utilize error information in a sophisticated manner. Despite that, we show that local associative learning can be particularly successful in learning appropriate fine-tuned synaptic connectivity, when operating within a cell structured for coincidence detection. Therefore, in learning and reasoning about the environment, our study highlights the importance of inductive biases with developmental origin (e.g. allothetic and idiothetic inputs arrive in different compartments of associative neurons) (*Lake et al., 2017*).

In conclusion, the present work addresses the age-old question of how to develop a CAN that performs accurate, gain-1 PI in the absence of external sensory cues. We show that this feat can be achieved in a network model of the HD system by means of a biologically plausible learning rule at the cellular level. Even though our network architecture is tailored to the one of the fly CX, the learning setup where idiothetic and allothetic cues are associated at the cellular level is general and can be applied to other PI circuits. Of particular interest is the rodent HD system: despite the lack of evidence for a topographically organized recurrent HD network in rodents, a one-dimensional HD manifold has been extracted in an unsupervised way (*Chaudhuri et al., 2019*). Therefore, our work lays the path to study the development of ring-like neural manifolds in mammals. Finally, it has recently been shown that grid cells in mammals form a continuous attractor manifold with toroidal topology (*Gardner et al., 2022*). It would be interesting to see if a similar mechanism underlies the emergence of PI in place and grid cells. Our model can be extended to higher dimensional CAN manifolds and provides a framework to interrogate this assumption.

## Materials and methods

In what follows, we describe our computational model for learning a ring attractor network that accomplishes accurate angular PI. The model described here focuses on the HD system of the fly; however, the proposed computational setup is general and could be applied to other systems. Unless otherwise stated, the simulation parameter values are the ones summarized in *Table 1*. Simulation results for a given choice of parameters are very consistent across runs, hence most figures are generated from a single simulation run, unless otherwise stated.

.

### Network architecture

We model a recurrent neural network comprising $N^{HD} = 60$ head-direction (HD) and $N^{HR} = 60$ head-rotation (HR) cells, which are close to the number of E-PG and P-EN1 cells in the fly central complex (CX), respectively (*Turner-Evans et al., 2020*; *Xu, 2020*). A scaled-down version of the network for $N^{HR} = N^{HD} = 12$ is shown in *Figure 1A*. The average spiking activity of HD and HR cells is modelled by firing-rate neurons. HD cells are organized in a ring and receive visual input, which encodes the angular position of the animal's head with respect to external landmarks. We use a discrete representation of angles and we model two HD cells for each head direction, as observed in the biological system (*Turner-Evans et al., 2017*). Therefore the network can represent head direction with an angular resolution $\Delta\phi = 12$ deg.

Motivated by the anatomy of the fly CX (*Green et al., 2017*; *Turner-Evans et al., 2020*), HR cells are divided in two populations (*Figure 1A*): a 'leftward' (L-HR) population (with increased velocity input when the head turns leftwards) and a 'rightward' (R-HR) population (with increased velocity input when the head turns rightwards). After learning, these two HR populations are responsible to move the HD bump in the anticlockwise and clockwise directions, respectively.

**Table 1.** Parameter values.

| Parameter | Value | Unit | Explanation |
|---|---|---|---|
| $N^{HD}$ | 60 | | Number of head direction (HD) neurons |
| $N^{HR}$ | 60 | | Number of head rotation (HR) neurons |
| $\Delta\phi$ | 12 | deg | Angular resolution of network |
| $\tau_s$ | 65 | ms | Synaptic time constant |
| $I_{inh}^{HD}$ | -1 | | Global inhibition to HD neurons |
| $\tau_l$ | 10 | ms | Leak time constant of axon-distal compartment of HD neurons |
| $C$ | 1 | ms | Capacitance of axon-proximal compartment of HD neurons |
| $g_L$ | 1 | | Leak conductance of axon-proximal compartment of HD neurons |
| $g_D$ | 2 | | Conductance from axon-distal to axon-proximal compartment |
| $I_{exc}^{HD}$ | 4 | | Excitatory input to axon-proximal compartment in light conditions |
| $\sigma_n$ | 0 | | Synaptic input noise level |
| $M$ | 4 | | Visual input amplitude |
| $M_{stim}$ | 16 | | Optogenetic stimulation amplitude |
| $\sigma$ | 0.15 | | Visual receptive field width |
| $\sigma_{stim}$ | 0.25 | | Optogenetic stimulation width |
| $I_o^{vis}$ | -5 | | Visual input baseline |
| $f_{max}$ | 150 | spikes/s | Maximum firing rate |
| $\beta$ | 2.5 | | Steepness of activation function |
| $x_{1/2}$ | 1 | | Input level for 50% of the maximum firing rate |
| $I_{inh}^{HR}$ | -1.5 | | Global inhibition to HR neurons |
| $k$ | 1/360 | s/deg | Constant ratio of velocity input and head angular velocity |
| $A_{active}$ | 2 | | Input range for which $f$ has not saturated |
| $w^{HD}$ | $13.\overline{3}$ | ms | Constant weight from HD to HR neurons |
| $\tau_\delta$ | 100 | ms | Plasticity time constant |
| $\Delta t$ | 0.5 | ms | Euler integration step size |
| $\tau_v$ | 0.5 | s | Time constant of velocity decay |
| $\sigma_v$ | 450 | deg/$\sqrt{s}$ | Standard deviation of angular velocity noise |
| $\eta$ | 0.05 | 1 /s | Learning rate |

Parameter values, in the order they appear in the Methods section. These values apply to all simulations, unless otherwise stated. Note that voltages, currents, and conductances are assumed unitless in the text; therefore capacitances have the same units as time constants.

The recurrent connections among HD cells and the connections from HR to HD cells are assumed to be plastic. On the contrary, connections from HD to HR cells are assumed fixed and determined as follows: for every head direction, one HD neuron projects to a cell in the L-HR population, and the other to a cell in the R-HR population. Because HD cells project to HR cells in a 1-to-1 manner, each HR neuron is simultaneously tuned to a particular head direction and a particular head rotation direction. The synaptic strength of the HD-to-HR projections is the same for all projections (these restrictions on the HD-to-HR connections are relaxed in Appendix 3). Finally, HR cells do not form recurrent connections.

## Neuronal model

We assume that each HD neuron is a rate-based associative neuron (*Figure 1D*), that is, a two-compartmental neuron comprising an axon-proximal and an axon-distal dendritic compartment (*Urbanczik and Senn, 2014*; *Brea et al., 2016*). The two compartments model the dendrites of that neuron that are closer to or further away from the axon initial segment. Note that here the axon-proximal compartment replaces the somatic compartment in the original model by *Urbanczik and Senn, 2014*. This is because the somata of fly neurons are typically electrotonically segregated from the rest of the cell and they are assumed to contribute little to computation (*Gouwens and Wilson, 2009*; *Tuthill, 2009*). We also note that to fully capture the input/output transformations that HD neurons in the fly perform, more compartments than two might be needed (*Xu, 2020*). Finally, only HD cells are associative neurons, whereas HR cells are simple rate-based point neurons.

HD cells receive an input current $\boldsymbol{I}^d$ to the axon-distal dendrites, which obeys

$$\tau_s \frac{d\boldsymbol{I}^d}{dt} = -\boldsymbol{I}^d + W^{rec}\boldsymbol{r}^{HD} + W^{HR}\boldsymbol{r}^{HR} + I_{inh}^{HD} + \sigma_n \boldsymbol{n}^d \tag{2}$$

where $\boldsymbol{I}^d$ is a vector of length $N^{HD}$ with each entry corresponding to one HD cell. In *Equation 2*, $\tau_s$ is the synaptic time constant, $W^{rec}$ is a $N^{HD} \times N^{HD}$ matrix of the recurrent synaptic weights among HD cells, $W^{HR}$ is a $N^{HD} \times N^{HR}$ matrix of the synaptic weights from HR to HD cells, $\boldsymbol{r}^{HR}$ and $\boldsymbol{r}^{HD}$ are vectors of the firing rates of HR and HD cells respectively, $I_{inh}^{HD}$ is a constant inhibitory input common to all HD cells, and $\boldsymbol{n}^d$ is a random noise input to the axon-distal compartment. $\boldsymbol{n}^d$ is drawn IID from $N(0, 1)$, and its variance is scaled by $\sigma_n^2$. Note that in the main text we set $\sigma_n$ to zero, but we explore different values for this parameter in Appendix 1. The constant current $I_{inh}^{HD}$ is in line with a global-inhibition model with local recurrent connectivity, as opposed to having long-range inhibitory recurrent connectivity (*Kim et al., 2017*). The inhibitory current $I_{inh}^{HD}$ suppresses HD bumps in general; however the exact strength of this inhibition is not important in our model.

Since several electrophysiological parameters of the fly neurons modeled here are unknown, we use dimensionless conductance values. Therefore, in *Equation 2*, which describes the dynamics of the axon-distal input of HD cells, currents (e.g. $\boldsymbol{I}^d$, $I_{inh}^{HD}$, and $\boldsymbol{n}^d$) are dimensionless. Membrane voltages are also chosen to be dimensionless, and because we measure firing rates in units of 1 /s, all synaptic weights (e.g. $W^{rec}$ and $W^{HR}$) then have, strictly speaking, the unit 'seconds' (s), even though we mostly suppress this unit in the text. Importantly, all time constants (e.g. $\tau_s$), which define the time scale of dynamics, are measured in units of time (in seconds).

Our model incorporates several time scales, whose interplay is not obvious. To facilitate understanding, we summarize the parameters that define the time scales in *Appendix 4—table 1*, and discuss their relation in Appendix 4.

The axon-distal voltage $\boldsymbol{V}^d$ of HD cells is a low-pass filtered version of the input current $\boldsymbol{I}^d$, that is,

$$\tau_l \frac{d\boldsymbol{V}^d}{dt} = -\boldsymbol{V}^d + \boldsymbol{I}^d \tag{3}$$

where $\tau_l$ is the leak time constant of the axon-distal compartment. The voltage $\boldsymbol{V}^d$ and the current $\boldsymbol{I}^d$ have the same unit (both dimensionless), which means that the leak resistance of the axon-distal compartment is also dimensionless, and we assume that it is unity for simplicity. We choose values of $\tau_l$ and $\tau_s$ (for specific values, see *Table 1*) so that their sum matches the phenomenological time constant of HD neurons (E-PG in the fly), while $\tau_s$ equals to the phenomenological time constant of HR neurons (P-EN1 in the fly, *Turner-Evans et al., 2017*). Note that $\boldsymbol{V}^d$ is the low-frequency component of the axon-distal voltage originating from postsynaptic potentials, that is excluding occasional high-frequency contributions from backpropagating action potentials.

The axon-proximal voltage $\boldsymbol{V}^a$ of HD cells is then given by

$$C\frac{d\boldsymbol{V}^a}{dt} = -g_L \boldsymbol{V}^a - g_D(\boldsymbol{V}^a - \boldsymbol{V}^d) + \boldsymbol{I}^{vis} + I_{exc}^{HD} + \sigma_n \boldsymbol{n}^a \tag{4}$$

where $C$ is the capacitance of the membrane of the axon-proximal compartment, $g_L$ is the leak conductance, $g_D$ is the conductance of the coupling from axon-distal to axon-proximal dendrites, $\boldsymbol{I}^{vis}$ is a vector of visual input currents to the axon-proximal compartment of HD cells, $I_{exc}^{HD}$ is an excitatory input to the axon-proximal compartment, and $\boldsymbol{n}^a$ is a random noise vector injected to the axon-proximal compartment, drawn IID from $N(0, 1)$. The excitatory current $I_{exc}^{HD}$ is assumed to be present

only in light conditions. The values of $C$, $g_L$, and $g_D$ in the fly HD (E-PG) neurons are unknown, thus we keep these parameters unitless, and set their values to the ones in *Urbanczik and Senn, 2014*. Note that since conductances are dimensionless here, $C$ is effectively a time constant.

Following *Hahnloser, 2003*, the visual input to the i-th HD cell is a localized bump of activity at angular location $\theta_i$:

$$I_i^{vis}(t) = M \exp\left(-\tfrac{1}{2\sigma^2} \sin^2\left(\tfrac{\theta_i+\theta_0(t)}{2}\right)\right) + I_o^{vis} \tag{5}$$

where $M$ scales the bump's amplitude, $\sigma$ controls the width of the bump, $\theta_i$ is the preferred orientation of the i-th HD neuron, $\theta_0(t)$ is the position of a visual landmark at time $t$ in head-centered coordinates, and $I_o^{vis} < 0$ is a constant inhibitory current that acts as the baseline for the visual input. We choose $M$ so that the visual input can induce a weak bump in the network at the beginning of learning, and we choose $\sigma$ so that the resulting bump after learning is ~60 deg wide. Note that the bump in the mature network has a square shape (*Figure 3—figure supplement 2B*); therefore we elect to make it slightly narrower than the average full width at half maximum of the experimentally observed bump (~80 deg; *Seelig and Jayaraman, 2015*; *Kim et al., 2017*; *Turner-Evans et al., 2017*). In addition, the current $I_o^{vis}$ is negative enough to make the visual input purely inhibitory, as reported (*Fisher et al., 2019*). The visual input is more inhibitory in the surround to suppress activity outside of the HD receptive field. Therefore, the mechanism in which the visual input acts on the HD neurons is disinhibition.

The firing rate of HD cells, which is set by the voltage in the axon-proximal compartment, is given by

$$\boldsymbol{r}^{HD} = f(\boldsymbol{V^a}) \tag{6}$$

where

$$f(x) = \frac{f_{max}}{1+\exp(-\beta(x-x_{1/2}))} \tag{7}$$

is a sigmoidal activation function applied element-wise to the vector $\boldsymbol{V^a}$. The variable $f_{max}$ sets the maximum firing rate of the neuron, $\beta$ is the slope of the activation function, and $x_{1/2}$ is the input level at which half of the maximum firing rate is attained. The value of $f_{max}$ is arbitrary, while $\beta$ is chosen such that the activation function has sufficient dynamic range, and $x_{1/2}$ is chosen such that for small negative inputs the activation function is non-zero.

We note that the saturation of the activation function $f$ in *Equation 7* is an essential feature of our model, especially for the convergence of the plasticity rule in *Equation 12*; see also the section 'Synaptic Plasticity Rule'. Even though, to the best of our knowledge, it is currently not known whether E-PG neurons actually reach saturation, other *Drosophila* neurons are known to reach saturation with increasing inputs, instead of some sort of depolarization block (*Wilson, 2013*; *Brandão et al., 2021*). Saturation with increasing inputs may be due to, for instance, short-term synaptic depression: beyond a certain frequency of incoming action potentials, the synaptic input current is almost independent of that frequency (*Tsodyks and Markram, 1997*; *Tsodyks et al., 1998*).

The firing rates of the HR cells are given by

$$\boldsymbol{r}^{HR} = f\left(W^{HD}\boldsymbol{r}_{LP}^{HD} + \boldsymbol{I}^{vel} + I_{inh}^{HR} + \sigma_n\boldsymbol{n}^{HR}\right) \tag{8}$$

where $\boldsymbol{r}^{HR}$ is the vector of length $N^{HR}$ of firing rates of HR cells, the $N^{HR} \times N^{HD}$ matrix $W^{HD}$ encodes the fixed connections from the HD to the HR cells, $\boldsymbol{r}_{LP}^{HD}$ is a low-pass filtered version of the firing rate of the HD cells where the filter accounts for delays due to synaptic transmission in the incoming synapses from HD cells, $\boldsymbol{I}^{vel}$ is the angular velocity input, $I_{inh}^{HR}$ is a constant inhibitory input common to all HR cells, and $\boldsymbol{n}^{HR}$ is a random noise input to the HR cells drawn IID from $N(0, 1)$. We set $I_{inh}^{HR}$ to a value that still allows sufficient activity in the HR cell bump, even when the animal does not move. The low-pass filtered firing-rate vector $\boldsymbol{r}_{LP}^{HD}$ is given by

$$\tau_s \frac{d\boldsymbol{r}_{LP}^{HD}}{dt} = -\boldsymbol{r}_{LP}^{HD} + \boldsymbol{r}^{HD} , \tag{9}$$

and the angular-velocity input to the i-th HR neuron is given by

$$\text{I}_i^{vel}(t) = q\,k\,v(t) \quad \text{with} \quad q = \begin{cases} -1 & \text{for } i \leq N^{HR}/2 \\ 1 & \text{for } i > N^{HR}/2 \end{cases} \tag{10}$$

where $k$ is the proportionality constant between head angular velocity and velocity input to the network, $v(t)$ is the head angular velocity at time $t$ in units of deg/s, and the factor $q$ is chosen such that the left (right) half of the HR cells are primarily active during leftward (rightward) head rotation. Note that the same $\tau_s$ is in both *Equation 2* and *Equation 9*. Finally, as mentioned earlier, the matrix $W^{HD}$ encodes the hardwired 1-to-1 HD-to-HR connections, i.e., $W_{ij}^{HD} = w^{HD}$ if the j-th HD neuron projects to the i-th HR neuron, and $W_{ij}^{HD} = 0$ otherwise. Specifically, for $j$ odd, HD neuron $j$ projects to L-HR neuron $i = \frac{j+1}{2}$, whereas for $j$ even, HD neuron $j$ projects to R-HR neuron $i = 30 + \frac{j}{2}$. The synaptic strength $w^{HD}$ is chosen such that the range of the firing rates of the HD cells is mapped to the entire range of firing rates of the HR cells. Specifically, we set $w^{HD} = \frac{A_{active}}{f_{max}}$, where $A_{active}$ is the range of inputs for which $f$ has not saturated, i.e., the input values for which $f$ remains between about 7% and 93% of its maximum firing rate $f_{max}$ (see *Equation 7*). Finally, the proportionality constant $k$ is set so that the firing rate of HR neurons does not reach saturation for the range of velocities relevant for the fly (approx. [-500, 500] deg/s), given all other inputs they receive.

## Synaptic plasticity rule

In our network, the associative HD neurons receive direct visual input in the axon-proximal compartment and indirect angular velocity input in the axon-distal compartment through the HR-to-HD connections (*Figure 1D*). We hypothesize that the visual input acts as a supervisory signal that controls the axon-proximal voltage $V^a$ directly, and the latter initiates spikes. Therefore, the goal of learning is for the axon-distal voltage $V^d$ to predict the axon-proximal voltage by changing the synaptic weights $W^{rec}$ and $W^{HR}$. This change is achieved by minimizing the difference between the firing rate $f(V^a)$ in the presence of visual input and the axon-distal prediction $f(V^{ss})$ of the firing rate in the absence of visual input. In the latter case and at steady-state, the voltage $V_i^{ss}$ for the i-th HD neuron is an attenuated version of the axon-distal voltage,

$$V_i^{ss} = \frac{g_D}{g_D + g_L} V_i^d, \tag{11}$$

with conductance $g_D$ of the coupling from the axon-distal to axon-proximal dendrites and leak conductance $g_L$ of the axon-proximal compartment, as explained in *Equation 4*, and $p = \frac{g_D}{g_D + g_L}$ in *Equation 1*. Therefore, following *Urbanczik and Senn, 2014*, we define the plasticity-induction variable $PI_{ij}$ for the connection between the j-th presynaptic neuron and i-th postsynaptic neuron as

$$PI_{ij} = \left[ f(V_i^a) - f(V_i^{ss}) \right] P_j \tag{12}$$

where $P_j$ is the postsynaptic potential of neuron $j$, which is a low-pass filtered version of the presynaptic firing rate $r_j$. That is,

$$P_j(t) = H(t) * r_j(t) \tag{13}$$

where * denotes convolution. The transfer function

$$H(t) = \frac{1}{\tau_l - \tau_s} \left[ \exp\left(-\frac{t}{\tau_l}\right) - \exp\left(-\frac{t}{\tau_s}\right) \right] u(t) \tag{14}$$

is derived from the filtering dynamics in *Equation 2* and *Equation 3* and accounts for the delays introduced by the synaptic time constant $\tau_s$ and the leak time constant $\tau_l$. In *Equation 14*, $u(t)$ denotes the Heaviside step function, that is, $u(t) = 1$ for $t > 0$ and $u(t) = 0$ otherwise. The plasticity-induction variable is then low-pass filtered to account for slow learning dynamics,

$$\tau_\delta \frac{d\delta_{ij}}{dt} = -\delta_{ij} + PI_{ij}, \tag{15}$$

and the final weight change is given by

$$\frac{dW_{ij}}{dt} = \eta \delta_{ij} \tag{16}$$

where $\eta$ is the learning rate and $W_{ij}$ is the connection weight from the j-th presynaptic neuron to the i-th postsynaptic neuron. Note that the synaptic weight $W_{ij}$ is an element of either the matrix $W^{rec}$ or the matrix $W^{HR}$ depending on whether the presynaptic neuron $j$ is an HD or an HR neuron, respectively. The value of the plasticity time constant $\tau_\delta$ is not known, therefore we adopt the value suggested by *Urbanczik and Senn, 2014*.

*Equation 12* is a 'delta-like' rule that can be interpreted as an extension of the Hebbian rule; compared to a generic Hebbian rule, we have replaced the postsynaptic firing rate $f(V_i^a)$ by the difference between $f(V_i^a)$ and the predicted firing rate $f(V_i^{ss})$ of the axon-distal compartment of the postsynaptic neuron. This difference drives plasticity in the model. We note that $f(V_i^a)$ is a continuous approximation of the spike train of the postsynaptic neuron, which could be available at the axon-distal compartment via back-propagating action potentials (*Larkum, 2013*). Furthermore, the axon-distal voltage $V_i^d$ and postsynaptic potentials are by definition available at the synapses arriving at the axon-distal compartment. Note that even though $f(V_i^{ss})$ is the firing rate in the absence of visual input, it can still be computed at the axon-distal compartment when the visual input is available; $V_i^d$ is the local voltage and therefore only a constant multiplicative factor (*Equation 11*) and the static nonlinearity $f$ need to be computed to obtain $f(V_i^{ss})$. Therefore, the learning rule is biologically plausible because all information is locally available at the synapse.

The learning rule used here differs from the one in the original work of *Urbanczik and Senn, 2014* because we utilize a rate-based version instead of the original spike-based version. Even though spike trains can introduce Poisson noise to $f(V_i^a)$, *Urbanczik and Senn, 2014* show that once learning has converged, asymmetries in the weights due the spiking noise are on average canceled out.

Another difference in our learning setup is that, unlike in *Urbanczik and Senn, 2014*, the input to the axon-proximal compartment does not reach zero in equilibrium (see, e.g. *Figure 3D*, and Appendix 5). Therefore, an activation function with a saturating non-linearity, as in *Equation 7*, is crucial for convergence, which could not be achieved with a less biologically plausible threshold-linear activation function. This lack of strict convergence in our setup is responsible for the square form of the bump (*Figure 3—figure supplement 2B* and Appendix 5).

## Training protocol

We train the network with synthetically generated angular velocities, simulating head turns of the animal. $W^{rec}$ and $W^{HR}$ are both initialized with random connectivity drawn from a normal distribution with mean 0 and standard deviation $1/\sqrt{N^{HD}}$, as common practise in the modeling literature. In further simulations with various other initial conditions (e.g. in the simulations with gain changes in *Figure 4* or in simulations in which we randomly shuffled weights after learning, not shown), we confirmed that the final PI performance is virtually independent of the initial distribution of weights $W^{rec}$ and $W^{HR}$.

The network dynamics are updated in discrete time steps $\Delta t$ using forward Euler integration. The entrained angular velocities cover the range of angular velocities exhibited by the fly, which are at maximum ~500 deg/s during walking or flying (*Geurten et al., 2014*; *Stowers et al., 2017*). The angular velocity $v(t)$ is modeled as an Ornstein-Uhlenbeck process given by

$$v(t + \Delta t) = (1 - \alpha)\, v(t) + \sigma_v \sqrt{\Delta t}\, n(t) \tag{17}$$

where $\alpha = \Delta t / \tau_v$ and $\tau_v$ is the time constant with which $v(t)$ decays to zero, $n(t)$ is noise drawn from a normal distribution with mean 0 and standard deviation 1 at each time step, and $\sigma_v$ scales the noise strength.

We pick $\sigma_v$ and $\tau_v$ so that the resulting angular velocity distribution in *Figure 3—figure supplement 2C* and its time course, for example in *Figure 2A*, are similar to what has been reported in flies during walking or flying (*Geurten et al., 2014*; *Stowers et al., 2017*). Finally, note that we train the network for angular velocities a little larger than what flies typically display (up to ±720 deg/s).

## Quantification of the mean learning error

In *Equation 12* we have used the learning error

$$E_i = f(V_i^a) - f(V_i^{ss}) \tag{18}$$

which controls learning in the i-th associative HD neuron. To quantify the mean learning error $err(t)$ in the whole network at time $t$, we average $E_i$ across all HD neurons and across a small time interval $[t, t + t_w]$, that is,

$$err(t) = \frac{1}{t_w N^{HD}} \sum_{i=1}^{N^{HD}} \int_t^{t+t_w} |E_i(\tau)| \, d\tau \tag{19}$$

with $t_w = 10$ s. In *Figure 3D*, we plot this mean error at every 1% of the simulation, for 12 simulations, and averaged across the ensemble of the simulations. Note that individual simulations occasionally display 'spikes' in the error. Large errors occur if the network happens to be driven by very high velocities that the network does not learn very well because they are rare; larger errors also occur for very small velocities, that is, when the velocity input is not strong enough to overcome the local attractor dynamics, as seen, for example, in *Figure 2C*. On average, though, we can clearly see that the mean learning error decreases with increasing time and settles to a small value (e.g. *Figure 3D* and *Figure 4A*).

## Population vector average

To decode from the activity of HD neurons an average HD encoded by the network, we use the population vector average (PVA). We thus first convert the tuning direction $\theta_i$ of the i-th HD neuron to the corresponding complex number $e^{j\theta_i}$ on the unitary circle, where $j$ is the imaginary unit. This complex number is multiplied by the firing rate $r_i^{HD}$ of the i-th HD neuron, and then averaged across neurons to yield the PVA

$$r_{av} = \frac{1}{N^{HD}} \sum_{i=1}^{N^{HD}} r_i^{HD} e^{j\theta_i} . \tag{20}$$

The PVA is a vector in the 2-D complex plane and points to the center of mass of activity in the HD network. Finally, we take the angle $\theta$ of the PVA as a measure for the current heading direction represented by the network.

## Diffusion coefficient

To quantify the variability of heading direction in the trained networks, we define the diffusion coefficient $D$ as:

$$D = \frac{\langle \Delta\theta^2 \rangle - \langle \Delta\theta \rangle^2}{t_{sim}} \tag{21}$$

where $\Delta\theta$ is the change in heading direction in a time interval $t_{sim}$. Therefore, $D$ is given by the variance of the distribution of displacements in a given time interval, divided by the time interval.

In the main text, we estimate $D$ during PI, i.e. with velocity inputs only. In this setting, $D$ is the rate at which the variance of the PI errors increases (see e.g. *Figure 2B*). Deviations from gain-1 PI contribute to this estimate; hence, to single out the effects of noise during training on the stability of the learned attractor in Appendix 1, we also estimate $D$ in the presence of test noise when no inputs are received at all.

## Fly connectome analysis

Our model assumes the segregation of visual inputs to HD (E-PG) cells from head rotation and recurrent inputs to the same cells. To test this hypothesis, we leverage on the fly hemibrain connectome (*Xu, 2020*; *Clements et al., 2020*). First, we randomly choose one E-PG neuron per wedge of the EB, for a total of 16 E-PG neurons. We reasoned this sample would be sufficient because the way E-PG neurons in the same wedge are innervated is expected to be similar. We then find all incoming connections to these neurons from visually responsive ring neurons R2 and R4d (*Omoto et al., 2017*; *Fisher et al., 2019*). These are the connections that arrive at the axon-proximal compartment in our model. We then find all incoming connections from P-EN1 cells, which correspond to the HR neurons, and from P-EN2 cells, which are involved in a recurrent excitatory loop from E-PG to P-EG to P-EN2 and back to E-PG (*Turner-Evans et al., 2020*). These are the connections that arrive at the axon-distal compartment in our model.

To further support the assumption that visual inputs are separated from recurrent and HR-to-HD inputs in the *Drosophila* EB, we perform binary classification between the two classes (R2 and R4d vs.

P-EN1 and P-EN2). We use SVMs with Gaussian kernel, and perform nested 5-fold cross validation, for a total of 30 model runs for every neuron tested (*Figure 1—figure supplement 1*).

## Quantification of PI performance

To quantify PI performance of the network and compare to fly performance, we use the measure defined by *Seelig and Jayaraman, 2015* and estimate the correlation coefficient between the unwrapped PVA and true heading in darkness. We estimate the correlation in 140 s long trials and report the point estimate and 95% confidence intervals (Student's t-test, $N = 100$).

## Resource availability

All code used in this work is available at https://github.com/panvaf/LearnPI (copy archived at swh:1:rev:c6e354f80bf435114e577af70892db41c3ce5315, *Vafidis, 2022*). The files required to reproduce the figures can be found at https://gin.g-node.org/pavaf/LearnPI.

## Acknowledgements

We thank Raquel Suárez-Grimalt and Marcel Heim for helpful discussions and Louis Kang for comments on the manuscript. This work was funded by the Deutsche Forschungsgemeinschaft (DFG, German Research Foundation; SFB 1315 – project-ID 327654276 to RK and DO; and the Emmy Noether Programme 282979116 to DO and Germany´s Excellence Strategy – EXC-2049 – 390688087 to DO), the German Federal Ministry for Education and Research (BMBF; Grant 01GQ1705 to RK), and the Onassis Foundation (PV). The funding sources were not involved in study design, data collection and interpretation, or the decision to submit the work for publication.

## Additional information

### Funding

| Funder | Grant reference number | Author |
| --- | --- | --- |
| German Research Foundation | SFB 1315 - project-ID 327654276 | David Owald Richard Kempter |
| Emmy Noether Programme | 282979116 | David Owald |
| Federal Ministry of Education and Research | 01GQ1705 | Richard Kempter |
| Onassis Foundation | | Pantelis Vafidis |
| Charité – Universitätsmedizin Berlin | | David Owald |

The funders had no role in study design, data collection and interpretation, or the decision to submit the work for publication.

### Author contributions

Pantelis Vafidis, Conceived the study, Performed analyses, Wrote the initial draft of the manuscript, Wrote the manuscript; David Owald, Supervised the research, Wrote the manuscript; Tiziano D'Albis, Conceived the study, Contributed to analyses, Supervised the research, Wrote the manuscript; Richard Kempter, Conceived the study, Supervised the research, Wrote the manuscript

### Author ORCIDs

Pantelis Vafidis http://orcid.org/0000-0002-9768-0609
David Owald http://orcid.org/0000-0001-7747-7884
Tiziano D'Albis http://orcid.org/0000-0003-1585-1433
Richard Kempter http://orcid.org/0000-0002-5344-2983

### Decision letter and Author response

Decision letter https://doi.org/10.7554/eLife.69841.sa1

Author response https://doi.org/10.7554/eLife.69841.sa2

## Additional files

### Supplementary files
• Transparent reporting form

### Data availability
All code used in this work is available at https://github.com/panvaf/LearnPI, (copy archived at swh:1:rev:c6e354f80bf435114e577af70892db41c3ce5315). The files required to reproduce the figures can be found at https://gin.g-node.org/pavaf/LearnPI.

The following previously published dataset was used:

| Author(s) | Year | Dataset title | Dataset URL | Database and Identifier |
|---|---|---|---|---|
| Cs Xu, Januszewski M, Lu Z, ya Takemura S, Hayworth KJ | 2020 | A Connectome of the Adult Drosophila Central Brain | https://elifesciences.org/articles/57443 | neuPrint, 57443 |

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

# Appendix 1

## Robustness to noise

In the main text, the only source of stochasticity in the network came from the angular velocity noise in the Ornstein-Uhlenbeck process (Materials and methods, *Equation 17*). Biological HD systems, however, are subject to other forms of biological noise like randomness of ion channels. To address that, we include Gaussian IID synaptic current noise to every location in the network where inputs arrive: the axon-proximal and axon-distal compartments of HD cells and the HR cells (see Materials and methods, parameterized by $\sigma_n$ in *Equation 2*, *Equation 4*, and *Equation 8*). We then ask how robustly can the network learn in the presence of such additional stochasticity.

To quantify the network's robustness to noise, we need to define a comparative measure of useful signals vs. noise in the network. By 'signals' we refer to the velocity/visual inputs and any network activity resulting from them, whereas 'noise' is the aforementioned Gaussian IID variables. We thus define the signal-to-noise ratio (*SNR*) to be the squared ratio of the active range $A_{active}$ of the activation function $f$ (defined in *Equation 7*) over two times the standard deviation of the Gaussian noise, $\sigma_n$, i.e.

$$SNR = \left( \frac{A_{active}}{2\sigma_n} \right)^2. \tag{22}$$

This definition is motivated by the fact that $A_{active}$ determines the useful range that signals in the network can have. If any of the signals exceed this range, they cannot impact the network in any meaningful way because the neuronal firing rate has saturated, unless they are counterbalanced by other signals reliably present. The factor 2 in the denominator is due to the fact that the noise can extend to both positive and negative values, whereas $A_{active}$ denotes the entire range of useful inputs.

Here, we vary the *SNR* and observe its impact on learning and network performance. *Appendix 1—figure 1A–D* shows the performance of a network that has been trained with $SNR \approx 2$. The resulting network connectivity remains circularly symmetric and maintains the required asymmetry in the HR-to-HD connections for L- and R-HR cells (data not shown). Therefore we plot only the profiles in *Appendix 1—figure 1B*, which look very similar to the ones in *Figure 3C* trained with $SNR = \infty$. The peak of the local excitatory connectivity in $W^{rec}$ is not as pronounced. This happens because the noise corrupts auto-correlations of firing during learning.

The network activity still displays a clear bump that smoothly follows the ground truth in the absence of visual input (*Appendix 1—figure 1A*). There are only minor differences compared to the network without noise (*Figure 2A*). The presence of the noise is most obvious in the HR cells, since HD cells that do not participate in the bump are deep into inhibition, and therefore synaptic input noise does not affect as much their activity. We note that the network can no longer sustain a bump in darkness when $SNR = 1$, that is when the standard deviation of the noise covers the full active range of inputs (data not shown).

Finally, the neural velocity slightly overestimates the head angular velocity (*Appendix 1—figure 1C* compared to *Figure 2C*), and the PI errors diffuse faster in the network with noise ($88.1 \deg^2/s$ in *Appendix 1—figure 1D* compared to $24.5 \deg^2/s$ in *Figure 2B*); these values are also indicated in *Appendix 1—figure 1E* (triangles). Importantly, we find that the diffusion assumption holds, because the estimation of the diffusion coefficient when varying simulation time (between 10 and 60 s) is consistent.

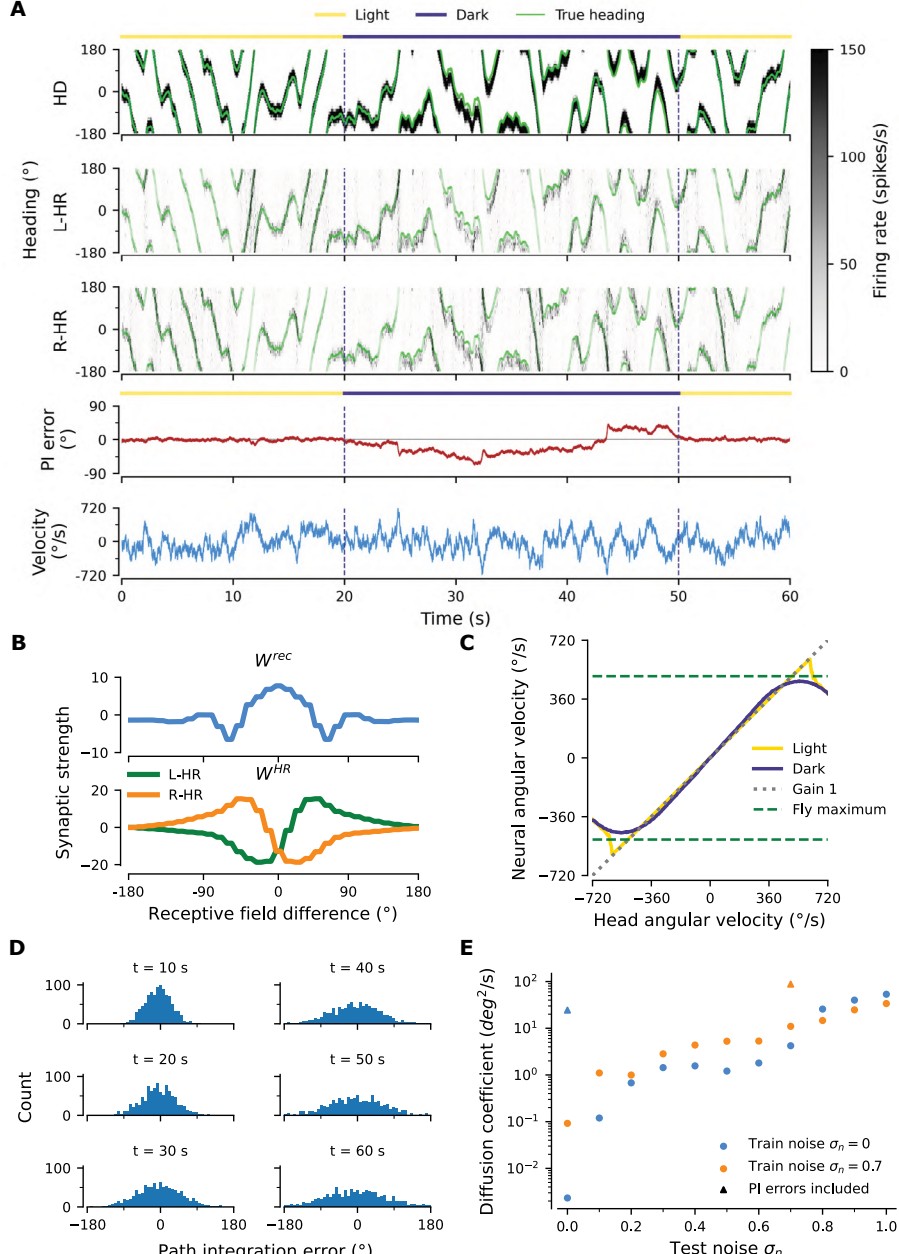

**Appendix 1—figure 1.** Robustness to injected noise. (**A**) PI example in a network trained with noise ($SNR \approx 2$, train noise $\sigma_n = 0.7$). Panels are organized as in **Figure 2A**, which shows the activity in a network trained without noise ($SNR = \infty$, $\sigma_n = 0$). (**B**) Profiles of learned weights. Both $W^{rec}$ and $W^{HR}$ are circularly symmetric. Panel is organized as in **Figure 3C**, which shows weight profiles in a network trained without noise ($SNR = \infty$, $\sigma_n = 0$). (**C**) The network achieves almost perfect gain-1 PI, despite noisy inputs. Compared to **Figure 2C** the performance is only slightly impaired. (**D**) Temporal evolution of distribution of PI errors during PI in darkness. Compared to **Figure 2B** the distribution widens faster, however it also does not exhibit side bias. (**E**) Diffusion coefficient for networks as a function of the level of test noise (for details, see section "Diffusion Coefficient" in Materials and methods). We distinguish between networks that experienced noise during training ($\sigma_n = 0.7$, orange) and networks that were trained without injected noise ($\sigma_n = 0$, blue), which were studied in the Results of the main text. Diffusion coefficients that include contributions from PI errors, estimated from (**D**) and **Figure 2B**, are also plotted (triangles).

So far we have addressed the diffusivity of PI in networks that receive velocity input, and we have compared the performance of networks that were trained with and without synaptic input noise. However, in mature networks, that is during testing, it is unclear how large the impact of synaptic

input noise is, compared to noise that originates from imperfect PI. To disentangle these two noise contributions during testing, we study diffusivity in networks that do not receive velocity inputs at all (i.e. without PI). In the absence of velocity inputs, we vary the level of synaptic input noise during testing (called 'test noise'), and estimate diffusion coefficients. Specifically, for each test noise magnitude $\sigma_n$ we run 1000 simulations of $t_{sim} = 10$ s, each time randomly initializing the network at one of the angular locations $\theta_i$ for which HD neurons are tuned for. For $\sigma_n = 0$, we run one simulation per $\theta_i$, since the simulation is deterministic.

Our results in *Appendix 1—figure 1E* show that the diffusion coefficients obtained in networks without velocity inputs (dots) are always much smaller that those with velocity inputs (triangles). Thus, imperfect integration of velocity inputs is by far the dominating source of noise in trained, mature networks. Omitting the velocity inputs, we detected small differences between networks that were trained without synaptic input noise (blue dots) and with such noise (orange dots, train noise $\sigma_n = 0.7$). The network trained with noise is sligthly more diffusive up to the level of test noise at which is was trained ($\sigma_n = 0 - 0.7$), and it is slightly less diffusive beyond that level ($\sigma_n > 0.7$).

We conclude that learning PI is robust to synaptic input noise during learning, and that synaptic input noise during testing degrades performance much less than errors due to deviations from perfect gain-1 PI, which are already quite small.

## Appendix 2

## Synaptic delays set a neural velocity limit during path integration

In the main text we trained networks for a set of angular velocities that cover the full range exhibited by the fly ($|v| < 500$ deg/s), and we showed that the mature network can account for several key experimental findings. However, the ability of any continuous attractor network to path-integrate is naturally limited for high angular velocities, due to the synaptic delays inherent in any such network (*Zhong et al., 2020*). To evaluate the ability of our network to integrate angular velocities, we sought to identify a limit of what velocities could be learned.

The width of the HD bump in our network is here termed *BW*, and it is largely determined by the width $\sigma$ of the visual receptive field. This is because during training we force the network to produce a bump with a width matching that of the visual input, and this width is then maintained when the latter is not present. The reason for this behavior is that the width of the learned local excitatory connectivity profile in $W^{rec}$ that guarantees such stable bumps of activity will be similar to the width of the bump, because recurrent connections during learning are only drawn from active neurons (non-zero $P_j$ in *Equation 12*). As mentioned in the main text, this emphasizes the Hebbian component of our learning rule (fire together — wire together). As a result, the width of local excitatory recurrent connections should be approximately *BW*.

In *Figure 3—figure supplement 1* we show that the higher angular velocities are served by the long-range excitatory connections in $W^{HR}$. However, these connections might not be strong enough to move the bump by themselves; a contribution from HD cells might still be needed to move the bump at such high angular velocities. In that case, the width of the connectivity bump in $W^{rec}$ might limit how *far away* from the current location the bump can be moved. In addition, there is a limitation in how *quickly* the bump can be moved: the learning rule in *Equation 1* tries to predict the next state of the network from the current state; but to activate the next HD neurons in line, current HD and HR cell activity must go through the synaptic delay $\tau_s$. Therefore, the maximum velocity that the network can achieve without external guidance (i.e. without visual input) should be inversely proportional to $\tau_s$, i.e.

$$v_{max} = \frac{b(\sigma, \Delta\phi)}{\tau_s} \tag{23}$$

where we assume that $b$ might reflect an effective HD connectivity bump width, which depends on $\sigma$ but also on the angular resolution of the HD network $\Delta\phi$, due to discretization effects. In reality, the HR-to-HD connectivity profiles in $W^{HR}$ likely also have a bearing on $b$.

We then systematically vary $\tau_s$ and test what velocities the network can learn. We indeed find that networks can path-integrate all angular velocities up to a limit, but not higher than that. As predicted, this limit is inversely proportional to $\tau_s$, for a wide range of delays (*Appendix 1—figure 1A*). Furthermore, $b$ matches *BW* reasonably well. Fitting *Equation 23* to the data we obtain $b(0.25, 6 \text{ deg}) \approx BW = 96 \text{ deg}$ for $N^{HD} = N^{HR} = 120$ and $b(0.15, 12 \text{ deg}) = 75 \text{ deg}$, $BW = 60 \text{ deg}$ for $N^{HD} = N^{HR} = 60$.

As mentioned in the main text, there are two limitations other than synaptic delays why the network could not learn high angular velocities: limited training of these velocities, and saturation of HR cell activity. These limitations kick in for $\tau_s < 150 \text{ ms}$, for which $v_{max}$ matches the maximum velocity the fly displays (500 deg/s). Therefore to create *Appendix 1—figure 1A* for these delays, we increased the standard deviation of the velocity noise in the Ornstein-Uhlenbeck process to $\sigma_v = 800 \text{ deg/s}$ to address the first limitation, and we increased the dynamic range of angular velocity inputs by decreasing the proportionality constant in *Equation 10* to $k = 1/540 \text{ s/deg}$ to address the second.

The velocity gain plot for an example network with high synaptic delays ($\tau_s = 190 \text{ ms}$) is shown in *Appendix 1—figure 1B*. Interestingly, we notice that the performance drop at the velocity limit is not gradual; instead, the neural velocity abruptly drops to a near-zero value once past the velocity limit. Further investigation reveals that for velocities higher than this limit, the network can no longer sustain a bump (*Appendix 1—figure 1C*). This happens because the HD network cannot activate neurons downstream fast enough to keep the bump propagating, and therefore the bump disappears and the velocity gain plot becomes flat.

*Equation 23* is similar to a relationship reported in *Turner-Evans et al., 2017* (their page 35, 1st paragraph), where it was demonstrated that the phase shift of the HR population bump compared to the HD bump limits angular velocity. Our result hence generalizes this finding in the case where recurrent connections between HD neurons are also allowed.

Finally, we note that so far we only tested the limits of network performance when increasing $\tau_s$. To demonstrate that smaller delays also work, as an extreme example we show PI performance in a network where $\tau_s = 1\,\mathrm{ms}$ in *Appendix 1—figure 1D and E*. A potential issue with such small synaptic delays is that the network would not be able to distinguish rightward from leftward rotation for small angular velocities, because the motion direction offset of the HR bumps would be small, and the activity in the two HR populations comparable. In such a setting it is harder to learn the asymmetries in the HR-to-HD connections required to differentiate leftward from rightward movement. Indeed, this effect is visible in *Appendix 1—figure 1E* where the amplitude of HR-to-HD connections has been suppressed, and in *Appendix 1—figure 1D* where the flat region for small angular velocities has been extended compared to *Figure 2C*.

Overall, these results indicate that the network learns to path-integrate angular velocities up to a fundamental limit imposed by the architecture of the HD system in the fly. Furthermore, we conclude that the phenomenological delays observed in the fly HD system in *Turner-Evans et al., 2017* are not fundamentally limiting the system's performance, since they can support PI for angular velocities much higher than the ones normally displayed by the fly.

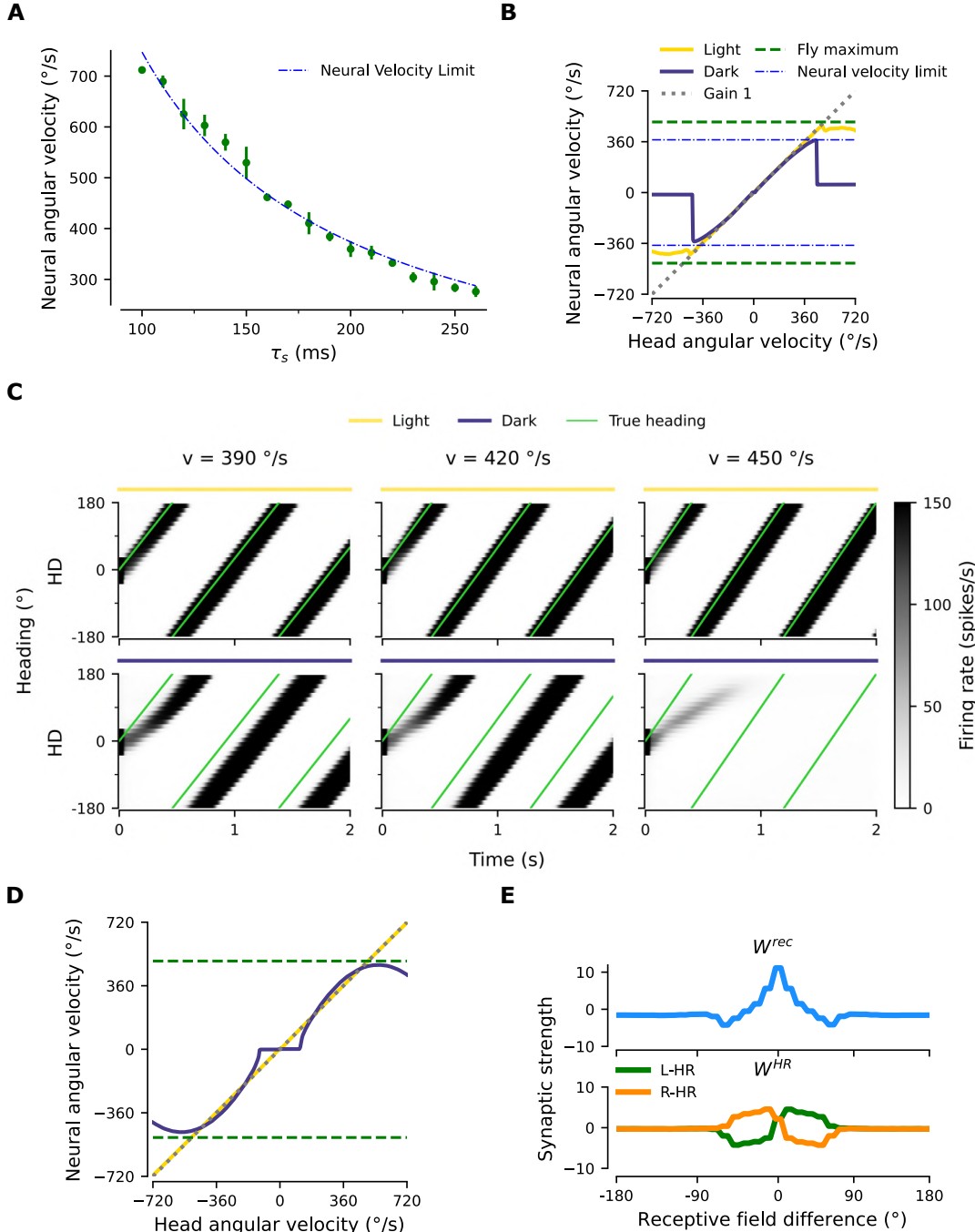

**Appendix 2—figure 1.** Limits of network performance when varying synaptic delays. (**A**) Maximum neural angular velocity learned is inversely proportional to the synaptic delay $\tau_s$ in the network, with constant $b = 75\,\mathrm{deg}$ in *Equation 23* (blue dot-dashed line). Green dots: point estimate of maximum neural velocity learned, green bars: 95% confidence intervals (Student's t-test, $N = 5$). (**B**) Example neural velocity gain plot (as in *Figure 2C*) in a network with increased synaptic delays ($\tau_s$ increased from the "standard" value 65 ms to the new value 190 ms). (**C**) Behavior of the activity of HD cells in the network with parameters as in (**B**) near the velocity limit. The example network is driven by a single velocity in every column, in light (top row) and darkness (bottom row) conditions. In darkness, near and below the limit (left and middle column), there is a delay in the appearance of the bump, which then path-integrates with gain 1; above the limit observed in (**B**), however, the bump cannot stabilize, resulting in the dip in neural velocity (right column). (**D**) PI performance for a network with drastically reduced synaptic delays ($\tau_s = 1\mathrm{ms}$). Compared to *Figure 2C* performance is worse for small angular velocities. This occurs because for small angular velocities, the offset of the HR bump in leftward vs. rightward movement is not as pronounced. As a result, it is harder to differentiate leftward from rightward movement. (**E**) For the same reason, the asymmetries in the learned HR-to-HD connectivity are not as prevalent as in *Figure 3C*.

## Appendix 3

### Robustness to architectural asymmetries

The networks we have trained in the Results had a circular symmetric initial architecture, including the hardwired HD-to-HR connections $W^{HD}$. However, such symmetry is unrealistic for any biological system that is assembled by imperfect processes; deviations from symmetry should be expected. Therefore, in this Appendix we let $W^{HD}$ vary randomly, and observe how PI performance is affected.

First, we remind the reader that the magnitude $w^{HD}$ of the HD-to-HR connections is chosen so that we take advantage of the full dynamic range of HR neurons; however, the exact magnitude should not be critical for our model. Homeostatic plasticity could adjust the magnitude, but for simplicity we have not incorporated such plasticity rules in our model. Instead, to see whether the exact values of synaptic weights, their circular symmetry, and the 1-to-1 nature of the HD-to-HR connections is crucial for our model, we draw connection strengths randomly.

In a first approach, we let HD neurons project also to adjacent HR neurons. Specifically, if $U(a, b)$ denotes the uniform distribution in the interval $(a, b)$, we sample the magnitude of weights from $U\left(\frac{w^{HD}}{2}, w^{HD}\right)$ for the main diagonal and $U\left(0, \frac{3 w^{HD}}{4}\right)$ for the side diagonals of $W^{HD}$ (*Appendix 3— figure 1A*). We then adjust the network connectivity ($W^{rec}$ and $W^{HR}$) in a learning phase, similar to the one illustrated in *Figure 3E and F*. After learning, as shown in *Appendix 3—figure 1C-E*, PI is still excellent because the learning rule can balance out any deviations from circular symmetry in $W^{HD}$. It does so by introducing deviations from circular symmetry in the learned weights, mainly in $W^{HR}$ (*Appendix 3—figure 1B*). Thus, small deviations from circular symmetry in the learned weights are essential for PI.

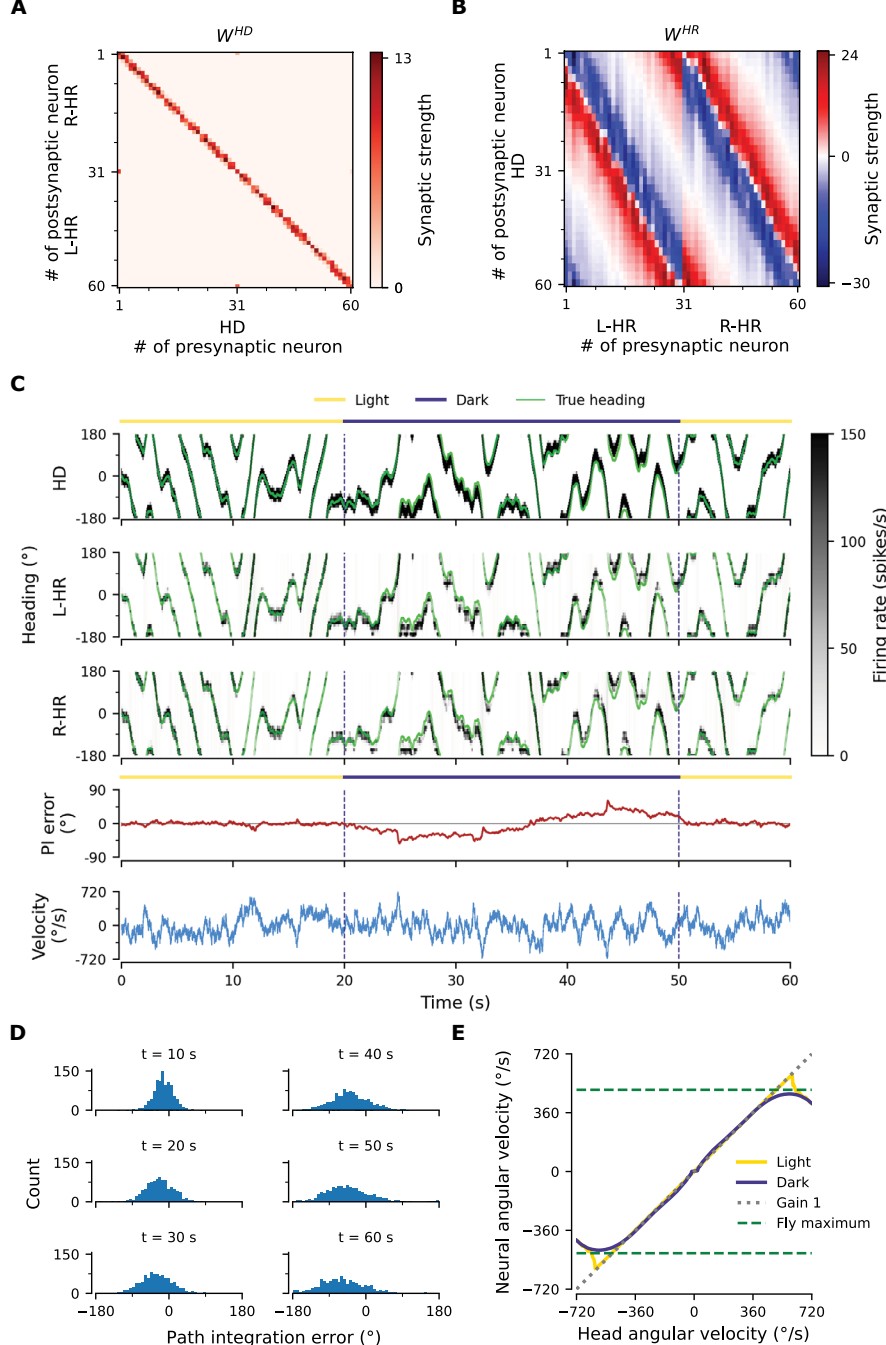

**Appendix 3—figure 1.** Performance of a network where HD-to-HR connection weights are allowed to vary randomly, and HD neurons are projecting to HR neurons also adjacent to the ones they correspond to, respecting the topography of the protocerebral bridge (PB). (**A**) The HD-to-HR connectivity matrix, $W^{HD}$. Note that, compared to what is described in the Materials and methods (final paragraph of 'Neuronal Model'), the order of HD neurons is rearranged: we have grouped HD neurons that project to the same wing of the PB together, so that the diagonal structure of the connections is clearly visible. (**B**) The learned HR-to-HD connections, $W^{HR}$, depart from circular symmetry (as, e.g., in *Figure 3B*), so that asymmetries in $W^{HD}$ could be counteracted. The recurrent connections $W^{rec}$ (not shown) remain largely unaltered compared to the ones shown in *Figure 3A*. (**C–E**) Despite the randomization and lack of 1-to-1 nature of HD-to-HR connections, PI in the converged network remains excellent (*Figure 2A–C*).

We illustrate the necessity to counterbalance small deviations of circular symmetry again in a second example that is based on the symmetric network studied in the Results. Here, we use the connectivity

of the network illustrated in *Figure 3A–C* and also preserve the 1-to-1 nature of HD-to-HR connections, but now we randomly vary their magnitude, while maintaining the same average connection strength; specifically, we sample the magnitude of weights from $U\left(\frac{w^{HD}}{2}, \frac{3w^{HD}}{2}\right)$. PI performance in this network is considerably impaired compared to the original network (compare *Appendix 3—figure 2* to *Figure 2A and C*). This is a further argument in favor of synaptic plasticity operating to fine-tune connectivity, because as mentioned we expect that such anatomical asymmetries are indeed present in the biological circuit. Therefore, even if the circular symmetric synaptic weights were passed down genetically with great accuracy, PI performance in flies should be considerably degraded for a biological circuit with anatomical asymmetries when no learning is involved.

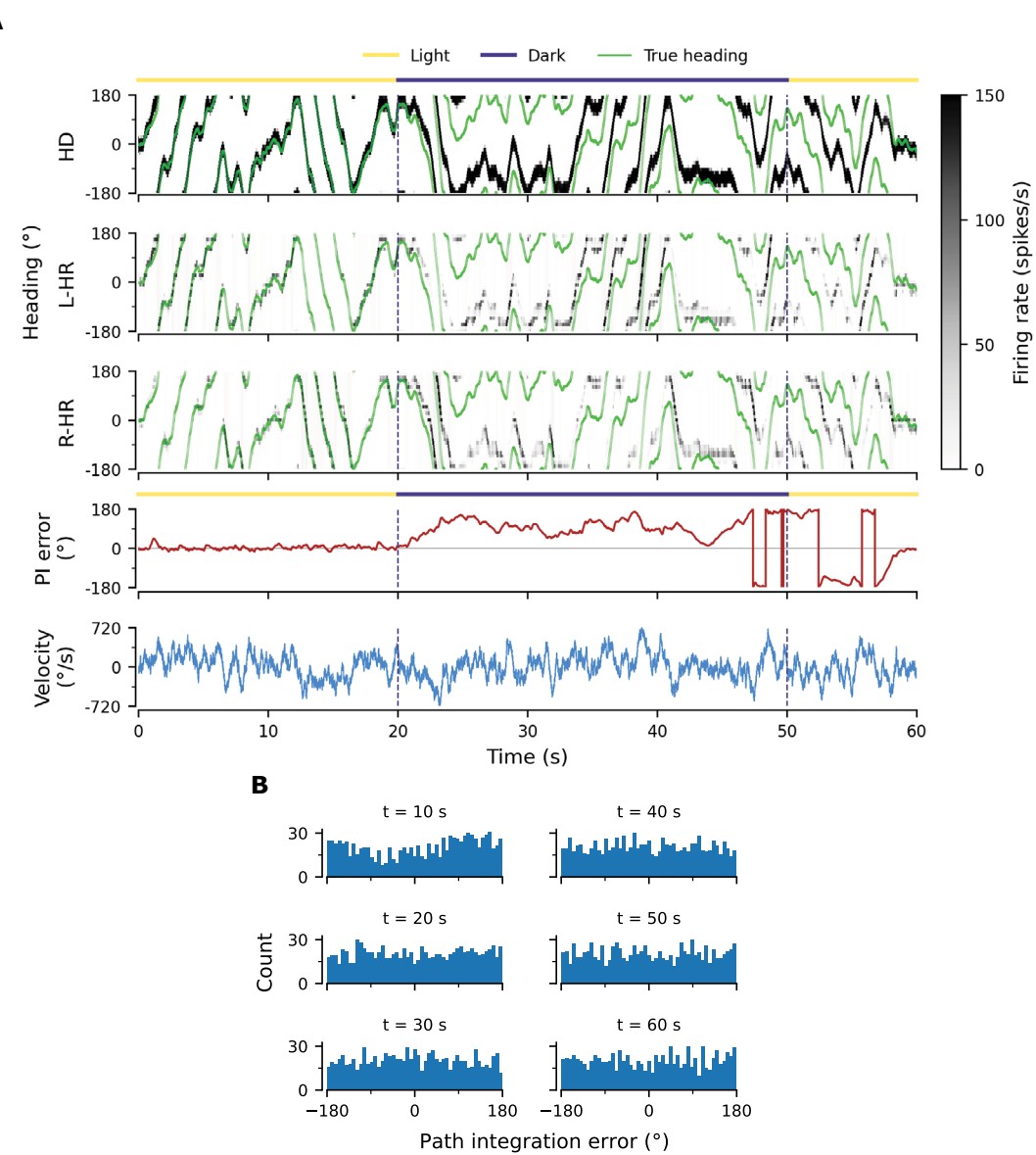

**Appendix 3—figure 2.** PI performance in a network with random HD-to-HR connection strengths and learned weights from network in *Figure 3*. Here we vary the magnitude of the main diagonal HD-to-HR connections but preserve the 1-to-1 nature of the connections. We assume that $W^{rec}$ and $W^{HR}$ are passed down genetically (i.e. there is no further learning of these connections), and therefore the same, circular symmetric profiles apply to every location in the circuit. We choose these (assumed here to be genetically stored) profiles to be the ones we learned in the network outlined in *Figure 3A and B*. (**A**) Example that shows that PI is impaired, because the circular symmetric profiles passed down genetically cannot counteract small asymmetries in the architecture that are likely to be present in any biological system. Notice that it can even take several seconds for the large PI error to be corrected by the visual input. (**B**) PI errors grow fast (compare to e.g. *Figure 2B*). Already by $20 \sec$ of PI the heading estimate is random.

To better quantify the effect of anatomical asymmetries, we incorporate both noise in the learned connectivity as in *Figure 3—figure supplement 3A*,B and noise in the HD-to-HR connections as in *Appendix 3—figure 1A*. We tune the noise independently for each weight matrix: for $W^{rec}$ and $W^{HR}$ we set the variance of the Gaussian noise to $p$ times the variance of the individual weight matrices, while for $W^{HD}$ we draw the connections connections from $U\left((1-p)\,w^{HD}, (1+p)\,w^{HD}\right)$ for the main diagonal and $U\left(0, p\,w^{HD}\right)$ for the side diagonals. We find that for $p = 0.3$ the correlation between the PVA and true heading in darkness drops to $0.27 \pm 0.09$ which is below reported fly PI performance (mean correlation across animals $\sim 0.5$ in *Seelig and Jayaraman, 2015*), while the structure of the weights is preserved. We observed a steep decline in the correlation coefficient between $p = 0.25$ (for which the correlation is $0.92 \pm 0.04$) and $p = 0.3$. Furthermore, we study the effect of perturbing individual weight matrices, and find that perturbing only $W^{HD}$ with $p = 0.3$ considerably affects performance (correlation $0.39 \pm 0.09$) while perturbation of $W^{rec}$ and $W^{HR}$ together has a much smaller effect on performance. This again argues in favor of learning $W^{rec}$ and $W^{HR}$ to counterbalance asymmetries in $W^{HD}$ (*Appendix 3—figure 1*). Furthermore, note that confounders other than imperfect weights might be responsible for the degradation of PI performance in the fly, which further argues in favor of learning.

As a final test for the capability of the learning rule to balance anatomical asymmetries, in *Appendix 3—figure 3* we use a completely random connectivity for HD-to-HR connections, drawing weights from a folded Gaussian distribution. We find that even then, PI performance of the converged network is great, albeit for a smaller range of velocities. In addition, bumps are not clearly visible in the HR populations anymore; in the main network, HR bumps were inherited from the HD bump due to the sparseness of the HD-to-HR connections. However, when HD-to-HR connections are random, HR cells are no longer mapped to a topographic state space.

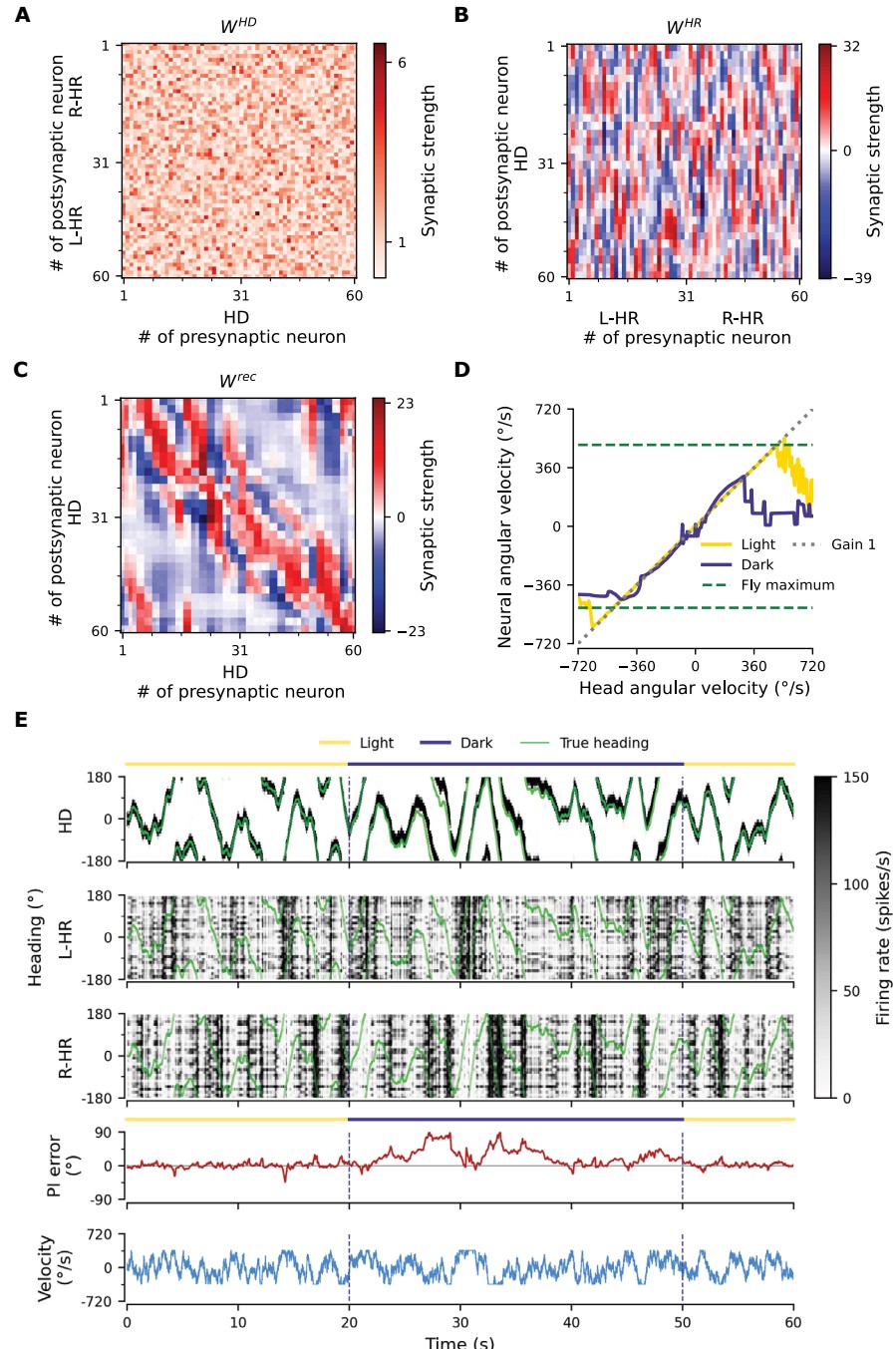

**Appendix 3—figure 3.** PI performance of a network where HD-to-HR connection weights are completely random. (**A**) The HD-to-HR weights are drawn from a folded normal distribution, originating from a normal distribution with 0 mean and $\pi(w^{HD})^2/200$ variance. (**B**) As a result, the learned HR-to-HD connections have also lost their structure. (**C**) The recurrent connections preserve some structure, since adjacency in the HD network is still important. (**D**) Impressively, the converged network can still PI with a gain close to 1, but for a reduced range of angular velocities compared to, e.g. the network in *Appendix 3—figure 1*. (**E**) A bump still appears in the HD network and gets integrated in darkness, albeit with larger errors. Note that bumps no longer appear in the HR populations; HR bumps are inherited from the HD bump only when adjacencies in the HD population are carried over to the HR populations by the HD-to-HR connections. Note that we have restricted angular velocities to the interval $[-360, 360]$ deg/s for this example, to showcase that PI is still accurate within this interval.

## Appendix 4

### Requirements on time scales

We devote this Appendix to discuss requirements for the time scales involved in our model (see *Appendix 4—table 1*). Several of these time scales are well constrained by biology, and thus we chose to keep them constant. These include the membrane time constants of the axon-proximal and axon-distal compartments, $C/g_L$ and $\tau_l$, respectively, which should be in the order of milliseconds; and the velocity decay time constant $\tau_v$, for which we choose a value in the same order of magnitude (0.5 s) as experimentally reported (*Turner-Evans et al., 2017*).

In general, the learning time scale given by $1/\eta$ should be the slowest one in our network model. The time scale should be large enough so that the network samples the input statistics for a long enough time. Varying $1/\eta$ from 2s to 20s to 200s, we find that the final learned weights are virtually identical (not shown). However, $1/\eta$ should not be too large to enable fast enough learning.

The synaptic time constant $\tau_s$ is determined from phenomenological delays in the network (*Turner-Evans et al., 2017*). Nevertheless, we also addressed the impact of varying $\tau_s$ in Appendix 2 and found that learning PI was robust in a wide range of values.

In additional simulations, we varied the weight update filtering time constant $\tau_\delta$ from 0 (effectively removing filtering) to 1000s (which is four orders of magnitude larger than the default value). We observed almost no effect on learning dynamics, and the performance of the final networks was almost identical (not shown). Since the specific value of $\tau_\delta$ is of little consequence in our network (if $1/\eta$ is large enough), there are hardly any limitations on its value compared to other time scales. Therefore, this justifies ignoring $\tau_\delta$ in the derivation of a reduced model without noise in Appendix 5.

**Appendix 4—table 1.** Default values for time scales in the model ordered with respect to their magnitude.

| Time scale | Expression | Value | Unit |
|---|---|---|---|
| Membrane time constant of axon-proximal compartment | $C/g_L$ | 1 | ms |
| Membrane time constant of axon-distal compartment | $\tau_l$ | 10 | ms |
| Synaptic time constant | $\tau_s$ | 65 | ms |
| Weight update filtering time constant | $\tau_\delta$ | 100 | ms |
| Velocity decay time constant | $\tau_v$ | 0.5 | s |
| Learning time scale | $1/\eta$ | 20 | s |

# Appendix 5

## Reduced model for a circular symmetric learned network

In this section, we derive a reduced model for the dynamics of the synaptic weights during learning. The goal is to gain an intuitive understanding of the structure obtained in the full model (*Figure 3* of the main text). Such a model reduction is obtained by (1) exploiting the circular symmetry in the system; (2) averaging weight changes across different speeds and moving directions; (3) writing dynamical equations in terms of convolutions and cross-correlations. With these methods, we derive a non-linear dynamical system for the weight changes as a function of head direction. Finally, we simulate this dynamical system and inspect how the different variables interact to obtain the final weights. We find that the reduced model results in nearly identical connectivity and learning dynamics to the full network in the main text, and explains how the latter assigns learning errors to the correct weights. Furthermore, it drastically reduces simulation times by two orders of magnitude.

Note that in this section we use slightly different notation compared to the main text. Notably, we refer to the recurrent head direction weight matrix simply as $W$ (omitting the superscript *rec*), and use capital letters for functions of time and small letters for functions of heading direction.

We study the learning equation (see *Equation 12–16* in the main text where the low-pass filtering with time constant $\tau_\delta$ has been ignored, since we find that the value of $\tau_\delta$ is not important for learning (see Appendix 4))

$$\frac{\mathrm{d}}{\mathrm{d}t} W_{ij}(t) = \eta \, E_i(t) \, P_j(t) \tag{24}$$

where

$$E_i(t) = f[V_i^a(t)] - f[V_i^{ss}(t)] \tag{25}$$

is the pre-synaptic error at the $i$-th cell and

$$P_j(t) = \int_0^\infty \mathrm{d}s \, H(s) f[V_j^a(t-s)] \tag{26}$$

is the post-synaptic potential at HD cell $j$, and $H$ is a temporal filter (with time constants $\tau_s$ and $\tau_l$, see *Equation 14* of the main text).

### Clockwise movement

Assuming that the head turns clockwise (which equals to rightward rotation, i.e. rotation towards decreasing angles) and anti-clockwise (leftward, i.e. towards increasing angles) with equal probability, we can approximate the weight dynamics by summing the average weight change $W_{ij}^+$ for clockwise movement and the average weight change $W_{ij}^-$ for anti-clockwise movement:

$$\frac{\mathrm{d}}{\mathrm{d}t} W_{ij}(t) = \frac{\mathrm{d}}{\mathrm{d}t} W_{ij}^+(t) + \frac{\mathrm{d}}{\mathrm{d}t} W_{ij}^-(t). \tag{27}$$

We start by assuming head movement at constant speed and we later generalize the results for multiple speeds. We compute the expected weight change $\frac{\mathrm{d}}{\mathrm{d}t} W_{ij}^+$ for one lap in the clockwise direction at speed $v^+ > 0$:

$$\frac{\mathrm{d}}{\mathrm{d}t} W_{ij}^+(t) = \frac{\eta v^+}{2\pi} \int_0^{2\pi/v^+} \mathrm{d}\tau \, E_i^+(\tau) \, P_j^+(\tau) \tag{28}$$

where

$$P_j^+(t) = \int_0^\infty \mathrm{d}s \, H(s) f[V_j^{a+}(t-s)] \tag{29}$$

is the post-synaptic potential for clockwise movement, and

$$E_i^+(t) = f[V_i^{a+}(t)] - f[V_i^{ss+}(t)] \tag{30}$$

is the error for a clockwise movement. Assuming that the axon-proximal voltage is at steady state (*Equation 4* of the main text with the l.h.s. set to zero and $I_{exc}^{HD}$ absorbed into $I_{vis}$), the clockwise axon-proximal voltage reads

$$V_i^{a+}(t) = V_i^{ss+}(t) + \frac{I_i^{vis}(t)}{g_D + g_L} \tag{31}$$

where (see *Equation 11* of the main text)

$$V_i^{ss+}(t) = \frac{g_D}{g_D + g_L} V_i^{d+}. \tag{32}$$

From *Equation 2 and 3* of the main text, we can write the axon-distal voltage $V_i^{d+}$ as a low-pass filtered version of the total axon-distal current $D_i^+$ for clockwise movement (see also *Equation 14* of the main text):

$$V_i^{d+} = \int_0^\infty ds\, H(s) D_i^+(t - s), \tag{33}$$

which yields

$$V_i^{ss+}(t) = \frac{g_D}{g_D + g_L} \int_0^\infty ds\, H(s) D_i^+(t - s). \tag{34}$$

Importantly, the visual input $I^{vis}$ is translation invariant:

$$I_j^{vis}(t) = I_i^{vis}\left(t + \frac{\theta_j - \theta_i}{v^+}\right) \tag{35}$$

where $\theta_j$ and $\theta_i$ are the preferred head directions of the j-th and i-th cell, respectively. As a result of this translation invariance, the recurrent weight matrix $W$ develops circular symmetry:

$$W_{ij} = W_{0,(j-i)N_{\mathrm{HD}}} \tag{36}$$

where $N_{\mathrm{HD}}$ is the number of HD cells in the system. Consequently, the post-synaptic potential $P_j^+$ is also translation invariant:

$$P_j^+(\tau) = P_i\left(\tau + \frac{\theta_j - \theta_i}{v^+}\right) = P_0\left(\tau + \overbrace{\frac{\theta_j - \theta_0}{v^+}}^{\theta :=}\right). \tag{37}$$

In this case, without loss of generality, we can rewrite *Equation 28* for a single row of the matrix $\frac{d}{dt} W_{ij}^+$ as a function of the angle difference $\theta := \theta_j - \theta_0$:

$$\frac{d}{dt} W_{ij}^+(t) = \frac{d}{dt} W_{0,(j-i)N_{\mathrm{HD}}}^+(t) = \frac{\eta v^+}{2\pi} \int_0^{2\pi/v^+} d\tau\, E_0^+(\tau) P_{(j-i)N_{\mathrm{HD}}}^+(\tau) \tag{38}$$

$$= \frac{\eta v^+}{2\pi} \int_0^{2\pi/v^+} d\tau\, E_0^+(\tau) P_0^+(\tau + \theta/v^+) \tag{39}$$

$$= \frac{\eta}{2\pi} \int_0^{2\pi} d\varphi\, E_0^+(\varphi/v^+) P_0^+[(\varphi + \theta)/v^+] \tag{40}$$

$$= \frac{\eta}{2\pi} \int_0^{2\pi} d\varphi\, \epsilon^+(\varphi) p^+(\varphi + \theta) \tag{41}$$

$$= \frac{\eta}{2\pi} (\epsilon^+ \star p^+)(\theta) \tag{42}$$

$$=: \frac{d}{dt} w^+(\theta). \tag{43}$$

where we defined $\epsilon^+(\varphi) := E_0^+(\varphi/v^+)$ and $p^+(\varphi) := P_0^+(\varphi/v^+)$, and $\star$ denotes circular cross-correlation.

From *Equation 29*, we derive

$$p^+(\varphi) := P_0^+(\varphi/v^+) = \int_0^\infty ds\, H(s) f[V_0^{a+}(\varphi/v^+ - s)] \tag{44}$$

$$\approx \int_0^{2\pi} d\beta\, \underbrace{\frac{1}{|v^+|} H(\beta/v^+)}_{=:\, h^+(\beta)} f[\underbrace{V_0^{a+}((\varphi - \beta)/v^+)}_{=:\, v^{a+}(\varphi - \beta)}] \tag{45}$$

$$= [h^+ * f(v^{a+})](\varphi) . \tag{46}$$

The approximation in *Equation 45* holds when a bump exists in the network and moves with a velocity below the velocity limit, and it is valid if the temporal filter $H$ is shorter than $2\pi/v^+$, that is for $H(t) \ll 1$ for $t > 2\pi/v^+$, which holds for the filtering time constants and velocity distribution we assumed (*Appendix 5—figure 1*). Therefore, plugging *Equation 46* into *Equation 43*, we obtain:

$$\frac{\mathrm{d}}{\mathrm{d}t} w^+(\theta) \approx \frac{\eta}{2\pi} \{\epsilon^+ \star [h^+ * f(v^{a+})]\}(\theta) . \tag{47}$$

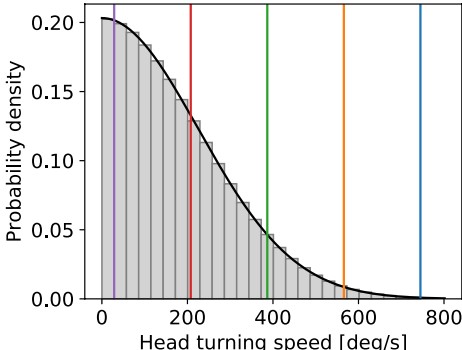
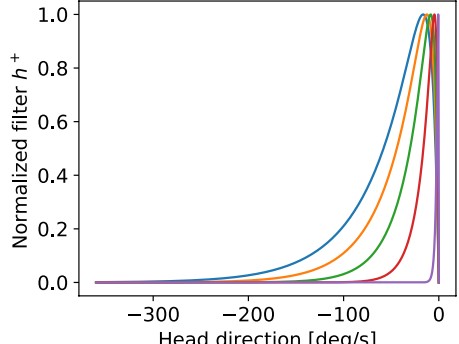

**Appendix 5—figure 1.** Left: assumed distribution of head-turning speeds (black) and discrete approximation used for the simulations. The colored vertical lines indicate speeds for which the filter $h^+$ is plotted in the right panel. Right: temporal filter $h^+(\theta)$ for several example speeds (see vertical lines in the left panel). Note that even for the largest speeds (blue curve) the filter decays within one turn around the circle.

By using the definition of $\epsilon(\varphi)^+$ we derive

$$\epsilon^+(\varphi) := E_0^+(\varphi/v^+) = f[\underbrace{V_0^{a+}(\varphi/v^+)}_{= v^{a+}(\varphi)}] - f[\underbrace{V_0^{ss+}(\varphi/v^+)}_{=: v^{ss+}(\varphi)}] \tag{48}$$

with (*Equation 31*)

$$v^{a+}(\varphi) = v^{ss+}(\varphi) + \underbrace{\frac{I_0^{vis}(\varphi/v^+)}{g_D + g_L}}_{=: \bar{I}_{vis}(\varphi)} \tag{49}$$

and (*Equation 34*)

$$v^{ss+}(\varphi) = \frac{g_D}{g_D+g_L} \int_0^\infty \mathrm{d}s \, H(s) D_0^+(\varphi/v^+ - s) \tag{50}$$

$$\approx \frac{g_D}{g_D+g_L} \int_0^{2\pi} \mathrm{d}\beta \, \underbrace{\frac{1}{|v^+|} H(\beta/v^+)}_{= h^+(\beta)} \underbrace{D_0^+\left(\frac{\varphi-\beta}{v^+}\right)}_{=: d^+(\varphi-\beta)} \tag{51}$$

$$\approx \frac{g_D}{g_D+g_L} (h^+ * d^+)(\varphi) \tag{52}$$

The approximation in *Equation 52* is valid if the temporal filter $H$ is shorter than $2\pi/v^+$, which again holds true for our parameter choices (*Appendix 5—figure 1*).

## Calculation of the axon-distal input

Let us compute the axon-distal current $D_i^+$ to the i-th neuron for clockwise movement. From *Equation 2* of the main text, setting the l.h.s. to zero, and splitting the rotation-cell activities in the two populations (L-HR and R-HR), we derive

$$D_i^+(t) = \underbrace{\sum_j W_{ij}(t) f[V_j^{a+}(t)]}_{:= D_i^{rec+}(t)} + \underbrace{\sum_j W_{ij}^R(t) f[V_j^{R+}(t)]}_{:= D_i^{R+}(t)} + \underbrace{\sum_j W_{ij}^L(t) f[V_j^{L+}(t)]}_{:= D_i^{L+}(t)} + I_{inhib}^{HD} .$$

(53)

where $W_{ij}^R$ ($W_{ij}^L$) are the weights from the right (left) rotation cells, and $V_j^{R+}$ ($V_j^{L+}$) are the voltages of the right (left) rotation cells (see *Equation 8–10* of the main text):

$$V_j^{R+}(t) = \frac{A_{active}}{f_{max}} \int_0^\infty ds\, H_s(s) f[V_j^{a+}(t-s)] + \bar{I}_{vel} + I_{inhib}^{HR}$$

(54)

$$V_j^{L+}(t) = \frac{A_{active}}{f_{max}} \int_0^\infty ds\, H_s(s) f[V_j^{a+}(t-s)] - \bar{I}_{vel} + I_{inhib}^{HR} .$$

(55)

The function $H_S(t) := \exp(-t/\tau_s)/\tau_s$ is a temporal low pass filter with time constant $\tau_s$ and the velocity input reads (*Equation 10* of the main text)

$$\bar{I}_{vel} := v^+/(2\pi) .$$

(56)

*Equation 54* and *Equation 55* show that the rotation-cell voltages are re-scaled and filtered versions of the corresponding HD-cell firing rates with a baseline shift $\bar{I}_{vel}$ that is differentially applied to right and left rotation cells.

From *Equation 53*, we derive

$$d^+(\varphi) := D_0^+\left(\frac{\varphi}{v^+}\right) = D_0^{rec+}\left(\frac{\varphi}{v^+}\right) + D_0^{R+}\left(\frac{\varphi}{v^+}\right) + D_0^{L+}\left(\frac{\varphi}{v^+}\right) + I_{inhib}^{HD} .$$

(57)

Assuming a large number $N_{HD}$ of HD cells evenly spaced around the circle, the recurrent axon-distal input reads

$$D_0^{rec+}\left(\frac{\varphi}{v^+}\right) = \sum_j W_{0j} f[V_j^{a+}(\varphi/v^+)] + I_{inhib}^{HD}$$

(58)

$$= \rho_{HD} \int_0^{2\pi} d\theta\, w(\theta) f\Big[\underbrace{V_0^{a+}\left(\frac{\varphi+\theta}{v^+}\right)}_{=: v^{a+}(\varphi+\theta)}\Big] + I_{inhib}^{HD}$$

(59)

$$= \rho_{HD} [w \star f(v^{a+})](\varphi) + I_{inhib}^{HD} .$$

(60)

where $\rho_{HD} = N_{HD}/2\pi$ is the density of the HD neurons around the circle and we used the fact that the axon-proximal voltage is translation invariant (see also *Equation 37*):

$$V_j^{a+}(\tau) = V_0^{a+}(\tau + \theta/v^+) .$$

(61)

Following a similar procedure for $D_0^{R+}$ and $D_0^{L+}$, we obtain:

$$d^+(\theta) = \Big[\rho_{HD} w \star f(v^{a+}) + \rho_{HR} w^R \star f(v^{R+}) + \rho_{HR} w^L \star f(v^{L+})\Big](\theta) + I_{inhib}^{HD}$$

(62)

where $\rho_{HR} = N_{HR}/2\pi$ is the density of the HR neurons for one particular turning direction (note that we assumed $\rho_{HR} = 2\rho_{HD}$ in the main text). In deriving *Equation 62* we defined

$$v^{R+}(\theta) := V_0^{R+}(t/v^+) \approx \frac{A_{active}}{f_{max}} [h_s^+ * f(v^{a+})](\theta) + \bar{I}_{vel} + I_{inhib}^{HR}$$

(63)

$$v^{L+}(\theta) := V_0^{L+}(t/v^+) \approx \frac{A_{active}}{f_{max}} [h_s^+ * f(v^{a+})](\theta) - \bar{I}_{vel} + I_{inhib}^{HR} .$$

(64)

where we defined the filter $h_s^+(\varphi) := \frac{1}{|v^+|} H_s(t/v^+)$, and the approximations are valid if $H_s(t/v^+) \ll 1$ for $t > 2\pi/v^+$, which holds true for the time constant and velocity distribution assumed in the main text.

Finally, we compute the rotation-cells' weights change. For these weights, the learning rule is the same as the one for the recurrent connections, except that the post-synaptic HD input is replaced by the post-synaptic HR input. Therefore, following the same procedure as in *Equation 38–Equation 46*, the rotation weight changes are given by:

$$\frac{d}{dt}w^{R+}(\theta) = \frac{\eta}{2\pi}\{\epsilon^+ \star [h^+ * f(v^{R+})]\}(\theta) \tag{65}$$

$$\frac{d}{dt}w^{L+}(\theta) = \frac{\eta}{2\pi}\{\epsilon^+ \star [h^+ * f(v^{L+})]\}(\theta). \tag{66}$$

In summary, for clockwise movement, we obtain the following system of equations:

$$\begin{cases}
d^+(\theta) & = [\rho_{HD}w \star f(v^{a+}) + \rho_{HR}w^R \star f(v^{R+}) + \rho_{HR}w^L \star f(v^{L+})](\theta) + I^{HD}_{inhib} \\[2mm]
v^{ss+}(\theta) & = \frac{g_D}{g_D+g_L}(h^+ * d^+)(\theta) \\[2mm]
v^{a+}(\theta) & = v^{ss+}(\theta) + \bar{I}_{vis}(\theta) \\[2mm]
v^{R+}(\theta) & = \frac{A_{active}}{f_{max}}[h^+_s * f(v^{a+})](\theta) + \bar{I}_{vel} + I^{HR}_{inhib} \\[2mm]
v^{L+}(\theta) & = \frac{A_{active}}{f_{max}}[h^+_s * f(v^{a+})](\theta) - \bar{I}_{vel} + I^{HR}_{inhib} \\[2mm]
\epsilon^+(\theta) & = f[v^{a+}(\theta)] - f[v^{ss+}(\theta)] \\[2mm]
\frac{d}{dt}w^+(\theta) & = \frac{\eta}{2\pi}\{\epsilon^+ \star \underbrace{[h^+ * f(v^{a+})]}_{=:\,p^+}\}(\theta) \\[2mm]
\frac{d}{dt}w^{R+}(\theta) & = \frac{\eta}{2\pi}\{\epsilon^+ \star \underbrace{[h^+ * f(v^{R+})]}_{=:\,p^{R+}}\}(\theta) \\[2mm]
\frac{d}{dt}w^{L+}(\theta) & = \frac{\eta}{2\pi}\{\epsilon^+ \star \underbrace{[h^+ * f(v^{L+})]}_{=:\,p^{L+}}\}(\theta). 
\end{cases} \tag{67}$$

## Anti-clockwise movement

We now consider anticlockwise movements with speed $v^- = -v^+$. First we note that the temporal filter

$$h^-(\theta) := \frac{1}{|v^-|}H(\theta/v^-) = \frac{1}{|v^+|}H(-\theta/v^+) = h^+(-\theta) \tag{68}$$

is a mirrored version about the origin of its clockwise counterpart $h^+$, whereas the visual input is unchanged because it is symmetric around the origin (see *Equation 5* of the main text)

$$I^{vis}_0(\theta/v^-) = I^{vis}_0(\theta/v^+). \tag{69}$$

Let us first assume that

$$d^-(\theta) = d^+(-\theta), \tag{70}$$

we shall verify the validity of this assumption self-consistently at the end of this section. From *Equation 68–Equation 70* it follows that $f(v^{a-}) = f\left[\frac{g_D}{g_D+g_L}(h^- * d^-) + \bar{I}^{vis}\right]$ is a mirrored version of $f(v^{a+})$, that is,

$$f[v^{a-}(\theta)] = f[v^{a+}(-\theta)], \tag{71}$$

and, as a result,

$$\epsilon^-(\theta) = \epsilon^+(-\theta). \tag{72}$$

We now compute the anticlockwise weight change for the recurrent weights

$$\frac{\mathrm{d}}{\mathrm{d}t} w^-(\theta) = \frac{\eta}{2\pi} \{\epsilon^- \star [h^- * f(v^{a-})]\}(\theta) . \tag{73}$$

The r.h.s. of *Equation 73*, without the $\eta/(2\pi)$ pre-factor reads:

$$\{\epsilon^- \star [h^- * f(v^{a-})]\}(\theta) = \int_0^{2\pi} \mathrm{d}\tau\, \epsilon^-(\tau) \int_0^{2\pi} \mathrm{d}s\, h^-(s) f[v^{a-}(\tau + \theta - s)] \tag{74}$$

$$= \int_0^{2\pi} \mathrm{d}\tau\, \epsilon^+(-\tau) \int_0^{2\pi} \mathrm{d}s\, h^+(-s) f[v^{a+}(-\tau - \theta + s)] \tag{75}$$

$$= \int_0^{2\pi} \mathrm{d}\tau\, \epsilon^+(\tau) \int_0^{2\pi} \mathrm{d}s\, h^+(s) f[v^{a+}(\tau - \theta - s)] \tag{76}$$

$$= \{\epsilon^+ \star [h^+ * f(v^{a+})]\}(-\theta) \tag{77}$$

where from *Equation 75* to *Equation 76* we used variable substitution. Therefore, the weight change for clockwise movement is the mirrored version around the origin of the weight change for anticlockwise movement:

$$\frac{\mathrm{d}}{\mathrm{d}t} w^-(\theta) = \frac{\mathrm{d}}{\mathrm{d}t} w^+(-\theta) , \tag{78}$$

meaning that, with learning, the recurrent weights develop into an even function:

$$w(\theta) = w(-\theta) . \tag{79}$$

Let us now study the anticlockwise weight change for the rotation weights. The rotation-cell voltages during anticlockwise movement read:

$$v^{R-}(\theta) = \frac{A_{\mathrm{active}}}{f_{\max}} [h_s * f(v^{a-})](\theta) - \bar{I}_{vel} + I_{inhib}^{\mathrm{HR}} \tag{80}$$

$$v^{L-}(\theta) = \frac{A_{\mathrm{active}}}{f_{\max}} [h_s * f(v^{a-})](\theta) + \bar{I}_{vel} + I_{inhib}^{\mathrm{HR}} . \tag{81}$$

Using *Equation 71* in *Equation 80* and *Equation 81* we find

$$v^{L-}(\theta) = v^{R+}(-\theta) . \tag{82}$$

$$v^{L-}(\theta) = v^{R+}(-\theta) . \tag{83}$$

Therefore, applying the same procedure outlined in *Equation 73*–*Equation 77*, to the anticlockwise change in the rotation weights yields

$$\frac{\mathrm{d}}{\mathrm{d}t} w^{R-}(\theta) = \frac{\mathrm{d}}{\mathrm{d}t} w^{L+}(-\theta) \tag{84}$$

$$\frac{\mathrm{d}}{\mathrm{d}t} w^{L-}(\theta) = \frac{\mathrm{d}}{\mathrm{d}t} w^{R+}(-\theta) , \tag{85}$$

meaning that, during learning, the right and left rotation weights develop mirror symmetry:

$$w^R(\theta) = w^L(-\theta) . \tag{86}$$

To verify that our original assumption in *Equation 70* holds, we compute the axon-distal input for anticlockwise movement:

$$d^-(\theta) = [\rho_{\mathrm{HD}} w \star f(v^{a-}) + \rho_{\mathrm{HR}} w^R \star f(v^{R-}) + \rho_{\mathrm{HR}} w^L \star f(v^{L-})](\theta) + I_{inhib}^{\mathrm{HD}} . \tag{87}$$

Using *Equation 71, 79, 80, 81, 86* in *Equation 87*, yields

$$d^-(\theta) = \rho_{\mathrm{HD}} w \star f(v^{a+}) + \rho_{\mathrm{HR}} w^L \star f(v^{L+}) + \rho_{\mathrm{HR}} w^R \star f(v^{R+})(-\theta) + I_{inhib}^{\mathrm{HD}} = d^+(-\theta) . \tag{88}$$

Finally, using *Equation 78, 84, and 85*, the total synaptic weight changes for both clockwise and anticlockwise movement read

$$\begin{cases} \frac{\mathrm{d}}{\mathrm{d}t}w(\theta) & = \frac{\mathrm{d}}{\mathrm{d}t}w^+(\theta) + \frac{\mathrm{d}}{\mathrm{d}t}w^+(-\theta) \\[2ex] \frac{\mathrm{d}}{\mathrm{d}t}w^R(\theta) & = \frac{\mathrm{d}}{\mathrm{d}t}w^{R+}(\theta) + \frac{\mathrm{d}}{\mathrm{d}t}w^{L+}(-\theta) \\[2ex] \frac{\mathrm{d}}{\mathrm{d}t}w^L(\theta) & = \frac{\mathrm{d}}{\mathrm{d}t}w^R(-\theta). \end{cases} \tag{89}$$

## Averaging across speeds

So far, we have only considered head turnings at a fixed speed $v^+$ (clockwise) and $v^- = -v^+$ (anticlockwise). However, in the full model described in the main text, velocities are sampled stochastically from an OU process. This random process generates a half-normal distribution of speeds with spread $\sigma_v/2$ (*Appendix 5—figure 1*, left, see also *Table 1* in the main text). We thus compute the expected weight changes with respect to this speed distribution:

$$\begin{cases} \frac{\mathrm{d}}{\mathrm{d}t}\langle w \rangle_v(\theta) & := \int_0^\infty \mathrm{d}v\, p(v) \frac{\mathrm{d}}{\mathrm{d}t}w_v(\theta) \\[2ex] \frac{\mathrm{d}}{\mathrm{d}t}\langle w^R \rangle_v(\theta) & := \int_0^\infty \mathrm{d}v\, p(v) \frac{\mathrm{d}}{\mathrm{d}t}w_v^R(\theta) \\[2ex] \frac{\mathrm{d}}{\mathrm{d}t}\langle w^L \rangle_v(\theta) & := \int_0^\infty \mathrm{d}v\, p(v) \frac{\mathrm{d}}{\mathrm{d}t}w_v^L(\theta) \end{cases} \tag{90}$$

where $w_v$ is the weight change for speed $|v^+| = |v^-| = v$ and $p(v)$ is an half-normal distribution with spread $\sigma_v/2$.

## Simulation of the reduced model

In this section, we show the dynamics of the reduced model numerically simulated according to *Equation 67, 89, and 90*. Weight changes are computed at discrete time steps and integrated using the forward Euler method. At each time step we compute the weight changes for each speed $v$ (*Equation 67* and *Equation 89*) and we estimate the expected weight change according to *Equation 90*. We then update the weights and proceed to the next step of the simulation. Note that *Equation 67* requires the firing rates of HD and HR cells at the previous time step (recurrent input, first line of *Equation 67*). Therefore, at each time step, we save the HD and HR firing rates for every speed value $v$ and provide them as input to the next iteration of the simulation.

*Appendix 5—figure 2* shows the evolution of the reduced system for 400 time steps, starting from an initial condition where all weights are zero. One can see that from time steps 75–100 the system switches from a linear regime (HD firing rates below saturation, see top panel) to a non-linear regime (saturated HD rates). Such a switch is accompanied by peaks in the average absolute error (third panel from the top). Notably, the rotation weights start developing a structure only after such switch has occurred (see two bottom panels)—a feature that has been observed also in the full model (*Figure 3E* of the main text).

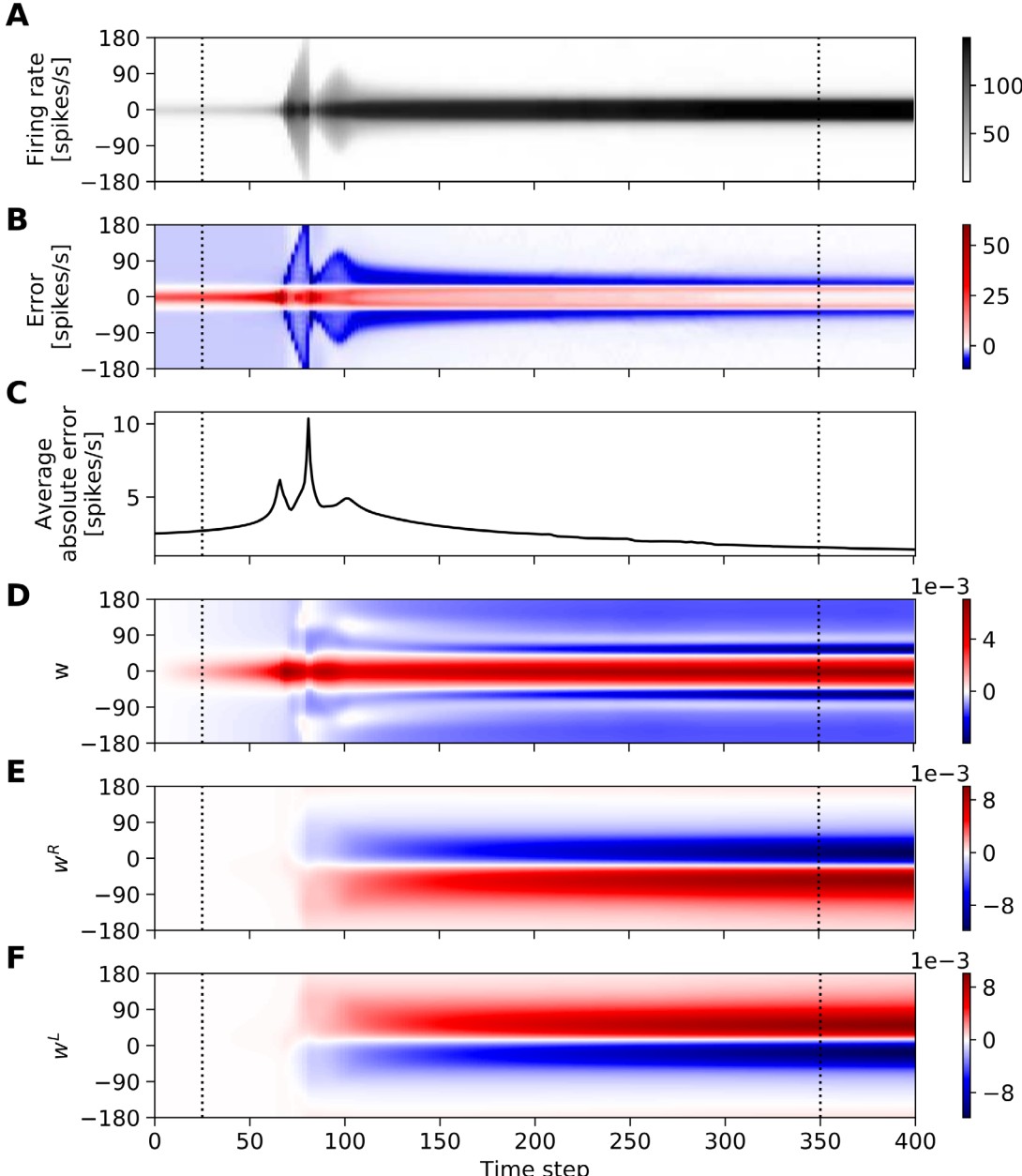

**Appendix 5—figure 2.** Evolution of the reduced model. The figure shows from top to bottom: (**A**) the HD-cells' firing rate $f(v^{a+})$; (**B**) the error $\epsilon$; (**C**) the average absolute error; (**D**) the recurrent weights $w$; (**E–F**) the rotation weights $w^R$ and $w^L$. The HD firing rate and the errors (panels A-C) are averaged across speeds and both movement directions. The vertical dashed lines denote the time points shown in and *Appendix 5—figure 4*.

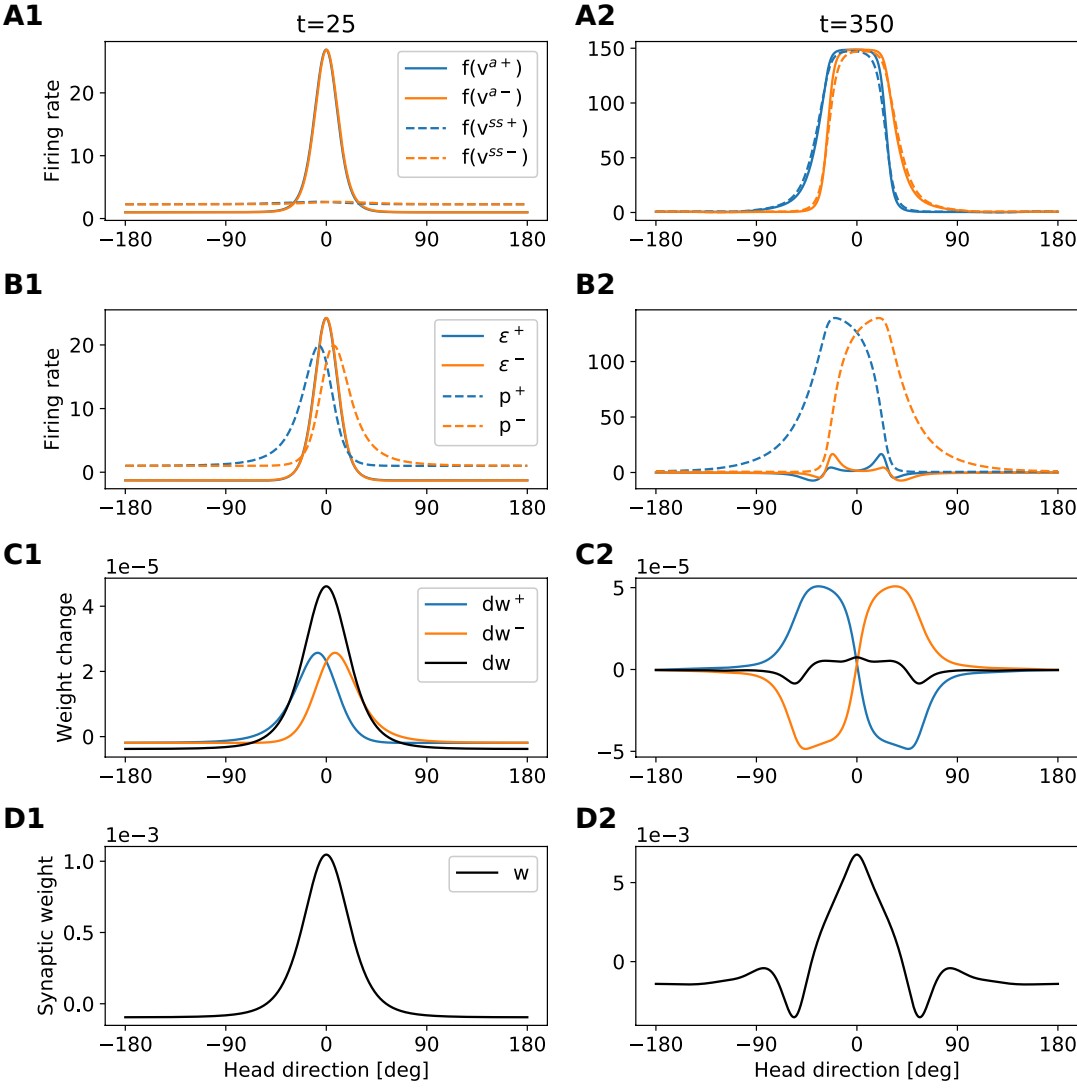

**Appendix 5—figure 3.** Development of the recurrent weights. The figure provides an intuition for the shape of the recurrent-weights profiles that emerge during learning. Each column refers to a different time step (see also dashed lines in *Appendix 5—figure 2*). Each row shows a different set of variables of the model (see legends in the first column). The figure is to be read from top to bottom, because variables in the lower rows are computed from variables in the upper rows. Blue (orange) lines always refer to clockwise (anticlockwise) motion. Black lines in C show the total weight changes for both clockwise and anti-clockwise motion, that is, $\mathrm{d}w = \mathrm{d}w^+ + \mathrm{d}w^-$.

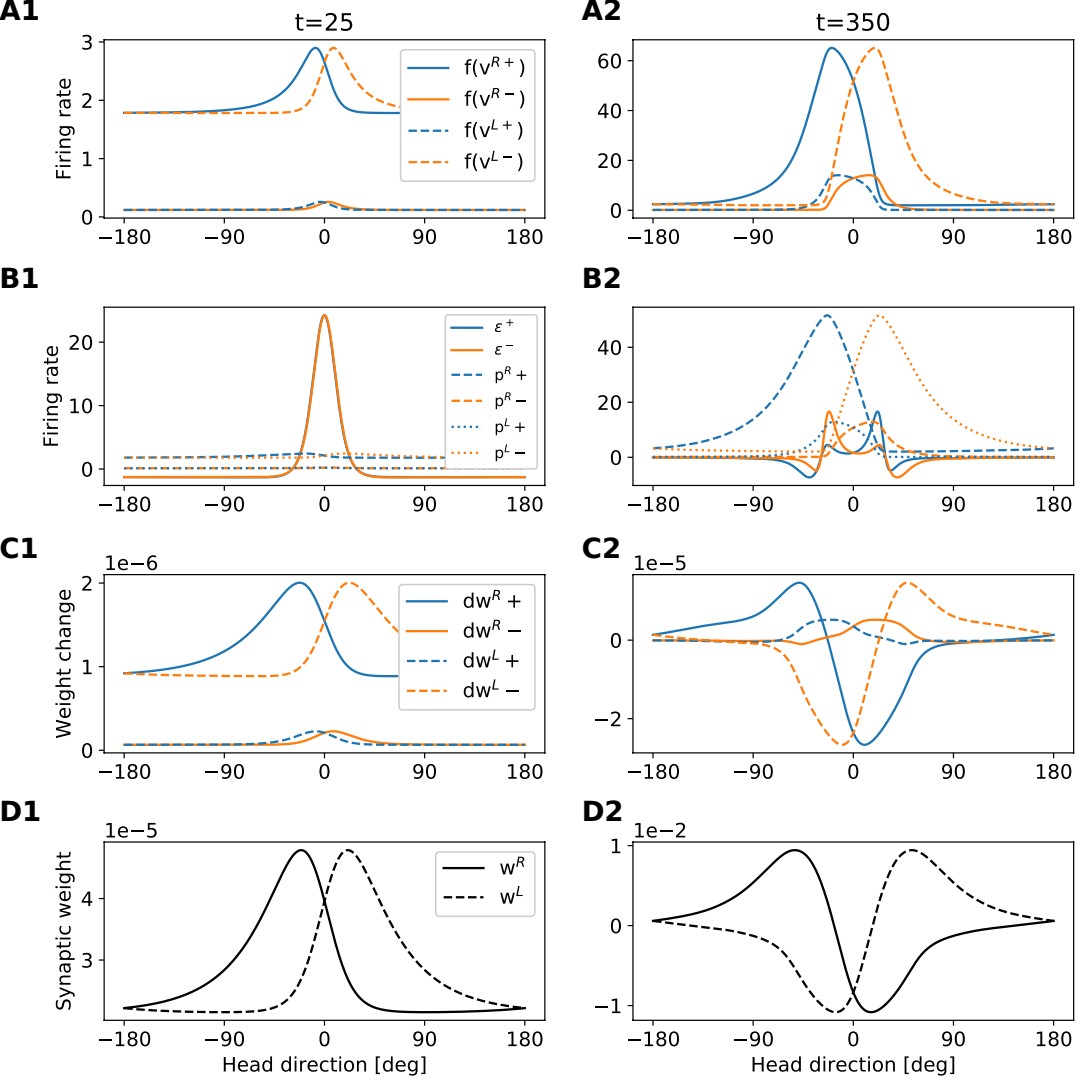

**Appendix 5—figure 4.** Development of the rotation weights. The figure provides an intuition for the shape of the rotation-weights profiles that emerge during learning. Each column refers to a different time step (see also dashed lines in *Appendix 5—figure 2*). Each row shows a different set of variables of the model (see legends in the first column). The figure is to be read from top to bottom, because variables in the lower rows are computed from variables in the upper rows. Blue (orange) lines always refer to clockwise (anticlockwise) motion.

## Development of the recurrent weights

*Appendix 5—figure 3* provides an intuition for the shape of the recurrent-weights profiles that emerge during learning. The first column shows the evolution of the recurrent weights in the linear regime ($t = 25$), that is, before the HD rates reach saturation. In this regime, both recurrent and rotation weights are small, and the steady-state axon-distal rate

$$f(v^{ss}) \approx f\left(\frac{g_D}{g_D+g_L}I_{inhib}\right) \tag{91}$$

is flat and close to zero. Therefore, the HD output rate $f(v^a)$ is dominated by the visual input $\bar{I}_{vis}$ (*Equation 67*, third line), which has the shape of a localized bump (panel A1). Thus the error $\epsilon$ has also the shape of a bump (B1). Additionally, the post-synaptic inputs $p^+$ and $p^-$ are shifted and filtered versions of this bump (*Equation 67*, seventh line). The recurrent weight changes $\mathrm{d}w^+$ and $\mathrm{d}w^-$ for clockwise and anticlockwise movement are given by the cross-correlation of the errors $\epsilon^+$ and $\epsilon^-$ with the post-synaptic inputs $p^+$ and $p^-$ (panel C1; see *Equation 67* seventh line and *Equation 73*). Note that because $a(x) \star b(x) = a(-x) * b(x)$, the operation of cross-correlation can

be understood graphically as a convolution between the mirrored first function $a$ and the second function $b$. Such a mirroring is irrelevant in C1 (linear regime) because the error is an even function, but becomes important in C2 (non-linear regime). As a result of this cross-correlation, the recurrent recurrent-weight changes $\mathrm{d}w^+$ and $\mathrm{d}w^-$ are shifted bumps (colored lines in C1), which merge into a single central bump after summing clockwise and anticlockwise contributions (black line in C1). Therefore, in the linear regime, the recurrent weights develop a single central peak in the origin (panel D1).

The second column of shows the development of the recurrent weights in the non-linear regime (time step 350). Panel A2 shows that in this scenario the HD firing-rate bumps are broader and approach saturation due to the strong recurrent input. The coupling between the axon-distal and axon-proximal compartment acts as a self-amplifying signal during learning which results in the activity of all active neurons participating in the bump reaching saturation. Additionally, because the recurrent input is filtered in time (*Equation 67*, second line), such bumps are also shifted towards the direction of movement. Importantly, due to the lack of visual input, within the receptive field the steady-state axon-distal rates are always smaller than the firing rates. As a result, the errors $\epsilon^+$ and $\epsilon^-$ show small negative bumps in the direction of movement, and small positive bumps in the opposite direction (panel B2). Additionally, the post-synaptic inputs $p^+$ and $p^-$ shift further apart from the origin. Consequently, the total weight change $\mathrm{d}w$ develops negative peaks around 60 deg (black line in C2, contrast to panel C1), and these peaks get imprinted in the final recurrent weights' profiles (panel D2).

## Development of the rotation weights

*Appendix 5—figure 4* provides an intuitive explanation for the shape of the rotation-weights profiles $w^R$ and $w^L$ that emerge during learning. The first column shows the evolution of the rotation weights in the linear regime ($t = 25$), i.e., before the HD rates reach saturation. In this regime, the rotation-cell firing rates are filtered versions of the HD bumps but re-scaled by a factor $A_{\mathrm{active}}/f_{\mathrm{max}} \approx 0.013$ and baseline-shifted by an amount $\pm\bar{I}_{vel} + I^{\mathrm{HR}}_{inhib}$ (*Equation 67* lines 4 and 5; panel A1, compare to panel A1). This baseline shift acts as a switch that determines from which rotation cells population connections will be mainly drawn from, depending on the direction of motion. Panel B1 shows that the errors $\epsilon^+$ and $\epsilon^-$ overlap and have the shape of a bump centered at the origin (same curves as in panel B1). Additionally, the post-synaptic potentials $p^{R\pm}$ and $p^{L\pm}$ in B1 are filtered versions of the curves in A1 (*Equation 67*, lines 7 and 8). As a result, the weight changes $\mathrm{d}w^{R\pm}$ and $\mathrm{d}w^{L\pm}$, that is, the errors cross-correlated by the post-synaptic potentials, appear similar to the bumps in A1, but they are smoother and further apart from the origin (panel C1). Finally, such weight changes get imprinted in the rotation weights (panel D1).

The second column shows the evolution of the rotation weights in the non-linear regime ($t = 350$), that is, after the HD rates reach saturation. In this case, the large recurrent input gives rise to larger rotation rates (A2, compare to A1) and larger post-synaptic potentials (B2, compare to B1). In panel B2, we can see that the errors $\epsilon^+$ and $\epsilon^-$ show positive and negatives peaks shifted from the origin (same curves as in panel B2), which generate weight changes with both positive and negative lobes (panel C2). Such weight changes get finally imprinted in the rotation weights (panel D2).

