## [Editor Report]

This paper will be of interest to neuroscientists studying the navigation system, and in particular, those who study the ability of animals to path integrate. This study proposes an elegant synaptic plasticity rule that maintains the connectivity required for path integration by integrating visual and self-motion input arriving at different dendritic locations in a neuron. This idea is applied to the central complex of *Drosophila*, a well-characterized experimental system.

---

## [Decision Letter]

**Decision letter after peer review:**

Thank you for submitting your article "Learning accurate path integration in ring attractor models of the head direction system" for consideration by *eLife*. Your article has been reviewed by 3 peer reviewers, and the evaluation has been overseen by a Reviewing Editor and Ronald Calabrese as the Senior Editor. The following individuals involved in review of your submission have agreed to reveal their identity: Hervé Rouault (Reviewer #4).

The reviewers have discussed their reviews with one another, and believe the manuscript is potentially suitable for publication, provided a number of comments are addressed or discussed. The Reviewing Editor has drafted a merged feedback to help you prepare a revised submission.

Essential revisions:

Plasticity:

– A main question is whether the type of synaptic plasticity proposed by the authors is necessary and operative in the CX. One motivation for the plasticity rule is that, as the authors state, ring attractors "require that synaptic connections are precisely tuned (Hahnloser, 2003). Therefore, if the circuit was completely hardwired, the amount of information that an organism would need to genetically encode connection strenghts would be exceedingly high." However, in Figure 3 and S3, it is shown that the learnedPI "far outperforms" flies, so the authors add random noise to the synaptic weights in order to make a more realistic model. But this means that the synaptic weights don't actually need to be that precise. Doesn't this mean that the connectivity may in fact be genetically encoded?

– Another proposed function of the plasticity rule is in adjusting to changes in relative gain between proprioceptive signals. The authors cite Jayakumar et al. (2019) in the introduction as motivation for this, without noting until later that the study used rodents rather than flies. In the discussion, it is mentioned that Seelig and Jayaraman (2015) actually did not find evidence for gain adjustments in flies. This seems to go against the plasticity mechanism being proposed as being relevant for flies. Is there any evidence for this in flies?

– Biologically plausible learning rule: One of the main assumptions that allows the authors to claim for a biologically plausible learning rule is that the 'firing rate' of the neuron is available for all synapses at the axon-distal compartment due to the assumed back-propagation (line 167). As this seems to be crucial, it will be good to give references showing that there is a back-propagation AP in E-PG neurons, or to explain what the alternatives are.

– Writing style: The paper would be easier to understand if some of the key relevant equations, for example those that describe the synaptic plasticity rule, were included in the main text.

– Analysis of connectome data. The connectome data is a nice support for why using a learning rule that relies on a two-compartment model. However, some points could be clarified. What is exactly is X,Y position? Can the authors plot the shape of the neuron on the same plot, and maybe also to give access to a 3D video of this cloud points? Also, why showing only 1 example, instead of analyzing all the EPG neurons and showing all of them in a supplementary figure. Then, the authors should do a statistical analysis to support their claim, such as doing some clustering analysis. You want to at least convince the reader that you can reject the null hypothesis that inputs are not segregated.

– Predictions: It would be good if the authors could try and give more concrete predictions, or better, suggest testable predictions. While a theory doesn't have to give predictions, as mentioned in the public review, in this specific case it would be good if the authors could spend one or two paragraphs on concrete predictions and maybe even suggest new experiments to the community. In other words, while not necessary for publishing the paper, the authors could go beyond saying that PI requires supervised learning during development, which seems more like a hypothesis of the model than a real prediction.

– Network initial architecture: It is not clear how the results depend on the specific initialization of the network and how sensitive the main conclusions of the paper are to these choices. More specifically:

– It is not completely clear what the initial recurrent connectivity in the network is, and how the results depend on this choice. For example, does the symmetry, which the authors rely upon to derive their reduced model, persist if the initial connectivity is random?

– Would the results and conclusions of the paper change if the authors use a different choice of the HD->HR connections? Could they be random? Could they be connected to more than one neuron? Could they be plastic as well?

Related to this issue, in line 266 the authors claim that 'circular symmetry is a crucial property for any ring attractor'. While this might be a common knowledge in the field, please see Darshan and Rivkind 21 that argue against it. Following this, it is unclear if symmetry is needed for the authors to derive the reduced model, or if it is a principle that they think the system must have. The former is rather technical, while the latter is an important principle. It would be good if the authors could address this in their discussion.

– Saturation: It seems that saturation is crucial in this work. Is it known that E-PG neurons reach saturation levels? For example, in cortex this rarely happens, where neurons are usually active around an expensive non-linearity regime and are far from saturation. It will be beneficial if the authors could show that similar phenomena hold with different types of non-linearities, in which neurons are not saturated. Alternatively, the authors can explain in the manuscript why they think that E-PG neurons in reality are saturated (maybe give references?).

– Line 221 (and 389) "…our work reproduces this feature of the fly HD as an emergent property of learning". It is not clear what the authors mean by this. Is the impairment of PI in small angular velocities a result of the finite number of neurons in the network, or an emergent property of the learning? For example, what would happen if the authors would take 2000 neurons, will they still see this 'emergent property'? This also goes against what is written in Line 465 of the discussion.

– Timescales: there are many timescales in the model. Although it is addressed, to some extent, in the supplementary material, it would be good to add a paragraph stating what are the requirements on these timescales. For example, does the model fail if one of the timescales is smaller\larger than the others? Did the authors assume in the derivation that one timescale is slower than the rest? Why tau_δ_ can be ignored in the derivation? Can the authors show the failure modes of the network\derivation when changing the timescale?

– Randomness in network connectivity- Figure S3. In case of the random connectivity, as the network is so small, the behavior of the network depends on the specific network realization and, therefore, it is hard to assess how representative FigS3D-E are. it will be good to show some statistical analysis of the diffusion across these networks.

– Noise in the dynamics- Figure S4. It is unclear if the diffusion is larger in this case just because the authors added noise, which stays after training, or it is something deeper than that. Could the authors compare the diffusivity in networks that were trained with and without noise? Also, in FigS4 panel E- it seems that in contrast to the finite-size network effects, in this case there is no problem to integrate at very small velocities. Why is that? Does the noise contribute to smoothing the barriers and helps to go from a discrete to a more continuous representation when doing PI? Does this have to do with the small number of involved neurons in the fly central complex? One of the reviewers disagrees with the remark about a feature build-in by hand in previous models l220-222. This is not something the learning rule per se allows to understand.

– What is the limitation of the adaptation in the network? For example, the authors claim that 'we test whether our network can rewire to learn an arbitrary gain between the two', but in fact only showed examples for a 50% increase or decrease in gain and only mentioned a negative gain without showing it. It will be useful to have a figure, showing how the performance varies with changing the gain. Is there a maximal\minimal gain that above\below it the network always fail to perform? Which of the network parameters are important for these gain changes?

– The authors seem to assimilate path integration with angular integration (l31-40). The way insects manage to perform path integration (i.e. integrate distances) is still undetermined to our knowledge. Clearly, the angular integration performed in the central complex is not enough. The authors should try to disambiguate this point.

– The authors mention the existence of two PEN populations (PEN-1 and PEN-2). Could the proposed model could shed light on the presence of these two populations. Would they account for the inhibitory and excitatory part of the HR curves on Figure 3C?

– On l 625, the authors state that the constant inhibitory drive contributes to the uniqueness of the bump. One of the reviewers was not convinced of this statement as this constant inhibition does not play a role in the competition between several bumps. Only the inhibitory recurrent connectivity within the HD population or going through the HR populations would play such a role.

– In Figure S5B, the bump velocity goes to zero around 500º/s. Could this come from a defect in the network that would, for instance, abolish the presence of a bump in left and right HR populations? In a more robust behavior, only one of the left or right bumps would disappear, and the HD bump would go at a max constant velocity.

[Editors' note: further revisions were suggested prior to acceptance, as described below.]

Thank you for resubmitting your work entitled "Learning accurate path integration in ring attractor models of the head direction system" for further consideration by *eLife*. Your revised article has been evaluated by Ronald Calabrese (Senior Editor) and a Reviewing Editor (Srdjan Ostojic).

The manuscript has been much improved but there are some remaining issues related to the lack of evidence for plasticity in CX. After consultation, the reviewers agreed that the current lack of evidence for plasticity should be pointed out earlier in the manuscript, and suggested presenting the need for plasticity in younger flies as a prediction of the model. One of the reviewers provided additional suggestions listed below.

*Reviewer #2 (Recommendations for the authors):*

The authors have made a number of improvements to the manuscript. My central concern, which is that there isn't any evidence for this plasticity being present in the CX, still stands. Obviously, a purely modeling study is unable to address this concern. The authors have argued in Appendix 3 that asymmetries in the architecture can severely disrupt the performance of the model and that this supports the need for plasticity. Here are a few suggestions related to this central issue.

1) The authors refer to the performance of flies from Seelig and Jayaraman (2015), stating at various points whether their models outperform the actual performance or not. This should actually be quantified in the figures and/or text. Specifically, both Figure 2 and Appendix 3 would benefit from a quantitative comparison to real fly performance.

2) It would be useful to show how much imprecision must be added to the synaptic connections before the model can account for the performance of actual flies. This would answer the question of "how much precision must be genetically encoded to account for fly behavior."

*Reviewer #3 (Recommendations for the authors):*

I find the paper to be timely and very interesting. The paper was well presented already in the first version, but I did have a few concerns. The authors did a great job in addressing all of these concerns and comments in the new version of their manuscript.

I strongly recommend it for publications in *eLife*.

---

## [Author Response]

Essential revisions:Plasticity:– A main question is whether the type of synaptic plasticity proposed by the authors is necessary and operative in the CX. One motivation for the plasticity rule is that, as the authors state, ring attractors "require that synaptic connections are precisely tuned (Hahnloser, 2003). Therefore, if the circuit was completely hardwired, the amount of information that an organism would need to genetically encode connection strengths would be exceedingly high." However, in Figure 3 and S3, it is shown that the learnedPI "far outperforms" flies, so the authors add random noise to the synaptic weights in order to make a more realistic model. But this means that the synaptic weights don't actually need to be that precise. Doesn't this mean that the connectivity may in fact be genetically encoded?

This is indeed an important point to expand on, and we thank the reviewer for bringing it up. A main finding of the present work is that a local synaptic plasticity rule operating in compartmentalized neurons can learn the precise synaptic connectivity required for accurate PI. Because simulation conditions are idealized,PI performance of the model can exceed that of the fly. However, there are many unknown factors that could limit the performance in the biological system, factors that, for simplicity, we do not model explicitly. Adding noise to the connectivity (as in Figure 3 —figure supplement 3; was ‘S3’ in the first submission) is a simple way to bring the model’s performance closer to the one of the fly; not an explanation of why the fly’s performance is worse. Figure 3 —figure supplement 3 thus shows how a perturbation of the learned network results in more realisticPI performance, deviating from the idealized simulation conditions.

In reality, there could be multiple reasons whyPI performance is worse in the fly than in the model. For instance, a confounder that would affect performance but not necessarily learning could be the presence of inputs that are unrelated to path integration, e.g., inputs related to circadian cycles (Raccuglia et al., 2019). In the presence of such confounders, a precise tuning of the weights might be crucial in order to reach the performance of the fly. In other words, only if the model outperforms the biological circuit in a simplified setting, it has a chance to perform as well in a realistic setting, with all the additional complexities the latter comes with. We have added these remarks in the final paragraph of “Relation to experimental literature” in the Discussion.

A further reason why synaptic plasticity might be necessary to achieve a PI performance comparable to the fly is to counteract asymmetries in the initial architecture, such as noisy hardwired connectivity: In the main text we assume that the architecture and the hardwired HD to HR connections are completely circular symmetric. However, as the reviewers have also pointed out in other comments below, this is unrealistic for a biological system. In the new Appendix 3 we show that if we add variability to these connections while maintaining the same profiles for HR to HD and recurrent connections, PI in the model is much worse than in the fly (Appendix 3 – Figure 2). This implies that even if the optimal weight connectivity was to be passed down genetically, it would still be impossible to path integrate as well as the fly, because minimal biological asymmetries in the circuit can have a big impact on PI. Instead, as we show in Appendix 3 – Figure 1, our learning rule can counteract asymmetries in the initial architecture by producing asymmetries in the learned connectivity, andPI remains excellent. Therefore the PI performance of the fly can be matched only if there is plasticity during development.

Finally, we note that since this is a modeling study, we cannot prove that plasticity is operative in the CX; instead, this is a prediction of our model that can only be tested with experiments.

– Another proposed function of the plasticity rule is in adjusting to changes in relative gain between proprioceptive signals. The authors cite Jayakumar et al. (2019) in the introduction as motivation for this, without noting until later that the study used rodents rather than flies. In the discussion, it is mentioned that Seelig and Jayaraman (2015) actually did not find evidence for gain adjustments in flies. This seems to go against the plasticity mechanism being proposed as being relevant for flies. Is there any evidence for this in flies?

To the best of our knowledge, there is no evidence of plasticity being used for gain adjustments in flies. However, it has not been tested in young animals, and this is why we propose it as a testable prediction. As noted in the Discussion (section “Relation to experimental literature”, third paragraph), in the Seelig and Jayaraman (2015) study, the experimenters used mature flies, whereas we propose that, in flies, this form of plasticity is strong only during development. In rodents, on the other hand, there is evidence that such plasticity ensues past the developmental stage.

To better distinguish properties ofPI in rodents and flies, we mention now explicitly in the Introduction that Jayakumar et al. (2019) used rodents. Furthermore we suggest (in the last paragraph of the Introduction and last paragraph of section “Testable predictions” in the Discussion) that our prediction could be tested in young flies.

– Biologically plausible learning rule: One of the main assumptions that allows the authors to claim for a biologically plausible learning rule is that the 'firing rate' of the neuron is available for all synapses at the axon-distal compartment due to the assumed back-propagation (line 167). As this seems to be crucial, it will be good to give references showing that there is a back-propagation AP in E-PG neurons, or to explain what the alternatives are.

Backpropagation of APs could occur with either active or passive mechanisms. In our setting, passive backpropagation would suffice, and passive spread of activity has been shown to not attenuate too fast in fly neurons (Gouwens and Wilson, 2009). The axon-proximal and axon-distal compartments belong to the same dendritic tuft, and we assume that the axon initial segment is close to the axon-proximal compartment. Thus, the generated AP would only need to travel a short distance compared to the electrotonic length, hence it would not be attenuated considerably (e.g. see figure 5 in Gouwens and Wilson, 2009). We have added these remarks in the second paragraph of “Relation to experimental literature” in the Discussion. Finally, we note that to the best of our knowledge, there is no evidence for active backpropagation of APs in E-PG neurons.

– Writing style: The paper would be easier to understand if some of the key relevant equations, for example those that describe the synaptic plasticity rule, were included in the main text.

The learning rule is the key equation, as it contains the most important variables of the model. In this revised manuscript, we have added a simplified version (i.e. without low-pass filtering the weight changes) of the learning rule reported in section “Learning rule” in Methods (see second to last paragraph of the section “Model setup” in Results).

– Analysis of connectome data. The connectome data is a nice support for why using a learning rule that relies on a two-compartment model. However, some points could be clarified. What is exactly is X,Y position? Can the authors plot the shape of the neuron on the same plot, and maybe also to give access to a 3D video of this cloud points? Also, why showing only 1 example, instead of analyzing all the EPG neurons and showing all of them in a supplementary figure. Then, the authors should do a statistical analysis to support their claim, such as doing some clustering analysis. You want to at least convince the reader that you can reject the null hypothesis that inputs are not segregated.

To clarify E-PG connectivity at the EB, we provide new figures and analysis in accordance with the reviewers’ remarks. First, we include the skeleton plot of the example EP-G neuron in Figure 1E, using coordinates Y and Z in line with the fly connectome. The orientation of the skeleton plot is the same as in the synapse location plot, whose placement is indicated by the zoom box. To avoid congestion, we decided to not plot the shape of the neuron on top of the synapse location plot; the shape of the neuron can be readily seen in the skeleton plot. In addition, we provide a 3D rotating video of the point cloud for this example neuron (Figure 1 – video 1).

Following the reviewer's suggestions, we plot the synapse locations for all 16 neurons we analyzed in the new Figure 1 —figure supplement 1A. Furthermore, to support the claim that visual inputs are separated from recurrent and HR to HD inputs, we perform binary classification between the two classes (R2 and R4d vs. P-EN1 and P-EN2), using SVMs and 5-fold cross validation (for details, see the last paragraph in “Fly Connectome Analysis” in Methods). We find (Figure 1 —figure supplement 1B) that, in held-out test data, the model is excellent (test accuracy > 0.95 across neurons and model runs) at predicting class identity from location alone. We report this finding in paragraph 3 of section “Model Setup” of the results.

– Predictions: It would be good if the authors could try and give more concrete predictions, or better, suggest testable predictions. While a theory doesn't have to give predictions, as mentioned in the public review, in this specific case it would be good if the authors could spend one or two paragraphs on concrete predictions and maybe even suggest new experiments to the community. In other words, while not necessary for publishing the paper, the authors could go beyond saying thatPI requires supervised learning during development, which seems more like a hypothesis of the model than a real prediction.

We agree with the reviewers that it is very important to suggest testable predictions. Therefore we have included a new section “Testable predictions” in the Discussion, where we delineate our model predictions and put forth ways to test them. Briefly, these predictions are:

Synaptic plasticity during development is crucial in setting up the HD systemHD neurons have a compartmentalized structure where idiothetic inputs are separated from allothetic inputsAllothetic inputs more readily control the firing rate of the HD neuronsPassive or active backpropagation of action potentials makes the output of the neuron available to the compartment that receives idiothetic inputsThe HD system can readily adapt to manipulations of gain during development

– Network initial architecture: It is not clear how the results depend on the specific initialization of the network and how sensitive the main conclusions of the paper are to these choices. More specifically:– It is not completely clear what the initial recurrent connectivity in the network is, and how the results depend on this choice. For example, does the symmetry, which the authors rely upon to derive their reduced model, persist if the initial connectivity is random?

The reviewers correctly point out that this information was missing from the manuscript. We now describe network initialization in the first paragraph of “Training protocol” in Methods. Briefly, trainable weights for all networks are initialized with random connectivity drawn from a normal distribution with mean 0 and standard deviation 1/ sqrt(N^HD^), where N^HD^ is the number of HD neurons in the network, as is common practice in the modeling literature.

Since this initialization results in weights that are much smaller that the final weights, to better address the question on the dependence of the results on initial conditions, we randomly shuffle the learned weights in Figure 3A,B to completely eradicate any structure and initialize the network with the shuffled weights. Author response image 1 shows that even in this scenario, learning converges to a low learning error, and PI performance is excellent, virtually indistinguishable from the one of the main text network (see Figure 2). We note however that learning is not able to completely eradicate the noise in the initial weights. Instead, it settles to a noisier connectivity and weights span a larger range (Figure 3A,B to Author response image 1). Despite that, PI remains accurate (see Author response image 1), therefore the deviations caused by individual weights should be balanced out when network activity as a whole is considered.

**Author response image 1. sa2fig1:** PI performance of a network that was initialized with randomly shuffled weights from Fig. 3A,B. (A), (B) Resulting weight matrices after ~22 hours of training. The weights matrices look very similar to the ones in the main text in Fig. 3A,B, albeit connectivity remains noisy and weights span a larger range. (C) Example of PI shows that the network can still path-integrate accurately. (D) Temporal evolution of distribution of PI errors during PI in darkness. Performance is comparable to Fig. 2B of the main network, with only minimal side bias. (E) PI performance is comparable to the one in Fig. 2C.

We now mention this also in the first paragraph of “Training protocol” in Methods (together with a hint on simulations in Figure 4 that illustrate gain changes), concluding that the final PI performance is virtually independent of the initial distribution of weights.

– Would the results and conclusions of the paper change if the authors use a different choice of the HD->HR connections? Could they be random? Could they be connected to more than one neuron? Could they be plastic as well?

To address these very relevant questions, we include the new Appendix 3; furthermore, we added in the section “Model Setup” (fourth paragraph) in the Results a brief statement that our assumption of “1-to-1 wiring and constant amplitude of the HD to HR connections” is uncritical. Briefly, in new simulations we drop the 1-to-1 and constant-magnitude assumptions of the HD to HR connections, and we find that PI performance is not impaired (Appendix 3 – Figure 1). The network can even path integrate when these connections are completely random, albeit for a smaller angular velocity range (Appendix 3 – Figure 3). Finally, we demonstrate that since circular symmetry in the hardwired HD-to-HR connections is unreasonable for a biological system, the learning rule is crucial in balancing out any deviations from circular symmetry in the initial architecture; otherwise, PI performance would be considerably impaired (Appendix 3 – Figure 2, also see our answer to the 1st reviewer comment above about the necessity of synaptic plasticity).

Regarding the possibility of the HD-to-HR connections being plastic, we note that our model does not incorporate plasticity for connections other than the incoming connections to associative neurons; therefore the HD-to-HR connections cannot be learned with our model. We assume that they could be prewired, e.g. during prenatal circuit assembly. If some other form of plasticity operated in these synapses, the network should still learn accurate PI, provided that HD-to-HR connections settle to final values after some time. This is supported by the fact that, as mentioned in the previous paragraph, the network learns accurate PI even for completely random HD-to-HR connectivity.

Related to this issue, in line 266 the authors claim that 'circular symmetry is a crucial property for any ring attractor'. While this might be a common knowledge in the field, please see Darshan and Rivkind 21 that argue against it. Following this, it is unclear if symmetry is needed for the authors to derive the reduced model, or if it is a principle that they think the system must have. The former is rather technical, while the latter is an important principle. It would be good if the authors could address this in their discussion.

The reviewers are correct to point out that circular symmetry is not required for building a ring attractor in general. Therefore we have corrected the statement (first paragraph of section “Learning results in synaptic connectivity that matches the one in the fly ” in the Results) from “any ring attractor” to “a symmetric ring attractor”, and included a discussion of the suggested paper in the section “Relation to theoretical literature” in the Discussion. It follows that the results in our Mathematical Appendix only hold for the case where the symmetries in the initial learning setup result in a circular symmetric connectivity. Therefore, anatomical symmetry is an assumption of our reduced model, but not a requirement for all ring attractors.

– Saturation: It seems that saturation is crucial in this work. Is it known that E-PG neurons reach saturation levels? For example, in cortex this rarely happens, where neurons are usually active around an expensive non-linearity regime and are far from saturation. It will be beneficial if the authors could show that similar phenomena hold with different types of non-linearities, in which neurons are not saturated. Alternatively, the authors can explain in the manuscript why they think that E-PG neurons in reality are saturated (maybe give references?).

From calcium imaging videos in the original study of Seelig and Jayaraman (2015) and later ones, it seems that E-PG neurons within the bump fire vigorously at similar levels, whereas neurons outside the bump fire very sparsely. Therefore the shape of the bump in the Seelig and Jayaraman (2015) experiments seems to not be very far from square, which might hint at saturation. Even though, to the best of our knowledge, it is currently not known whether E-PG neurons actually reach saturation, other *Drosophila* neurons are known to reach saturation with increasing inputs, instead of some sort of depolarization block (Wilson, 2013; Brandao et al., 2021). Saturation with increasing inputs may come about due to, for instance, short-term synaptic depression: it has been reported that beyond a certain frequency of incoming action potentials, the synaptic input current is almost independent of that frequency (Tsodyks and Markram, 1997; Tsodyks et al., 1998). We now better explain our assumptions underlying saturation in the section “Neuronal model” in the Methods.

– Line 221 (and 389) "…our work reproduces this feature of the fly HD as an emergent property of learning". It is not clear what the authors mean by this. Is the impairment ofPI in small angular velocities a result of the finite number of neurons in the network, or an emergent property of the learning? For example, what would happen if the authors would take 2000 neurons, will they still see this 'emergent property'? This also goes against what is written in Line 465 of the discussion.

The reviewers are right to point this out. Following their suggestion, in Author response image 2 we increase the number of neurons twofold and fourfold and we provide the velocity-gain plots for these simulations. In both cases, the flat segment at small velocities has completely disappeared. We further confirm this by only looking at small velocities of +- 20 deg/s, and reducing the velocity interval to 0.5 deg/s. Therefore, as suggested by the reviewer, the impairment is indeed caused by the limited number of neurons that the fly has at its disposal, and it is not an emergent property from learning. Hence we have corrected all statements regarding the emergent property. Additionally, we now mention that the flat region disappears if we increase the number of neurons (see final paragraph of the section “Mature network can path-integrate in darkness”).

**Author response image 2. sa2fig2:** PI performance for networks with more neurons. (A) For N^HD^ = N^HR^ = 120 and training time ~4.5 hours, PI performance is excellent, and the flat area for small angular velocities observed in Figure 2C is no longer present. (B) To confirm that small angular velocities are no longer impaired, we limit the range of tested velocities and reduce the interval between tested velocities to 0.5 deg/s. (C), (D) same as (A), (B), for N^HD^ = N^HR^ = 240 and training time ~2 hours.

– Timescales: there are many timescales in the model. Although it is addressed, to some extent, in the supplementary material, it would be good to add a paragraph stating what are the requirements on these timescales. For example, does the model fail if one of the timescales is smaller\larger than the others? Did the authors assume in the derivation that one timescale is slower than the rest? Why tau_δ_ can be ignored in the derivation? Can the authors show the failure modes of the network\derivation when changing the timescale?

There are indeed many time scales in the model, which are briefly summarized here:

synaptic time constant τ_s_ = 65 msmembrane time constant of axon-distal compartment τ_l_ = 10 msmembrane time constant of axon-proximal compartment, C/g_L_ = 1 msweight update filtering time constant τ_δ_ = 100 mstime constant of velocity decay τ_v_ = 0.5 slearning time scale (set by 1/η = 20 s)

These time constants are now summarized in the Table 1 in the new Appendix 4. Several of these parameters are well constrained by biology, and thus we chose to keep them constant, while others have quite liberal requirements. We discuss all these in detail in the new Appendix 4.

– Randomness in network connectivity- Figure S3. In case of the random connectivity, as the network is so small, the behavior of the network depends on the specific network realization and, therefore, it is hard to assess how representative FigS3D-E are. it will be good to show some statistical analysis of the diffusion across these networks.

The reviewers are correct to point out that variability exists between different realizations of networks with added noise in the learned weights (now shown in Figure 3 —figure supplement 3, which is identical to the old Figure S3). A useful quantity to characterize the performance of a network is the diffusion coefficient during path integration, which quantifies how fast the width of the PI error distribution in e.g. Figure 3 —figure supplement 3D increases (see new section “Diffusion Coefficient” in Methods). Therefore, we created multiple networks with perturbed weights, and estimated the diffusion coefficient for all of them. We report the point estimate and 95 % confidence intervals (Students t-Test) of the grand average to be 82.3 +- 15.7 deg^2/s, which is considerably larger than the diffusion coefficient for networks without a perturbation in the weights (24.5 deg^2/s, see Appendix 1 – Figure 1E). We report these results in the section “Learning results in synaptic connectivity that matches the one in the fly” (last paragraph).

– Noise in the dynamics- Figure S4. It is unclear if the diffusion is larger in this case just because the authors added noise, which stays after training, or it is something deeper than that. Could the authors compare the diffusivity in networks that were trained with and without noise? Also, in FigS4 panel E- it seems that in contrast to the finite-size network effects, in this case there is no problem to integrate at very small velocities. Why is that? Does the noise contribute to smoothing the barriers and helps to go from a discrete to a more continuous representation when doing PI? Does this have to do with the small number of involved neurons in the fly central complex? One of the reviewers disagrees with the remark about a feature build-in by hand in previous models l220-222. This is not something the learning rule per se allows to understand.

In the previous comment we addressed the diffusivity of path integration in networks that receive velocity input. To evaluate whether synaptic input noise during training affects learning, we now systematically explore the impact of this “training noise” on diffusivity for various levels of synaptic input noise during testing (called “test noise”). However, we find that the contribution ofPI errors to the diffusion coefficient is always greater than the one of test noise, even though thePI errors themselves are quite small. Hence, to assess robustness to noise, we compute the diffusion coefficient in networks initialized at various locations and left to diffuse without any velocity or visual input (see the new Appendix 1). We then estimate and plot in panel E of a revised Appendix 1-Figure 1 (was labeled Figure S4 in the previous version) the diffusion coefficient for networks that have been trained without input noise (σ_n_ = 0 or SNR = 0, blue dots) and with input noise (σ_n_ = 0.7 or SNR = 2; orange dots), for various test noise levels. Briefly, we find that the network trained with noise is less diffusive for test noise levels above σ_n_ = 0.7 (which is equal to the amount of train noise it has been trained on). We report these findings in the last two paragraphs of Appendix 1.

The lack of “stickiness” at small angular velocities is indeed an effect that can be attributed to noise applied during testing, as suggested by the reviewer. This noise allows random transitions to adjacent attractor states. Furthermore, these random transitions are biased towards the side of the drift velocity input, even for small velocity inputs. Therefore, the flat region for small angular velocities seen before disappears, and the bump moves with the drift speed on average. Indeed the problem with integration of small velocities is related to the small number of neurons, as previously noted, and should not be considered a property that results from learning.

– What is the limitation of the adaptation in the network? For example, the authors claim that 'we test whether our network can rewire to learn an arbitrary gain between the two', but in fact only showed examples for a 50% increase or decrease in gain and only mentioned a negative gain without showing it. It will be useful to have a figure, showing how the performance varies with changing the gain. Is there a maximal\minimal gain that above\below it the network always fail to perform? Which of the network parameters are important for these gain changes?

To address the reviewer’s comment, we now test the ability of the network to rewire for a larger set of gain values. Specifically, we include 2 more gains in Figure 4, and include the new Figure 4 —figure supplement 1, where we show an even broader range of tested gains from ⅛ to 4.5 (Figure 4 —figure supplement 1A), along with extreme examples (Figure 4 —figure supplement 1B,C) to show the limits of the system. In Figure 4 —figure supplement 1D-F we also include simulation results where the network was instructed to reverse its gain. We discuss all of these new results in the last paragraph of “Fast adaptation of neural velocity gain” in the Results. A metric to assess deviation from an instructed gain, as well as the relevant quantities setting the limits for gain adaptation are described in the caption of Figure 4 —figure supplement 1.

– The authors seem to assimilate path integration with angular integration (l31-40). The way insects manage to perform path integration (i.e. integrate distances) is still undetermined to our knowledge. Clearly, the angular integration performed in the central complex is not enough. The authors should try to disambiguate this point.

We thank you for pointing out this imprecision. We now disambiguate between path integration and angular integration in the first paragraph of the Introduction.

– The authors mention the existence of two PEN populations (PEN-1 and PEN-2). Could the proposed model could shed light on the presence of these two populations. Would they account for the inhibitory and excitatory part of the HR curves on Figure 3C?

As indicated in Figure 1E and also noted in the first paragraph of “Fly Connectome Analysis” in Methods, PEN-1 neurons correspond to HR cells, and PEN-2 cells take part in the excitatory recurrent loop of HD cells. Furthermore, as noted in the next-to last paragraph in section “Relation to experimental literature” in the Discussion, the PEN-1 neurons account for the excitatory part of the learned HR-to-HD connectivity, whereas the negative part could be mediated by other neurons, or interactions between different neurons altogether.

– On l 625, the authors state that the constant inhibitory drive contributes to the uniqueness of the bump. One of the reviewers was not convinced of this statement as this constant inhibition does not play a role in the competition between several bumps. Only the inhibitory recurrent connectivity within the HD population or going through the HR populations would play such a role.

We agree with the reviewer on this pont, and the statement has been corrected (see end of second paragraph of section “Neuronal Model” in Methods). Global inhibition indeed suppresses bumps in general. Only sufficient local activity combined with local excitation can overcome the inhibition. Instead, it is the slightly negative recurrent profile for large offsets that contributes to the competition between bumps.

– In Figure S5B, the bump velocity goes to zero around 500º/s. Could this come from a defect in the network that would, for instance, abolish the presence of a bump in left and right HR populations? In a more robust behavior, only one of the left or right bumps would disappear, and the HD bump would go at a max constant velocity.

The reviewers are correct to point out that the behavior of the network would be more robust if it path-integrated at the maximum velocity when the instructed velocity is higher than that. The observed behavior however is not the outcome of some defect; rather, it reflects the fact that, as we detail in Appendix 2, the velocity of bump movement is limited by how far away from the current location the bump can be propagated and how fast it can do that. In Appendix 2 we argue that contributions from HD neurons might be required to move that bump. For velocities larger than the velocity limit, there are no immediate excitatory connections between the currently active HD neurons and the HD neurons that should be activated the next instance (the phase shift required for a given maximum velocity can be obtained by solving eq. 23 for b). Instead, connections are inhibitory for such large offsets, because bump phase shifts larger than the one corresponding to the velocity limit move the bump to the negative sidelobes of the recurrent connectivity. Therefore, without enough excitation to overcome the global inhibition and keep the bump moving, the bump disappears.

[Editors' note: further revisions were suggested prior to acceptance, as described below.]

The manuscript has been much improved but there are some remaining issues related to the lack of evidence for plasticity in CX. After consultation, the reviewers agreed that the current lack of evidence for plasticity should be pointed out earlier in the manuscript, and suggested presenting the need for plasticity in younger flies as a prediction of the model. One of the reviewers provided additional suggestions listed below.Reviewer #2 (Recommendations for the authors):The authors have made a number of improvements to the manuscript. My central concern, which is that there isn't any evidence for this plasticity being present in the CX, still stands. Obviously, a purely modeling study is unable to address this concern. The authors have argued in Appendix 3 that asymmetries in the architecture can severely disrupt the performance of the model and that this supports the need for plasticity. Here are a few suggestions related to this central issue.

We thank the reviewer for their recommendations, and address them subsequently.

1) The authors refer to the performance of flies from Seelig and Jayaraman (2015), stating at various points whether their models outperform the actual performance or not. This should actually be quantified in the figures and/or text. Specifically, both Figure 2 and Appendix 3 would benefit from a quantitative comparison to real fly performance.

We now also quantify PI performance of the network by using the same measure as Seelig and Jayaraman (2015): the correlation coefficient between the PVA and true heading in darkness (see new section “Quantification ofPI performance” in Methods). We report this correlation for the main text network in the Results on Figure 2 (3rd paragraph of “Mature network can path-integrate in darkness”) and extensively in Appendix 3 (see answer to next suggestion). These correlation values allowed us to compare the performance of the network model to that of the fly.

2) It would be useful to show how much imprecision must be added to the synaptic connections before the model can account for the performance of actual flies. This would answer the question of "how much precision must be genetically encoded to account for fly behavior."

We now add noise to all connections. In the new paragraph 5 of Appendix 3, we specify the amount of noise required to drop below fly performance. Furthermore, we comment on the sharpness of performance drop and which weights are more susceptible to noise.

Reviewer #3 (Recommendations for the authors):I find the paper to be timely and very interesting. The paper was well presented already in the first version, but I did have a few concerns. The authors did a great job in addressing all of these concerns and comments in the new version of their manuscript.I strongly recommend it for publications in eLife.

We thank the reviewer for their remarks.